

# What seismicity offshore Sicily suggests about lithosphere dynamics and microplate fragmentation models in the Central Mediterranean

Giancarlo Neri[1], Cristina Totaro[1], Barbara Orecchio[1], Debora Presti[1]

[1] Department of Mathematics, Computer Sciences, Physics, and Earth Sciences,

University of Messina

Viale F. Stagno D'Alcontres, 31

98166 Messina, Italy

Corresponding author:
Giancarlo Neri
University of Messina,
Department of Mathematics, Computer Sciences, Physics, and Earth Sciences,
Viale F. Stagno D'Alcontres, 31
98166 Messina, Italy
email: geoforum@unime.it

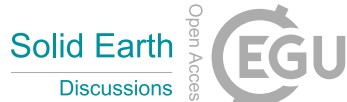

**Abstract**
We analyze an updated dataset of earthquakes of Southern Italy, focusing in particular on
hypocenter locations and seismogenic stress distributions in the southern and eastearn offshores of
Sicily, the two sectors of the study region where seismic and geodetic information needed for
geodynamic modeling is still poor because of poor geometry of monitoring networks. Using
Bayesian non-linear methods for hypocentral locations and hypocenter error estimates we improve
the earthquake locations performed by more traditional linearized techniques, and this helps us to
make significant progress in the interpretation of seismicity and seismogenic stress distributions
especially where seismometric network geometry is more critical. Epicenter maps and hypocenter
vertical sections, together with (i) best quality focal mechanisms coming from seismic waveform
inversion and (ii) orientations of stress principal axes estimated by inversion of focal mechanisms,
help us to better recognize geodynamic engines and plate margin deformation in the study area.
NW-trending convergence between Africa and Eurasia is recognized as the main source of tectonic
stress in the study region, producing clearly detectable signatures in terms of $\sigma_1$ orientations also in
the offshore sectors of the western Ionian and the Sicily Channel. Seismicity and seismogenic stress
tensor highlight nearly uniform compressional dynamics related to plate convergence in the Sicily
Channel, in contrast to rifting and microplate divergence proposed in that sector by other



investigators. In the western Ionian, seismicity and stress inversion results reveal superposition of
convergence-related compression and extensional dynamics. The latter, characterized by minimum
compressive stress oriented SW-NE, can be related to a rifting process (opening SW-NE)
hypothesized by previous investigators on the basis of marine geophysics analyses performed
between the Alfeo-Etna and the Ionian Faults. The seismicity and seismogenic stress detected in the
Western Ionian show that assumptions of microplate rigidity in this area made by previous workers
when modeling poor geodetic data available can be inappropriate. Our findings indicate that more
complex rheologic models should be adopted for reconstruction of tectonic deformation and
microplate relative motions in the Central Mediterranean region.





**1 Introduction**

The progressive increase in the last few decades of available data concerning crustal motions and strains has allowed the researchers to intensify the investigations on the geometry, kinematics and dynamics of the plates, microplates and tectonic units in the Mediterranean region, in particular in the central part of it corresponding to south Italy and the Tyrrhenian and Ionian seas (Figs 1 to 3). In spite of the progresses made in the geophysical data acquisition and analysis, different views still exist, however, concerning the fragmentation of lithosphere and microplate architecture and kinematics in this portion of the Africa-Eurasia convergent margin (see e.g. Anderson and Jackson, 1987; Oldow et al., 2002; Battaglia et al., 2004; Serpelloni et al., 2007; Nocquet, 2012; Sani et al., 2016, among others). A wide description of the debate and state of art of knowledge on these topics is given in Sect. 3.

The relatively large number of models and hypotheses still resisting to checks by new data and analyses is mainly due to poor distribution of GPS stations in the offshore areas and to the often concomitant assumption of internally rigid crustal blocks adopted when modeling the plate/microplate kinematics. The puzzle is complicated by too weak evidence of potential microplate boundaries furnished by earthquake activity resulting scarce as regard to (i) number and energy of seismic events and (ii) geometrical continuity of seismolineaments. Other kinds of data and information from geophysics and geology have not, yet, solved the ambiguities (Oldow et al., 2002; D'Agostino et al., 2008; Nocquet, 2012).

In the present study we use an updated set of earthquake data with the purpose of contributing to the current debate on microplate architecture and dynamics in the Central Mediterranean, paying particular attention to a few sectors offshore Sicily, namely the Western Ionian and the Sicily Channel (WI and SC in Fig. 4a), resulting of crucial relevance for the solution of some still existing

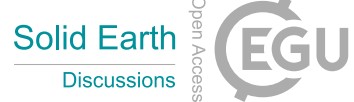

uncertainties. The potentialities of the methods of seismological analysis we have planned to use, in
part of recent conception and in all cases already proven to be effective in the study region (see e.g.
Neri et al., 2005; Billi et al., 2010; D'Amico et al., 2010; Presti et al., 2013), make us confident that
the above declared goal of the investigation may be reached.


**2 Geodynamic frame of the study region**

In the Mediterranean region (Fig. 1), after a long period between Late Paleogene and Neogene of
Africa NW-ward subduction beneath Eurasia, subduction has almost ceased (Billi et al., 2011).
With the progression of Africa-Eurasia convergence, a tectonic reorganization of the plate boundary
has started to accomodate contraction, in particular contractional deformation in several segments
of the boundary (such as Sicily) has shifted from the former subduction zone to the margins of the
back-arc oceanic basins (Fig. 2). The tectonic reorganization of the boundary is still strongly
controlled by the inherited tectonic fabric and rheological attributes, which are strongly
heterogeneous along the boundary.

In Italy (Fig. 1) the Neogene convergence and associated subduction between Africa and Eurasia
resulted in the NW-trending Apennine and the W-trending Maghrebian fold-thrust belts in
peninsular Italy and Sicily, respectively (Malinverno and Ryan, 1986). The two belts are connected
through the Calabrian Arc (Minelli and Faccenna, 2010), below which a narrow remnant of the
former subducting slab seems to be still active, but close to cessation (Neri et al., 2009; 2012;
Orecchio et al., 2015; Chiarabba and Palano, 2017).

Contraction in Sicily is, at present, mainly accomodated at the rear of the fold-thrust belt, in the
southern Tyrrhenian sea (Fig. 2), where a series of contractional earthquakes located in an E-





trending belt between Ustica and Eolian islands have been recorded in the last decades (Pondrelli et
al., 2004; Billi et al., 2007; Presti et al., 2013; Orecchio et al., 2017). Toward the east, the
compressional seismic belt is delimited by the seismically active Tindari Fault System (TFS in Fig.
2), to the east of which, both earthquakes and GPS data provide evidence for an ongoing
extensional tectonics possibly connected with the residual subduction beneath the Calabrian Arc
and related back-arc extension (Palano et al., 2015b). The age for the onset of the ongoing
contraction in the south-Tyrrhenian margin is unknown, but the cessation of volcanism at Ustica
(i.e., the already mentioned volcanic island located along the south-Tyrrhenian contractional belt)
during Middle-Late Pleistocene may be connected with the onset of contractional tectonics in this
area. This age corresponds approximately with the cessation of contractional displacements along
the outermost Gela nappes in southern Sicily (Fig. 2; Ghisetti et al., 2009).

The plate margins strongly simplified in Fig. 1 have undergone significant changes over time as a
consequence of the above processes and of the whole geodynamic activity occurring in the
Mediterranean region. Recent investigations performed thanks to the increasing set of geodetic data
available have brought the researchers to propose different scenarios of microplate fragmentation
and reorganization along the boundary, in particular along the south Italy part of the boundary (see
references quoted in the introduction). The next section presents a list of microplate architecture
scenarios proposed in the most relevant literature and evidences the doubts and open questions still
exixting in the scientific community concerning the microplate geometry and kinematics in the
Italian part of the Africa-Eurasia plate boundary.


**3 The microplate puzzle of the Central Mediterranean**





Two of the former studies reporting on the possible fragmentation of lithosphere in the Italian
portion of the Africa-Eurasia boundary were those by Anderson (1987) and Anderson and Jackson
(1987). These authors explained the focal mechanisms and related slip vectors of the most
significant earthquakes located around the Adriatic sea assuming that a rigid block (Adria) is
independent from the two main plates and rotates anticlockwise around a pole located in northern
Italy (Fig. 3b). They, however, remarked the main weakness of their model represented by the lack
of significant seismicity at the presumed southern border of Adria microplate. Oldow et al. (2002)
and Oldow and Ferranti (2006) proposed a quite different architecture of microplates and blocks in
the Italian region, with a southeastern 'Adria' block separated from a northwestern 'Adria' one by a
line crossing south Italy and the northern Adriatic sea (Fig. 3c).

Closer to Anderson and Jackson's (1987) reconstruction, Battaglia et al. (2004) explained GPS data
re-introducing an Adriatic domain strictly corresponding to the Adriatic sea, independent from
Africa and Eurasia, but separated in two blocks, Northern and Southern Adria, confining in the
Central Adriatic sea (Fig. 3d). Serpelloni et al. (2007) proposed an even more complex
fragmentation of the whole plate boundary, suggesting in particular that a Sicilian domain is
moving independently from Africa according to the presence of a right-lateral and extensional
decoupling zone corresponding to the Tunisia-Libya and Sicily Channel deformation zone (Fig. 3e).
According to the same authors, the tectonics and kinematics of the Italian region are further
complicated by Ionian oceanic lithosphere subducted beneath Calabria and by Adria independent
microplate rotating anticlockwise with respect to Africa and Eurasia main plates. Serpelloni and
coworkers remark, however, that lack of GPS data and poor seismic network geometry in wide
offshore sectors like the Ionian basin leave some uncertainties in the geodynamic modeling, in
particular they suspect (but cannot prove) the existence of an independent Ionian microplate
rotating counterclockwise between Africa and Adria plates (Fig. 3e). In this connection the same
authors leave open the question "where strain locates between Hyblean and Apulia domains?".





D'Agostino et al. (2008) interpreted their GPS and earthquake slip data (i) by limiting the
anticlockwise rotating rigid Adria microplate to the northern Adriatic region and (ii) by introducing
a larger, newly defined microplate to the south, including the Apulia promontory, the Ionian sea and
the Hyblean region in southern Sicily (Fig. 3f). According to these authors, the hypothesis of
Hyblean region belonging to such a hypothetical microplate would be supported by apparently low
GPS-derived deformation in the western Ionian. In any case the authors admit that lack of data in
the Ionian offshore of Sicily does not allow decisive checking of their assumption of two rigid
microplates located between Africa and Eurasia, one rotating clockwise (Apulia-Ionian-Hyblean)
and the other anticlockwise (Adria).

More recently, by analysis of a relatively long period of 18 years of GPS observations, Palano et al.
(2012) supported the thesis of an Hyblean block independent with respect to Africa and Apulia (Fig.
3g-h). They located the regional contraction existing between the Tyrrenian and Hyblean blocks in
two distinct belts identified in the northern Sicily offshore (Ustica-Eolie) and across the Sicily front
(Sicilian basal thrust according to Lavecchia et al., 2007). The authors discussed also the role
played by the Ionian domain and suggested two possible scenarios, one assuming that the Ionian is
rigidly connected with the Hyblean block (Fig. 3g), the other assuming that the Ionian domain
diverges from the Hyblean block and moves to northeast wrt to Eurasia (Fig. 3h). They concluded
that the lack of islands (i.e. of data) in the Ionian offshore does not allow to make a choice among
these scenarios. Based on the analysis of GPS velocities and earthquake focal mechanisms in the
Central and Eastern Mediterranean, Pérouse et al. (2012) proposed that the area including the
Hyblean Plateau, the Ionian basin, the Apulian peninsula, the south Adriatic sea and the Sirte plain
may be considered as a single rigid block rotating clockwise wrt Africa, inducing an opening of a
couple of mm/yr in the Sicily Channel - Pelagian rift (Fig. 3i). The same authors, similar to
D'Agostino et al. (2008 and 2011), suggested that the 2-2.5 mm/yr slow trenchward motion of the
Calabrian Arc wrt to the Apulian-Ionian-Hyblean-Sirte domain may be explained in terms of



ultraslow residual subduction or, alternatively, by pure gravitational collapse into an inactive
subduction scenario. More recently, the Oldow et al's (2002) scheme of Fig. 3c was reproposed
with modifications by Sani et al. (2016) (Fig. 3j).

In his review of papers and investigations regarding the crustal kinematics in the Mediterranean
region, Nocquet (2012) drew the conclusion that it is quite difficult to state from the available data
and analyses where stable Africa finishes and other eventual blocks like Apulia begin in the Central
Mediterranean area at the longitude of Italy. In a very recent analysis performed by integration of
multibeam, seismic reflection, magnetic and gravity data, Polonia et al. (2017) have concluded that:
(i) tearing at the southwestern edge of the SEward-retreating Calabria subduction slab may be the
deep source of shallow deformation detected in correspondence of the transtensional Ionian Fault in
the Ionian basin (Fig. 2); (ii) the NW-SE trending belt comprised between the Ionian Fault and the
Alfeo-Etna Fault (Fig. 2) hosts a rifting zone opening SW-NE where serpentinite diapirs have been
identified by the authors. On their hand, Gutscher et al. (2017), by analysis of multi-beam
bathymetric data and seismic profiles, proposed the Alfeo-Etna right-lateral fault system as shallow
tectonic expression of the retreating slab tear or STEP fault. In the view of these authors some
sinistral lateral component appearing in the southeasternmost part of the Ionian Fault should
exclude the latter as potential expression of the STEP fault. Another major fault system marking the
transition between the Ionian basin and the Hyblean plateau, e.g. the Malta escarpment (Fig. 2),
shows not to be currently active along most of its lenght and shows signs of recent faulting only in
its northernmost segment (see, e.g., Argnani, 2009; Gutscher et al., 2016). West of the Malta
escarpment (Fig. 2), a detailed analysis of reflection profiles and stratigraphic and structural data
allowed Cavallaro et al. (2016) to evidence clear time evolution from tensional to compressive
regimes in the Sicily Channel WNW-trending main structural system, until its present-day
behaviour as transcurrent fault system under compression due to NW-SE Africa-Eurasia
convergence. Cavallaro et al. (2016) proposed, in particular, that local volcanism stopped in late





Miocene and Miocene-time normal faults were reactivated during Zanclean-Piacenzian age as right-
lateral strike-slip faults.

The above description of scenarios and findings highlights uncertainties still existing concerning the
plate boundary evolution and the present-day architecture and kinematics of microplates and
lithospheric blocks in the Central Mediterranean. Poor distribution of GPS and seismic networks in
the offshore sectors of Western Ionian and Sicily Channel cause most of these uncertainties (see,
among others, the already quoted papers by Serpelloni et al., 2007; D'Agostino et al., 2008; Palano
et al., 2012; Nocquet, 2012). In the present study, we start with a regional-scale analysis of recent
seismicity updated to the end of 2016. Then, we focus on the most critical sectors of the Western
Ionian and the Sicily Channel where new seismic data and improvements of knowledge concerning
earthquake and seismogenic stress distribution can help answering the questions left open by the
previous investigations.


**4 Data, methods of analysis and results**

We have taken from the Italian national seismic catalog (http://istituto.ingv.it/l-ingv/archivi-e-
banche-dati/) and from the databases of the local seismic networks operating in Sicily and Calabria
(Orecchio et al., 2011) the data of the earthquakes of magnitude over 2.5 that occurred between
1981 and 2016 at depths less than 100 km in the area 10°-20°E 35°-41°N bounded by the dashed
rectangle in Fig. 4a. We have selected only the events for which a minimum of 15 P+S arrival times
were available. For these events we have performed hypocenter locations by the standard, linearized
location method Simulps (Evans et al., 1994) and the 3D seismic velocity structure proposed for the
study region by Orecchio et al. (2011). The epicenter maps of the earthquakes located with this
procedure, corresponding to different hypocenter depth ranges, are shown in Fig. 4b-d. Then, we





have selected from the literature and the international catalogs all the fault-plane solutions estimated
by waveform inversion for the earthquakes of magnitude over 2.5 occurring in the period 1977-
2016 at depths less than 70 km in the same area of the above locations (dashed rectangle of Fig. 4).
The map of these fault-plane solutions is shown in Fig. 5, the list of focal parameters is furnished in
Table A1, Appendix A.

A new step of analysis was that of using the Bayesian non-linear location algorithm named Bayloc
(Presti et al., 2004 and 2008) for relocation of the earthquakes located by Simulps in the two sectors
indicated by WI (Western Ionian) and SC (Sicily Channel) in Fig. 4a. The same 3D local velocity
structure by Orecchio et al. (2011) has been used also in this new phase of analysis. Starting from
seismic phase arrival times at the recording stations, Bayloc computes for an individual earthquake
a probability cloud marking the hypocenter location uncertainty. Then, Bayloc estimates the spatial
distribution of probability relative to a set of earthquakes by summing the probability densities of
the individual events. This method has been shown to help detection of the seismogenic structures
through better hypocenter location and more accurate estimation of location errors compared to
linearized methods (Presti et al., 2008) but computational reasons make its application easier when
carried out in small areas (Presti et al., 2004). For this reason we have used Simulps for locating the
whole dataset of events of Fig. 4 and Bayloc for locations in the two sectors of more crucial
relevance in the present study, namely WI and SC. Details on the methodological aspects of Bayloc
can be found in the above quoted papers. The epicenter maps and hypocenter vertical sections
obtained by Bayloc are shown in the Figs 6 and 7. Epicenter and hypocentre errors of the order of 3
km and 5 km have been estimated for the earthquakes of Sector WI, rising to values of 4 km and 9
km in Sector SC.

The information furnished in the bibliographic sources of the focal mechanisms of Fig. 5 indicates
that these focal mechanisms should be characterized by fault parameter errors of the order 10-15





degrees, then generally smaller than errors of focal mechanisms computed by inversion of P-onset
polarities in areas of critical network geometry like ours (D'Amico et al., 2011; Presti et al., 2013;
Musumeci et al., 2014). This level of uncertainty makes the dataset of Fig. 5 suitable for application
of the method by Gephart and Forsyth (1984) and Gephart (1990) for calculating the seismogenic
stress tensor directions in the study region. This method searches for the stress tensor showing the
best agreement with the available focal mechanisms (FMs). Four stress parameters are calculated:
three of them define the orientations of the main stress axes; the other is a measure of relative stress
magnitudes, $R = (\sigma_2-\sigma_1)/(\sigma_3-\sigma_1)$, where $\sigma_1$, $\sigma_2$ and $\sigma_3$ are the values of the maximum, intermediate
and minimum compressive stresses, respectively. In order to define discrepancies between the stress
tensor and observations (FMs), a misfit variable is introduced: for a given stress model, the misfit of
a single focal mechanism is defined as the minimum rotation about any arbitrary axis that brings
one of the nodal planes, and its slip direction and sense of slip, into an orientation that is consistent
with the stress model. Searching through all orientations in space by a grid technique operating in
the whole space of stress parameters, the minimum sum of the misfits of all FMs available is found.
The confidence limits of the solution are computed by a statistical procedure described in the papers
by Parker and Mc Nutt (1980) and Gephart and Forsyth (1984). The size of the average misfit
corresponding to the best stress model provides a guide as to how well the assumption of stress
homogeneity is fulfilled (Michael 1987). In the light of results from a series of tests carried out by
Wyss et al. (1992) and Gillard et al. (1996) to identify the relationship between FM uncertainties
and average misfit in the case of uniform stress, we will assume that the condition of homogeneous
stress distribution is fulfilled if the misfit, F, is smaller than 6°, and that it is not fulfilled if F>9°. In
the range 6°<F<9°, the solution is considered as acceptable, but may reflect some heterogeneity.
For the application of Gephart and Forsyth's (1984) method in the present study, we have focused
on the area contoured by the thin line in Fig. 5 including southern Sicily, the Sicily Channel and the
Western Ionian, e.g. the sectors where the knowledge of seismogenic stress distributions is poorer

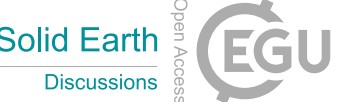

than elsewhere in the region (see, among others, Totaro et al., 2016). The stress inversion results
obtained in the present study are reported in Table 1 and Fig. 8.


**5 Discussion**

The epicenter and focal mechanism maps of Figs 4 and 5 evidence two main features of the regional
seismicity already known from previous investigations. The first of these features is represented by
clear spatial grouping of shallow earthquakes in correspondence of the Apennine-Maghrebian chain
from the southern Apennines to Sicily (Fig. 4c) marking response to perpendicular-to-chain
extensional stress (Fig. 5). The second feature is the nearly east-trending compressional seismic belt
appearing from Figs 4c and 5 in the southern Tyrrhenian sea offshore Sicily, ending to east in the
Eolian Islands area where the orientation of the belt and the faulting style clearly change to NW-SE
and dextral strike-slip, respectively. Moderate activity can be detected from the same figures in the
Sicily Channel (compressional regime) and in the Western Ionian sea (compression prevailing but
not exclusive). Also, a clear drop of activity can be noted in the Ionian offshore of Southern
Calabria (Figs 4c and 5). These features confirm the picture of regional seismicity given by the
previous investigators who distinguished a compressional domain of the southern Tyrrhenian and
Ionian seas due to Africa-Eurasia NW-trending convergence and an extensional one along the
Apennine-Maghrebian chain due to different factors in the different segments of the chain (see e.g.,
Presti et al., 2013; Totaro et al., 2016; Orecchio et al., 2017).

The previous sections have described the efforts made by many investigators to identify opening
zones between diverging microplates in the Sicily Channel and the Ionian sea with the purpose of
explaining the space variations of crustal motions measured by GPS networks in the Central
Mediterranean region. Lack of data in wide offshore sectors is the main reason why the debate on



microplate geometry and kinematics in the region is still open. The results of the present study may,
in our opinion, furnish a useful contribution to this debate. Stress tensor inversion of earthquake
fault-plane solutions in the area including Southern Sicily, the Sicily Channel and the Western
Ionian sea (Fig. 8a, set ALL) reveals moderate stress heterogeneity (F-value of 8.3°) around a best
model of stress characterized by a sub-horizontal, NW-trending $\sigma_1$ clearly reconductible to Africa-
Eurasia convergence (Fig. 8a and Table 1). Starting from this result, we have sub-divided the
dataset of focal mechanisms of Fig. 8a in tens of subsets according to the epicenter distribution,
focal depth and magnitude of the earthquakes, in order to search for subsets satisfying the condition
of stress homogeneity. As explained in the previous Section, this condition can be considered
reasonably satisfied in the present study when the F-value of inversion is lower than 6°. In order to
guarantee the significance of stress computations in the different subsets we have decided to fix a
minimum number of 20 earthquakes (= focal mechanisms) for the creation of the individual subset.
The stress inversion results obtained after the first step of partitioning of the dataset into different
subsets indicated new strategies of partitioning or data grouping. For sake of conciseness, we do not
report the stress inversion results obtained for all the subsets investigated, we only report in Fig. 8
and Table 1 the results that we consider more meaningful, that is, able to better outline the stress
patterns and tectonic features in the study area.

Fig. 8b and Table 1 (lines W and E) report the stress inversion results obtained by subdividing the
study area in two sectors W and E located, respectively, west and east of a NW-trending separation
line indicated as AB in the same Fig. 8b. A F-value of 5.9° shows that stress is homogeneous or
close to homogeneity in the W sector (Table 1). The best model of stress in this western sector is
similar to that obtained by inversion of the mechanisms of the whole study area (Fig. 8a and Table
1). The 95% confidence limits of stress orientations in W (Fig. 8b) show an acceptable level of
constraint of $\sigma_1$ orientation. On the other hand, the confidence area of $\sigma_3$ orientation in the same
sector is relatively large and extends from vertical to SW-NE horizontal direction, i.e. $\sigma_3$ and $\sigma_2$ are



unconstrained on the SW-NE vertical plane, plausibly in relation to co-existence of reverse and
strike-slip seismic faulting in the study volume under SE-NW compression due to plate
convergence. These results are in good agreement with the geostructural and geodynamic
reconstruction of this area proposed by Cavallaro et al. (2016) who evidenced, in particular, long
term evolution from tensional to compressive regimes in the Sicily Channel until the present-day
state of compression led by NW-SE Africa-Eurasia convergence.

The inversion of the fault-plane solutions available in sector E of Fig. 8b leads to a F-value of 8.3°
indicating a certain degree of stress heterogeneity as already inferred from the analysis of the whole
dataset of Fig. 8a. The best model of stress in sector E (Fig. 8b and Table 1) is quite similar to those
of sets ALL and W (Table 1), and this indicates that NW-SE compression from plate convergence is
again detectable in this eastern part of the study area. However, the 95% confidence limits of stress
orientations in E reveal that $\sigma_1$ orientation is practically unconstrained from horizontal NW-SE to
vertical, and this leads us to suppose that some extensional process opening SW-NE acts together
with NW-trending plate convergence in this sector.

By comparing the location of the separation line between the E and W domains of Fig. 8b with the
structural map of Fig. 2 we can note that the separation line approximately corresponds with the
location of the Alfeo-Etna Fault. Again, by comparing the Bayloc's epicenter distribution of the
earthquakes shallower than 30 km in Sector WI (Fig. 6c) with the structural map of Fig. 2 we may
note (i) a certain degree of activity in correspondence with the northern segment of the Malta
escarpment, the only considered active in this structural system by several authors (see, e.g.,
Argnani, 2009; Gutscher et al., 2016), (ii) a clear belt of activity trending NW-SE between the
Alfeo-Etna and the Ionian faults and (iii) a drop of seismicity going to NE across the Ionian Fault.
In the 30-70 km depth range (Fig. 6d) there is no seismicity in correspondence with the Malta
escarpment and most of seismic activity is dispersed around the Ionian Fault. The SW-NE vertical





section of Fig. 6b highlights that sesmicity is as shallow as 20 km west of the Alfeo-Etna Fault but
deepens to depths of the order of 40 km between the Alfeo-Etna and the Ionian faults, and remains
stable at this depth level for several tens of km around the Ionian Fault. In the same vertical section
the presence of sesmic activity can also be noted in the upper 20 km halfway between the Alfeo-
Etna and the Ionian faults (AEF and IF).

Previous investigators (see, e.g., Polonia et al., 2011; 2016; Palano et al., 2015b) have proposed that
the Ionian fault zone corresponds with the southwestern edge of the Ionian subducting slab dipping
to NW and presumably still affected by slow SE-ward residual rollback. The Ionian fault zone
would represent the shallow expression of a STEP fault (Polonia et al., 2016). The above
commented epicenter and hypocenter distributions of Fig. 6 support this interpretation, evidencing
however that the dislocation process between the subducting slab (northeast) and the adjacent
lithosphere (southwest) is distributed over a relatively wide zone probably because the subduction
and slab rollback kinematics are very slow and do not mimic fast STEP dynamics (Gallais et al.,
2013; Orecchio et al., 2014). The earthquake activity detected in the upper 20 km halfway between
the Alfeo-Etna and Ionian faults (Fig. 6b) occurs in a NW-trending highly heterogeneous and
fractured zone identified by Polonia et al. (2017) as the site of a rifting process with SW-NE
opening direction where serpentinite diapirs rise from deeper depths. This extensional process can
be a cause of stress heterogeneity evidenced by stress inversion in the E domain of Fig. 8b where
SW-NE extension appears to be superimposed to NW-SE compression related to Africa-Eurasia
convergence. The rifting process identified in this part of the Ionian sea by Polonia et al. (2017)
may, therefore, contribute to explain our results from analysis of seismicity and stress inversion of
earthquake fault-plane solutions. This interpretation appears also to match well with the space
variation of GPS vectors in Sicily and Calabria (Fig. 9; vector data from Palano et al., 2012)
showing clear rotation of crustal motions when crossing the onshore prolongation of the NW-
trending zone between the Alfeo-Etna and Ionian faults. Having more than 20 focal mechanisms in



the NW-trending zone where Polonia et al. (2017) detected the rifting process, we have performed a
stress inversion run in this zone (RZ in Fig. 8c). The results (shown in the same Fig. 8c and in Table
1) are similar to those obtained in the E domain of Fig. 8b and confirm our hypothesis of
combination of NW-SE compression due to plate convergence and SW-NE rifting. We have also
attempted to detect possible depth variations of the stress field in this rifting zone RZ and found that
the convergence-related compression seems to prevail at depths deeper than 30 km, while the
combination of the two stress factors appears clearer at shallower depth. However, this latter result
must be considered as uncertain or highly preliminary because the number of data available in the
0-30 km and 30-70 km depth ranges is 15 and 12, respectively, then lower than the threshold of 20
adopted in this work for stress inversion. We do not report these latter results in graphical and
numerical form, we wait for future analyses with more data before drawing conclusions on the
depth variation of stress in the rifting zone of Polonia et al. (2017). We conclude our discussion on
stress inversion results by remarking that partitioning of the dataset according to earthquake
magnitude has not evidenced any relevant change of stress or variation of heterogeneity level
among subsets. An implication of this result is that a quite frequent situation documented since
decades in the literature is not observed in the case of our dataset: we specifically refer to the
situation of stronger earthquakes marking a uniform regional stress locally disturbed, however, by
smaller scale stress heterogeneities marked by weaker events (see, e.g., Rebaï et al., 1992;
Christova, 2015).

The Bayloc's maps of epicenters in Sector SC (Fig. 7) show that seismicity is mainly located in
correspondence with a ca. NNE-SSW trending fault system crossing the WNW-ESE Pelagian rift
area. The depths of these events cover all the range of investigation of our analysis, i.e. 0-70 km.
The seismic activity of this NNE-SSW trending fault system covering this relatively large depth
range has already been detected by previous investigators (Calò and Parisi, 2014). Our results show
also that some activity occurs in the Hyblean corner of Sicily and south of it, but in this case the



seismicity is shallow and never exceeds the depth of 30 km. Minor activity is located in
correspondence with the fault system of the Pelagian rift. These results, together with the nearly
homogeneous stress tensor marking SE-NW plate convergence in sector W of Fig. 8b, do not
support active rifting in the Sicily Channel. In other words our results do not support the hypothesis
of microplate separation and divergence in the Sicily Channel advanced in previous works based on
analysis of poorly distributed geodetic data and microplate rigidity assumptions (see, e.g.,
D'Agostino et al., 2008; Perouse et al., 2012). On the other hand, our results match well with the
results recently obtained in the same area by detailed analysis of reflection profiles and stratigraphic
and structural data indicating clear time evolution from tensional to compressive regimes in the
Sicily Channel WNW-trending main structural system, until its present-day behaviour as
transcurrent fault system under compression due to NW-SE Africa-Eurasia convergence (Cavallaro
et al., 2016). According to these authors, Miocene-time extensional structures of the Sicily Channel
WNW-trending structural system were reactivated during Zanclean-Piacenzian age as right-lateral
strike-slip faults under convergence-related compression.

If we re-enlarge the frame of analysis to include the southern Tyrrhenian region (Fig. 9) we find
additional signature of the SE-NW Africa-Eurasia convergence in the southern Tyrrhenian east-
trending compressional belt located offshore northern Sicily (see also Presti et al., 2013 and Totaro
et al., 2016 for stress orientations in the southern Tyrrhenian seismic belt). On this scale of analysis
our results match well with the very recent Nijholt et al's (2018) conclusion according to which in
this south-central part of the Mediterranean region "the Calabrian arc is now further transitioning
towards a setting dominated by Africa-Eurasia plate convergence, whereas during the past 30 Myrs
slab retreat continually was the dominant factor". Also, our investigation leads us to evidence that:
(i) perpendicular-to-convergence opening in the Ionian sea (between the Alfeo-Etna and Ionian
faults) introduces some seismogenic stress heterogeneity in the eastern compartment of our stress
inversion area (Figs 8b, 8c and 9); (ii) residual, very slow subduction and SE-ward rollback of the



Ionian lithosphere northeast of the Ionian Fault reduces convergence-related compression in the
shallow Ionian offshore of southern Calabria and causes there a detectable drop of seismicity
indicated by the arrow in Fig. 4c. This process of reduced compression can also influence stress
parameters in this sector with particular reference to the Ionian offshore of Calabria, but with the
data available we cannot analyze this aspect in a greater detail. We look at a future availability of
additional data in the eastern sector of Fig. 8b to explore in a greater detail the local space variations
of stress and the contributions by the different tectonic factors.

Our results may contribute to answer some questions put by previous investigators such as, for
example, Serpelloni et al. (2007) (where strain locates between Hyblean and Apulia domains?) or
Palano et al. (2012) (is Ionian rigidly connected with the Hyblean block or diverging from it and
moving to northeast wrt to Europe?). Our results allow us to state that (i) the Western Ionian is a
main site of strain release between Hyblean and Apulia domains and (ii) the Ionian is not "rigidly
connected with the Hyblean block". Following the reasoning of Palano et al. (2012) who concluded
with the proposal of two alternative scenarios reported in our Fig. 3g-h, we tend to favour the
hypothesis of a Ionian block diverging from the Sicilian-Hyblean-Malta block (3h), with the
boundary between the respective blocks located, however, as in Fig. 3g or slightly eastward. Our
results lead us to believe that the rifting process suggested by Polonia et al. (2017) between Alfeo-
Etna and Ionian faults in the Ionian basin should be taken in consideration in the future modeling of
microplate geometry and kinematics. Concerning the doubts of D'Agostino et al. (2008) whether
their assumption of a rigid Apulian-Ionian-Hyblean microplate is correct or not, our results in the
Western Ionian showing evidence of a rifting process as proposed by Polonia et al. (2017) suggest
that it is not.

We strongly feel that the usual assumption of rigid blocks or microplates in the physical modeling
of geodynamic processes should be overcome and different crustal rheologies should be tested





when modeling tectonic deformation and microplate relative motions in the Central Mediterranean
region. For this purpose, we have recently started a research collaboration with the geophysical
team of University of Milan for Finite Element Modeling of tectonic stress and strain distributions
through high-resolution 3D thermo-rheological representation of lithosphere in the region of our
interest. This line of research is grounded on methodological developments described in the papers
by Splendore and Marotta (2013) and Marotta et al. (2015), among others.


**6 Conclusion**

Seismicity spatial patterns, earthquake focal mechanisms and seismogenic stress tensor orientations
in and around Sicily, analyzed by use of seismometric data recorded in the last few decades, mark
the dominant action of Africa-Eurasia NW-oriented plate convergence in this part of the
Mediterranean region. Evidences of other tectonic factors acting together (or superimposed to) plate
convergence are found in the Western Ionian sea where (i) residual slow subduction and SE-ward
trench retreat reduce plate coupling in the Southern Calabria offshore and (ii) a rifting process with
opening direction perpendicular to convergence at the southwestern edge of the subducting slab
adds extensional stress to convergence-related compression in the offshore of Eastern Sicily. No
seismic evidence of active rifting or microplate separation and divergence is detected in the
Pelagian area of the Sicily Channel, where clear signatures of plate convergence are found, in
agreement with findings of recent analyses of reflection profiles and structural data performed in the
specific area (Cavallaro et al., 2016). In other words, our results do not support the hypothesis of
Sicily Channel rifting dynamics advanced by previous investigators using geodetic data under
microplate rigidity assumption (e.g. D'Agostino et al., 2008; Perouse et al., 2012). Also, our results
answer several open questions on tectonic strain in the Western Ionian left by Serpelloni et al.
(2007) (where strain locates between Hyblean and Apulia domains?), or by Palano et al. (2012) (is



Ionian rigidly connected with the Hyblean block or diverging from it and moving to northeast wrt to
Europe?) or, again, by D'Agostino et al. (2008) who admit that lack of GPS data in the Ionian
offshore does not allow decisive checking of their model assuming a rigid Apulia-Ionian-Hyblean
microplate. Even taking into account the intrinsic limitations of our seismic datasets relative to
(possibly short) time intervals of a few decades, but also considering their significance where GPS
data are lacking or poor, the seismic data in our possess indicate that (i) strain "between Hyblean
and Apulia domains" mainly locates around the southwestern edge of the Ionian subduction slab in
the westernmost Ionian, (ii) the Ionian is not "rigidly connected with the Hyblean block" and (iii)
the assumption of "a rigid Apulia-Ionian-Hyblean microplate" needs to be revised. Our results show
that the assumption of rigid blocks or microplates made by previous investigators in their kinematic
reconstructions should be overcome and different crustal rheologies should be tested when
modeling tectonic deformation and microplate relative motions in this region. High-resolution 3D
thermo-rheological representations of lithosphere in the frame of Finite Element Modeling of
tectonic stress and strain distributions can, in our opinion, be an appropriate road towards
geodynamic modeling of Southern Italy and the Central Mediterranean region. We are starting to
work in this direction.



**Appendix A**
**Table A1**: Database of earthquake focal mechanisms of southern Italy and surroundings reported in
Fig. 5. ID is the order number. O.T., Lon, Lat and Depth are the GMT origin time, the longitude E
(°), the latitude N (°) and the focal depth (km) of the earthquake, respectively. Strike, dip, and rake
are the fault parameters in degrees of the focal solution. M is the earthquake magnitude. Source is
the bibliographic source of the solution (Italian CMT= http://rcmt2.bo.ingv.it/Italydataset.html,
CMT=       http://www.globalcmt.org/,      RCMT=       http://rcmt2.bo.ingv.it/,      TDMT=
http://cnt.rm.ingv.it/tdmt; the other sources are reported in the reference list at the end of the
article).

| Id | Data | O.T. | Lon | Lat | Depth | Strike | Dip | Rake | M | Source |
|----|------|------|-----|-----|-------|--------|-----|------|---|--------|
| 1 | 19770605 | 13:59:23 | 14.46 | 37.84 | 11.3 | 61 | 26 | -139 | 4.6 | Italian CMT |
| 2 | 19770815 | 21:10:40 | 16.98 | 38.85 | 40.0 | 307 | 38 | 120 | 5.2 | CMT |
| 3 | 19780311 | 19:20:49 | 16.03 | 38.10 | 15.0 | 270 | 41 | -72 | 5.2 | Italian CMT |
| 4 | 19780415 | 23:33:47 | 14.63 | 37.77 | 34.0 | 135 | 60 | -176 | 6.0 | Italian CMT |
| 5 | 19790120 | 13:49:59 | 12.86 | 38.67 | 9.0 | 72 | 29 | 53 | 5.2 | Italian CMT |
| 6 | 19791208 | 00:06:33 | 11.49 | 37.95 | 15.0 | 235 | 45 | 67 | 5.3 | CMT |
| 7 | 19800220 | 02:34:03 | 16.21 | 39.30 | 12.0 | 14 | 43 | -78 | 4.8 | Italian CMT |
| 8 | 19800309 | 12:03:40 | 16.12 | 39.94 | 19.0 | 157 | 35 | -80 | 4.6 | Italian CMT |
| 9 | 19800514 | 01:41:04 | 15.85 | 40.46 | 24.0 | 119 | 38 | -112 | 4.5 | Italian CMT |
| 10 | 19800528 | 19:51:19 | 14.25 | 38.48 | 19.0 | 83 | 43 | 99 | 5.7 | Italian CMT |
| 11 | 19800601 | 02:32:52 | 14.33 | 38.39 | 10.0 | 65 | 39 | 91 | 4.9 | Italian CMT |
| 12 | 19801123 | 18:34:54 | 15.39 | 40.82 | 14.0 | 135 | 41 | -80 | 6.9 | Italian CMT |
| 13 | 19801124 | 00:24:00 | 15.26 | 40.89 | 10.0 | 131 | 29 | -110 | 4.9 | Italian CMT |
| 14 | 19801124 | 03:03:54 | 15.33 | 40.90 | 10.0 | 115 | 44 | -125 | 5.1 | Italian CMT |
| 15 | 19801125 | 17:06:44 | 15.47 | 40.70 | 10.0 | 122 | 30 | -119 | 5.1 | Italian CMT |
| 16 | 19801125 | 18:28:21 | 15.36 | 40.15 | 15.0 | 129 | 26 | -65 | 5.4 | Italian CMT |
| 17 | 19801203 | 23:54:24 | 15.48 | 40.74 | 10.0 | 148 | 36 | -76 | 4.9 | Italian CMT |
| 18 | 19810116 | 00:37:47 | 15.23 | 40.13 | 15.0 | 115 | 30 | -93 | 5.2 | Italian CMT |
| 19 | 19810607 | 13:00:57 | 12.47 | 37.67 | 18.0 | 48 | 29 | 48 | 4.9 | Italian CMT |
| 20 | 19810622 | 09:36:18 | 14.09 | 38.49 | 13.0 | 71 | 47 | 116 | 4.8 | Italian CMT |
| 21 | 19811129 | 05:06:47 | 15.64 | 40.74 | 33.0 | 104 | 41 | -138 | 4.9 | Italian CMT |
| 22 | 19820321 | 09:44:00 | 15.64 | 39.70 | 18.9 | 15 | 39 | -127 | 5.0 | Italian CMT |
| 23 | 19820815 | 15:09:50 | 15.36 | 40.81 | 10.0 | 158 | 48 | -45 | 4.8 | Italian CMT |
| 24 | 19821116 | 23:41:29 | 19.35 | 40.12 | 10.0 | 297 | 35 | 54 | 5.6 | CMT |
| 25 | 19870128 | 05:33:22 | 15.47 | 40.95 | 10.0 | 160 | 45 | -79 | 4.6 | Italian CMT |
| 26 | 19870813 | 07:22:10 | 15.06 | 37.90 | 35.9 | 352 | 42 | -10 | 4.8 | Italian CMT |
| 27 | 19880108 | 13:05:46 | 16.01 | 40.08 | 10.0 | 148 | 30 | -86 | 4.8 | Italian CMT |
| 28 | 19880109 | 01:02:48 | 19.49 | 40.37 | 15.0 | 321 | 12 | 62 | 5.9 | CMT |


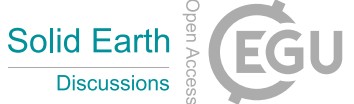

| 29 | 19880518 | 05:17:40 | 19.88 | 37.70 | 23.0 | 163 | 38 | 95 | 5.3 | CMT |
| 30 | 19890103 | 16:52:24 | 11.80 | 35.79 | 15.0 | 247 | 90 | 180 | 5.1 | CMT |
| 31 | 19890824 | 02:13:12 | 19.72 | 37.05 | 15.0 | 356 | 38 | 131 | 5.2 | CMT |
| 32 | 19900505 | 07:21:19 | 15.58 | 40.24 | 26.0 | 184 | 73 | 13 | 5.8 | Italian CMT |
| 33 | 19900505 | 07:38:12 | 15.81 | 40.75 | 15.0 | 282 | 83 | 173 | 5.0 | Italian CMT |
| 34 | 19901029 | 08:16:14 | 14.67 | 36.23 | 23.0 | 198 | 72 | -13 | 4.5 | Italian CMT |
| 35 | 19901213 | 00:24:24 | 14.90 | 37.25 | 15.0 | 274 | 64 | 174 | 5.6 | Italian CMT |
| 36 | 19910526 | 12:26:00 | 15.77 | 40.73 | 8.0 | 183 | 71 | -9 | 5.2 | Italian CMT |
| 37 | 19920123 | 04:24:20 | 19.97 | 38.22 | 15.0 | 351 | 42 | 97 | 5.6 | CMT |
| 38 | 19920406 | 13:08:34 | 14.61 | 37.83 | 21.0 | 100 | 37 | -97 | 4.7 | Italian CMT |
| 39 | 19930626 | 17:47:54 | 14.21 | 37.92 | 10.0 | 170 | 53 | 6 | 4.4 | Italian CMT |
| 40 | 19940416 | 23:09:39 | 19.99 | 36.99 | 15.0 | 340 | 18 | 134 | 5.5 | CMT |
| 41 | 19950529 | 06:52:27 | 12.07 | 37.90 | 11.0 | 82 | 70 | -180 | 4.8 | Italian CMT |
| 42 | 19960201 | 17:58:02 | 19.57 | 37.84 | 15.0 | 156 | 48 | 49 | 5.3 | CMT |
| 43 | 19960403 | 13:04:34 | 15.49 | 40.76 | 10.0 | 123 | 30 | -110 | 4.9 | Italian CMT |
| 44 | 19961214 | 00:18:45 | 13.84 | 37.81 | 40.0 | 123 | 23 | -43 | 4.7 | Italian CMT |
| 45 | 19970112 | 12:10:51 | 19.67 | 40.96 | 10.0 | 4 | 70 | -167 | 4.9 | RCMT |
| 46 | 19970119 | 19:42:38 | 19.67 | 40.82 | 10.0 | 2 | 65 | -177 | 4.7 | RCMT |
| 47 | 19970325 | 00:46:14 | 16.03 | 36.93 | 33.0 | 104 | 78 | 179 | 4.5 | Italian CMT |
| 48 | 19971202 | 19:22:47 | 19.40 | 36.68 | 15.0 | 206 | 49 | -20 | 5.2 | CMT |
| 49 | 19980117 | 12:32:51 | 12.90 | 38.40 | 10.0 | 58 | 29 | 71 | 4.8 | Italian CMT |
| 50 | 19980620 | 02:25:47 | 13.08 | 38.46 | 10.0 | 69 | 22 | 76 | 5.2 | Italian CMT |
| 51 | 19980621 | 08:59:47 | 13.10 | 38.50 | 10.0 | 69 | 36 | 77 | 4.6 | Italian CMT |
| 52 | 19980621 | 12:59:04 | 12.67 | 38.43 | 10.0 | 88 | 38 | 102 | 4.6 | Italian CMT |
| 53 | 19980909 | 11:27:59 | 16.07 | 39.67 | 15.0 | 139 | 29 | -83 | 5.6 | Italian CMT |
| 54 | 19980914 | 05:24:47 | 13.60 | 38.46 | 10.0 | 72 | 30 | 80 | 5.0 | Italian CMT |
| 55 | 19990214 | 11:45:54 | 15.06 | 38.17 | 33.0 | 18 | 39 | -108 | 4.7 | Italian CMT |
| 56 | 19991030 | 07:09:09 | 19.98 | 37.76 | 5.0 | 80 | 8 | -173 | 4.8 | RCMT |
| 57 | 19991124 | 21:10:49 | 19.76 | 40.27 | 10.0 | 141 | 39 | 101 | 4.3 | RCMT |
| 58 | 20000426 | 13:37:48 | 10.10 | 40.98 | 10.0 | 179 | 39 | 83 | 4.8 | RCMT |
| 59 | 20000428 | 17:41:02 | 19.70 | 37.83 | 33.0 | 305 | 64 | -4 | 4.5 | RCMT |
| 60 | 20000627 | 04:07:56 | 10.03 | 40.95 | 10.0 | 184 | 27 | 82 | 4.3 | RCMT |
| 61 | 20010411 | 00:10:28 | 16.12 | 40.43 | 16.6 | 130 | 20 | -90 | 3.0 | Orecchio&al.(2014) |
| 62 | 20010417 | 00:42:36 | 19.47 | 40.64 | 10.0 | 343 | 43 | 97 | 4.2 | RCMT |
| 63 | 20010422 | 13:56:36 | 15.10 | 37.72 | 10.0 | 316 | 56 | 27 | 4.2 | Italian CMT |
| 64 | 20010526 | 06:02:20 | 16.34 | 37.46 | 33.0 | 71 | 54 | 134 | 4.5 | Italian CMT |
| 65 | 20010930 | 23:44:58 | 16.04 | 40.22 | 12.6 | 30 | 70 | -80 | 2.7 | Orecchio&al.(2014) |
| 66 | 20011018 | 11:02:44 | 16.61 | 39.10 | 10.0 | 332 | 44 | -88 | 4.3 | Italian CMT |
| 67 | 20011125 | 19:34:20 | 13.96 | 37.91 | 20.0 | 137 | 31 | -57 | 4.7 | Italian CMT |
| 68 | 20020405 | 04:52:24 | 14.74 | 38.48 | 10.0 | 90 | 41 | 108 | 4.4 | Italian CMT |
| 69 | 20020417 | 06:42:54 | 16.67 | 39.37 | 15.0 | 141 | 28 | 13 | 5.3 | Totaro&al.(2016) |
| 70 | 20020418 | 20:56:48 | 15.58 | 40.69 | 10.0 | 340 | 49 | -52 | 4.4 | Italian CMT |
| 71 | 20020624 | 01:20:43 | 10.29 | 36.03 | 15.0 | 28 | 48 | 128 | 5.2 | CMT |
| 72 | 20020906 | 01:21:29 | 13.57 | 38.42 | 15.0 | 26 | 50 | 40 | 5.9 | Italian CMT |
| 73 | 20020906 | 01:45:30 | 13.73 | 38.44 | 4.0 | 252 | 48 | 126 | 4.7 | Italian CMT |
| 74 | 20020909 | 06:06:37 | 19.98 | 37.63 | 33.0 | 12 | 61 | -165 | 4.4 | RCMT |

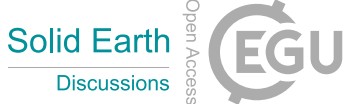

| 75 | 20020910 | 02:32:51 | 13.70 | 38.47 | 5.0 | 71 | 29 | 126 | 4.4 | Italian CMT |
| 76 | 20020920 | 23:06:04 | 13.74 | 38.46 | 5.0 | 46 | 33 | 77 | 4.7 | Italian CMT |
| 77 | 20020927 | 06:10:45 | 13.66 | 38.41 | 15.0 | 35 | 24 | 65 | 5.2 | Totaro&al.(2016) |
| 78 | 20020928 | 02:46:46 | 13.71 | 38.47 | 5.0 | 79 | 39 | 103 | 4.6 | Italian CMT |
| 79 | 20021002 | 22:57:26 | 13.72 | 38.46 | 5.0 | 33 | 41 | 59 | 4.9 | Italian CMT |
| 80 | 20021027 | 02:50:26 | 15.16 | 37.79 | 10.0 | 320 | 60 | 171 | 4.9 | Italian CMT |
| 81 | 20021027 | 07:32:09 | 15.18 | 37.92 | 10.0 | 67 | 54 | 19 | 4.5 | Italian CMT |
| 82 | 20021029 | 10:02:22 | 15.27 | 37.67 | 10.0 | 316 | 61 | -173 | 4.7 | Italian CMT |
| 83 | 20021029 | 16:39:48 | 15.56 | 37.69 | 10.0 | 207 | 54 | -28 | 4.2 | Italian CMT |
| 84 | 20021209 | 09:35:06 | 19.97 | 37.87 | 15.0 | 255 | 28 | -20 | 5.2 | CMT |
| 85 | 20030616 | 08:27:51 | 19.73 | 37.69 | 10.0 | 321 | 31 | 43 | 4.7 | RCMT |
| 86 | 20030707 | 15:08:12 | 14.90 | 36.01 | 10.0 | 350 | 62 | 4 | 4.3 | Italian CMT |
| 87 | 20041011 | 07:31:41 | 15.48 | 37.88 | 6.6 | 89 | 90 | -45 | 3.6 | D'Amico&al.(2010) |
| 88 | 20041022 | 21:10:13 | 15.32 | 38.08 | 10.7 | 78 | 61 | -37 | 3.4 | D'Amico&al.(2010) |
| 89 | 20041218 | 09:12:48 | 10.15 | 40.89 | 10.0 | 144 | 49 | 51 | 4.6 | RCMT |
| 90 | 20050131 | 01:05:35 | 19.97 | 37.37 | 12.0 | 346 | 16 | 120 | 5.7 | CMT |
| 91 | 20050131 | 10:44:50 | 16.86 | 39.66 | 30.0 | 23 | 79 | -41 | 4.1 | D'Amico&al.(2011) |
| 92 | 20050207 | 20:05:37 | 10.91 | 36.10 | 10.0 | 67 | 44 | 128 | 4.8 | RCMT |
| 93 | 20050207 | 20:46:26 | 10.87 | 36.22 | 10.0 | 51 | 41 | 124 | 5.1 | RCMT |
| 94 | 20050212 | 12:13:45 | 19.70 | 39.92 | 0.0 | 187 | 42 | 53 | 4.2 | RCMT |
| 95 | 20050419 | 22:36:23 | 15.66 | 38.14 | 7.1 | 220 | 42 | -10 | 3.1 | D'Amico&al.(2010) |
| 96 | 20050423 | 19:10:48 | 15.82 | 38.43 | 13.6 | 120 | 50 | -64 | 2.8 | D'Amico&al.(2010) |
| 97 | 20050423 | 19:11:43 | 16.71 | 39.47 | 23.0 | 128 | 58 | 14 | 4.1 | D'Amico&al.(2011) |
| 98 | 20050602 | 03:05:51 | 15.31 | 39.59 | 7.0 | 278 | 73 | -172 | 3.7 | Li&al.(2007) |
| 99 | 20050721 | 15:41:43 | 14.85 | 39.40 | 7.0 | 184 | 68 | 41 | 3.8 | Li&al.(2007) |
| 100 | 20050818 | 22:02:27 | 15.12 | 37.80 | 6.7 | 82 | 50 | -18 | 3.1 | D'Amico&al.(2010) |
| 101 | 20050907 | 12:40:33 | 16.32 | 38.71 | 16.0 | 80 | 90 | -42 | 3.6 | D'Amico&al.(2011) |
| 102 | 20050927 | 22:33:09 | 17.10 | 38.62 | 29.0 | 38 | 79 | 141 | 3.9 | Li&al.(2007) |
| 103 | 20051030 | 19:09:47 | 15.93 | 38.53 | 22.0 | 241 | 66 | -84 | 3.4 | Li&al.(2007) |
| 104 | 20051118 | 18:35:25 | 17.07 | 39.17 | 23.0 | 120 | 34 | 3 | 3.6 | D'Amico&al.(2011) |
| 105 | 20051121 | 10:57:41 | 14.16 | 37.61 | 61.0 | 102 | 79 | 179 | 4.6 | RCMT |
| 106 | 20051203 | 08:33:02 | 17.00 | 39.20 | 15.0 | 290 | 64 | -18 | 3.8 | Presti&al.(2013) |
| 107 | 20060107 | 22:08:44 | 17.21 | 39.27 | 16.0 | 174 | 69 | 61 | 3.2 | Orecchio&al.(2014) |
| 108 | 20060117 | 03:33:58 | 17.13 | 39.20 | 34.0 | 146 | 62 | -21 | 3.7 | Orecchio&al.(2014) |
| 109 | 20060227 | 04:34:01 | 15.20 | 38.15 | 9.0 | 62 | 50 | -71 | 4.1 | D'Amico&al.(2010) |
| 110 | 20060227 | 09:11:59 | 15.18 | 38.14 | 10.5 | 39 | 48 | -90 | 3.1 | D'Amico&al.(2010) |
| 111 | 20060227 | 14:16:06 | 15.18 | 38.14 | 9.1 | 76 | 48 | -58 | 3.1 | D'Amico&al.(2010) |
| 112 | 20060329 | 20:20:00 | 13.89 | 37.73 | 10.0 | 338 | 80 | -42 | 3.9 | TDMT |
| 113 | 20060417 | 02:44:06 | 17.14 | 39.57 | 10.0 | 114 | 74 | -3 | 4.4 | D'Amico&al.(2011) |
| 114 | 20060423 | 14:42:38 | 15.02 | 37.04 | 24.0 | 100 | 88 | 147 | 3.9 | TDMT |
| 115 | 20060510 | 00:44:33 | 19.31 | 40.23 | 5.0 | 292 | 72 | 10 | 4.5 | RCMT |
| 116 | 20060510 | 21:33:06 | 19.78 | 40.06 | 18.0 | 289 | 59 | 7 | 4.3 | RCMT |
| 117 | 20060520 | 07:05:56 | 14.95 | 37.65 | 12.0 | 280 | 75 | 47 | 3.7 | TDMT |
| 118 | 20060525 | 23:14:41 | 19.91 | 36.55 | 23.0 | 346 | 23 | 129 | 5.2 | CMT |
| 119 | 20060530 | 11:30:40 | 16.52 | 37.63 | 46.0 | 347 | 85 | 0 | 4.5 | RCMT |
| 120 | 20060613 | 14:15:38 | 19.96 | 40.27 | 10.0 | 283 | 67 | 25 | 4.7 | RCMT |



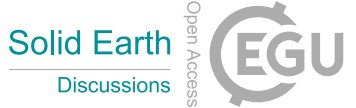

| 121 | 20060619 | 20:55:35 | 14.89 | 37.83 | 16.0 | 350 | 28 | 11 | 3.8 | Neri&al.(2014) |
| 122 | 20060619 | 21:20:13 | 14.88 | 37.83 | 18.0 | 354 | 32 | 2 | 3.1 | Totaro&al.(2016) |
| 123 | 20060619 | 21:27:12 | 14.87 | 37.82 | 20.0 | 0 | 37 | 9 | 3.1 | Totaro&al.(2016) |
| 124 | 20060620 | 13:16:36 | 14.86 | 37.83 | 18.0 | 0 | 26 | 22 | 3.3 | Totaro&al.(2016) |
| 125 | 20060621 | 07:17:50 | 14.83 | 37.83 | 18.0 | 354 | 23 | 11 | 3.3 | Totaro&al.(2016) |
| 126 | 20060622 | 19:34:58 | 16.60 | 39.73 | 15.0 | 110 | 33 | -33 | 4.4 | D'Amico&al.(2011) |
| 127 | 20060702 | 17:52:00 | 15.10 | 38.13 | 10.0 | 70 | 59 | -49 | 2.6 | D'Amico&al.(2010) |
| 128 | 20060718 | 07:42:40 | 15.17 | 38.12 | 9.1 | 90 | 41 | -48 | 3.1 | D'Amico&al.(2010) |
| 129 | 20060730 | 09:53:36 | 16.31 | 37.99 | 6.0 | 292 | 64 | -7 | 2.7 | Orecchio&al.(2014) |
| 130 | 20060805 | 20:47:19 | 14.73 | 38.55 | 10.0 | 40 | 30 | 39 | 3.3 | Totaro&al.(2016) |
| 131 | 20060806 | 07:49:49 | 19.53 | 40.05 | 16.0 | 20 | 44 | 146 | 4.8 | CMT |
| 132 | 20060808 | 21:20:12 | 19.60 | 40.04 | 20.0 | 13 | 38 | 138 | 4.8 | CMT |
| 133 | 20060809 | 02:10:01 | 19.59 | 40.11 | 19.0 | 312 | 69 | 16 | 4.2 | RCMT |
| 134 | 20060819 | 16:29:11 | 14.43 | 38.58 | 26.0 | 229 | 78 | 28 | 3.3 | Totaro&al.(2016) |
| 135 | 20060830 | 22:45:03 | 15.72 | 37.32 | 30.0 | 190 | 64 | -23 | 3.1 | Orecchio&al.(2014) |
| 136 | 20060901 | 21:50:41 | 19.87 | 40.05 | 50.0 | 129 | 64 | 63 | 4.4 | present work |
| 137 | 20060907 | 15:31:43 | 16.19 | 40.57 | 34.0 | 178 | 55 | 35 | 4.1 | Italian CMT |
| 138 | 20060909 | 15:45:23 | 14.23 | 38.70 | 8.0 | 222 | 68 | 13 | 3.3 | Totaro&al.(2016) |
| 139 | 20061006 | 21:16:23 | 15.57 | 38.10 | 9.6 | 18 | 52 | -90 | 3.2 | D'Amico&al.(2010) |
| 140 | 20061022 | 05:13:10 | 16.70 | 39.05 | 10.0 | 312 | 38 | -30 | 3.0 | Orecchio&al.(2014) |
| 141 | 20061104 | 05:59:22 | 15.01 | 38.03 | 10.6 | 59 | 49 | -36 | 3.0 | D'Amico&al.(2010) |
| 142 | 20061107 | 11:13:36 | 18.26 | 35.91 | 35.0 | 201 | 30 | 93 | 4.3 | RCMT |
| 143 | 20061118 | 00:01:56 | 17.23 | 39.06 | 24.0 | 113 | 46 | -51 | 3.0 | Orecchio&al.(2014) |
| 144 | 20061123 | 13:31:56 | 12.94 | 35.97 | 10.0 | 357 | 70 | -2 | 4.8 | RCMT |
| 145 | 20061124 | 04:37:40 | 15.76 | 36.26 | 11.0 | 188 | 82 | 0 | 4.4 | Italian CMT |
| 146 | 20061219 | 14:58:06 | 14.91 | 37.78 | 23.0 | 18 | 16 | -40 | 4.2 | Italian CMT |
| 147 | 20061220 | 11:38:08 | 14.26 | 38.54 | 6.0 | 201 | 64 | 13 | 3.6 | Neri&al.(2014) |
| 148 | 20061226 | 00:49:00 | 16.17 | 39.22 | 2.0 | 223 | 38 | -12 | 3.1 | Orecchio&al.(2014) |
| 149 | 20070130 | 22:18:07 | 16.14 | 39.91 | 8.0 | 84 | 90 | 19 | 3.6 | Orecchio&al.(2014) |
| 150 | 20070202 | 06:51:01 | 16.35 | 39.55 | 12.0 | 31 | 48 | -61 | 3.2 | Totaro&al.(2013) |
| 151 | 20070326 | 13:55:26 | 16.96 | 39.28 | 20.0 | 301 | 61 | 8 | 3.7 | Totaro&al.(2016) |
| 152 | 20070410 | 19:17:23 | 12.91 | 36.93 | 22.0 | 100 | 75 | 164 | 4.1 | TDMT |
| 153 | 20070421 | 19:41:27 | 13.43 | 38.57 | 12.0 | 250 | 52 | 10 | 3.9 | Neri&al.(2014) |
| 154 | 20070426 | 00:49:36 | 16.37 | 39.54 | 16.0 | 290 | 8 | 20 | 3.8 | Orecchio&al.(2014) |
| 155 | 20070503 | 18:43:56 | 17.63 | 39.02 | 18.0 | 92 | 25 | -31 | 3.5 | Orecchio&al.(2014) |
| 156 | 20070517 | 05:48:13 | 14.69 | 38.57 | 8.0 | 22 | 50 | 8 | 3.5 | Presti&al.(2013) |
| 157 | 20070525 | 09:39:45 | 16.83 | 39.66 | 12.0 | 91 | 29 | -48 | 4.2 | D'Amico&al.(2011) |
| 158 | 20070609 | 05:56:38 | 16.62 | 39.18 | 18.0 | 71 | 49 | -59 | 3.1 | Orecchio&al.(2014) |
| 159 | 20070615 | 22:56:01 | 15.29 | 36.97 | 18.0 | 12 | 87 | 20 | 3.6 | TDMT |
| 160 | 20070617 | 12:11:58 | 15.79 | 38.37 | 10.0 | 262 | 38 | -43 | 2.9 | D'Amico&al.(2010) |
| 161 | 20070706 | 23:28:43 | 17.25 | 39.18 | 28.0 | 118 | 38 | -35 | 3.5 | Orecchio&al.(2014) |
| 162 | 20070714 | 18:13:03 | 14.75 | 38.63 | 4.0 | 30 | 31 | 38 | 3.1 | Presti&al.(2013) |
| 163 | 20070731 | 06:53:16 | 14.74 | 37.47 | 32.0 | 142 | 78 | -21 | 3.4 | Totaro&al.(2016) |
| 164 | 20070801 | 00:07:54 | 17.18 | 39.00 | 18.0 | 80 | 67 | -45 | 4.1 | D'Amico&al.(2011) |
| 165 | 20070818 | 14:04:07 | 15.17 | 38.22 | 12.0 | 44 | 50 | -23 | 3.9 | D'Amico&al.(2010) |
| 166 | 20070818 | 14:21:11 | 15.12 | 38.19 | 10.0 | 26 | 69 | 18 | 3.4 | D'Amico&al.(2010) |





| 167 | 20070905 | 21:24:13 | 14.84 | 38.56 | 6.0 | 10 | 54 | -2 | 3.3 | Totaro&al.(2016) |
| 168 | 20070913 | 15:19:52 | 15.16 | 38.25 | 8.0 | 246 | 82 | -60 | 2.9 | Totaro&al.(2016) |
| 169 | 20070923 | 07:12:46 | 14.79 | 38.59 | 8.0 | 27 | 60 | 28 | 3.6 | Presti&al.(2013) |
| 170 | 20070930 | 15:41:20 | 14.80 | 38.59 | 6.0 | 70 | 73 | 16 | 3.1 | Presti&al.(2013) |
| 171 | 20071213 | 23:38:24 | 16.61 | 38.93 | 30.0 | 98 | 9 | -17 | 3.2 | Orecchio&al.(2014) |
| 172 | 20071214 | 00:42:55 | 16.62 | 38.93 | 26.0 | 331 | 70 | -60 | 3.0 | Orecchio&al.(2014) |
| 173 | 20071217 | 09:44:39 | 16.36 | 39.39 | 40.0 | 162 | 90 | -71 | 3.5 | Orecchio&al.(2014) |
| 174 | 20071220 | 03:25:32 | 16.19 | 39.36 | 2.0 | 210 | 66 | -71 | 3.5 | Orecchio&al.(2014) |
| 175 | 20080115 | 02:38:31 | 16.33 | 39.81 | 18.0 | 327 | 80 | 39 | 3.2 | Orecchio&al.(2014) |
| 176 | 20080118 | 13:01:00 | 16.53 | 39.14 | 12.0 | 57 | 78 | -67 | 3.9 | TDMT |
| 177 | 20080209 | 07:46:36 | 15.56 | 37.84 | 6.9 | 40 | 90 | -10 | 3.0 | D'Amico&al.(2010) |
| 178 | 20080221 | 05:00:09 | 17.97 | 37.82 | 30.0 | 333 | 27 | 134 | 4.7 | Italian CMT |
| 179 | 20080305 | 04:08:21 | 19.67 | 40.22 | 0.0 | 173 | 47 | 152 | 4.1 | RCMT |
| 180 | 20080310 | 10:33:27 | 16.85 | 39.66 | 20.0 | 121 | 39 | -7 | 3.5 | Presti&al.(2013) |
| 181 | 20080408 | 17:20:01 | 16.66 | 39.16 | 10.0 | 235 | 49 | -35 | 4.4 | Italian CMT |
| 182 | 20080413 | 10:10:02 | 16.52 | 39.16 | 14.0 | 205 | 69 | -90 | 3.6 | D'Amico&al.(2010) |
| 183 | 20080413 | 13:06:57 | 15.70 | 38.25 | 14.3 | 6 | 47 | -36 | 2.8 | Orecchio&al.(2014) |
| 184 | 20080414 | 18:44:34 | 16.52 | 39.15 | 12.0 | 48 | 41 | -62 | 3.1 | Orecchio&al.(2014) |
| 185 | 20080419 | 21:41:11 | 17.47 | 39.13 | 16.0 | 107 | 42 | -39 | 3.6 | Presti&al.(2013) |
| 186 | 20080426 | 22:23:06 | 16.53 | 39.14 | 18.0 | 256 | 60 | -31 | 3.2 | Orecchio&al.(2014) |
| 187 | 20080501 | 21:05:49 | 15.07 | 37.80 | 2.0 | 97 | 76 | -2 | 2.8 | D'Amico&al.(2010) |
| 188 | 20080513 | 21:28:30 | 15.06 | 37.80 | 12.0 | 76 | 46 | -20 | 3.5 | D'Amico&al.(2010) |
| 189 | 20080702 | 17:43:33 | 16.23 | 38.97 | 30.0 | 266 | 69 | -30 | 3.2 | Orecchio&al.(2014) |
| 190 | 20080703 | 20:56:52 | 13.71 | 38.45 | 24.2 | 182 | 68 | 27 | 3.3 | TDMT |
| 191 | 20080705 | 17:04:36 | 15.87 | 38.20 | 2.0 | 311 | 59 | 2 | 2.6 | D'Amico&al.(2010) |
| 192 | 20080709 | 23:08:27 | 16.23 | 38.97 | 24.0 | 268 | 76 | -32 | 3.3 | Orecchio&al.(2014) |
| 193 | 20080710 | 01:45:45 | 16.24 | 38.97 | 12.0 | 76 | 72 | -40 | 3.0 | Orecchio&al.(2014) |
| 194 | 20080710 | 12:50:20 | 16.24 | 38.96 | 14.0 | 71 | 75 | -39 | 3.3 | Orecchio&al.(2014) |
| 195 | 20080711 | 07:15:00 | 16.24 | 38.96 | 16.0 | 50 | 52 | -29 | 3.0 | Orecchio&al.(2014) |
| 196 | 20080711 | 07:20:21 | 16.25 | 38.96 | 12.0 | 78 | 80 | -58 | 2.9 | Orecchio&al.(2014) |
| 197 | 20080813 | 13:39:30 | 16.42 | 37.48 | 34.0 | 181 | 71 | 11 | 3.2 | Orecchio&al.(2014) |
| 198 | 20080901 | 14:45:40 | 15.06 | 37.97 | 8.1 | 70 | 31 | -80 | 3.1 | D'Amico&al.(2010) |
| 199 | 20080902 | 09:16:45 | 15.06 | 37.99 | 10.3 | 279 | 64 | -44 | 3.3 | D'Amico&al.(2010) |
| 200 | 20080902 | 21:57:20 | 15.69 | 38.25 | 34.0 | 351 | 72 | -65 | 3.1 | Orecchio&al.(2014) |
| 201 | 20080910 | 11:30:48 | 17.13 | 39.18 | 34.0 | 81 | 69 | -90 | 4.6 | Orecchio&al.(2014) |
| 202 | 20080912 | 20:12:11 | 17.37 | 39.17 | 34.0 | 332 | 85 | -2 | 3.6 | Orecchio&al.(2014) |
| 203 | 20080927 | 08:28:27 | 17.21 | 39.18 | 30.0 | 123 | 71 | -8 | 4.0 | Totaro & al. (2016) |
| 204 | 20081024 | 16:55:37 | 16.44 | 38.61 | 30.0 | 0 | 60 | 70 | 3.3 | Orecchio&al.(2014) |
| 205 | 20081024 | 18:47:54 | 16.47 | 38.59 | 28.0 | 323 | 38 | -24 | 3.4 | Orecchio&al.(2014) |
| 206 | 20081027 | 10:55:55 | 15.13 | 38.11 | 2.0 | 50 | 28 | -71 | 3.5 | D'Amico&al.(2010) |
| 207 | 20081102 | 06:46:44 | 16.49 | 37.64 | 40.0 | 141 | 67 | -79 | 3.6 | Orecchio&al.(2014) |
| 208 | 20081107 | 15:00:59 | 16.46 | 39.15 | 14.0 | 139 | 72 | -65 | 3.3 | Orecchio&al.(2014) |
| 209 | 20081120 | 14:09:21 | 17.49 | 39.14 | 15.0 | 166 | 82 | -2 | 4.5 | Italian CMT |
| 210 | 20081128 | 08:04:47 | 17.02 | 39.89 | 34.0 | 97 | 50 | 21 | 3.6 | Orecchio&al.(2014) |
| 211 | 20081128 | 23:39:21 | 13.69 | 37.54 | 35.0 | 337 | 74 | 9 | 4.4 | Italian CMT |
| 212 | 20081209 | 12:55:27 | 17.20 | 39.04 | 24.0 | 104 | 21 | -31 | 3.6 | Orecchio&al.(2014) |



| 213 | 20081225 | 18:55:58 | 15.96 | 40.34 | 6.0 | 100 | 59 | -23 | 2.6 | Orecchio&al.(2014) |
| 214 | 20090205 | 14:50:14 | 16.03 | 37.39 | 28.0 | 167 | 78 | 18 | 3.3 | Orecchio&al.(2014) |
| 215 | 20090316 | 00:28:06 | 15.96 | 37.67 | 28.0 | 34 | 60 | -24 | 3.0 | Orecchio&al.(2014) |
| 216 | 20090319 | 08:27:54 | 12.72 | 36.52 | 28.0 | 255 | 48 | -180 | 4.4 | Italian CMT |
| 217 | 20090325 | 12:23:25 | 19.63 | 40.34 | 10.0 | 3 | 35 | 90 | 4.3 | RCMT |
| 218 | 20090407 | 20:24:54 | 16.81 | 39.19 | 14.0 | 161 | 70 | -33 | 3.2 | Orecchio&al.(2014) |
| 219 | 20090413 | 11:39:58 | 16.39 | 39.53 | 8.0 | 260 | 40 | -7 | 3.3 | Totaro&al.(2013) |
| 220 | 20090427 | 09:42:16 | 15.08 | 38.07 | 30.0 | 69 | 78 | -19 | 3.6 | Presti&al.(2013) |
| 221 | 20090701 | 17:58:54 | 15.01 | 38.34 | 2.0 | 40 | 90 | 19 | 3.1 | Presti&al.(2013) |
| 222 | 20090727 | 22:15:14 | 15.69 | 37.12 | 30.0 | 353 | 48 | -13 | 3.2 | Orecchio&al.(2014) |
| 223 | 20090804 | 16:17:16 | 15.71 | 37.12 | 18.0 | 22 | 73 | -13 | 3.6 | Orecchio&al.(2014) |
| 224 | 20090829 | 06:55:17 | 15.47 | 37.92 | 8.0 | 56 | 80 | -47 | 2.9 | Orecchio&al.(2014) |
| 225 | 20090907 | 21:26:31 | 13.98 | 38.73 | 18.0 | 63 | 39 | 65 | 4.8 | Totaro&al.(2016) |
| 226 | 20090917 | 22:53:00 | 19.96 | 39.95 | 10.0 | 307 | 55 | 44 | 4.5 | RCMT |
| 227 | 20091012 | 20:07:49 | 15.96 | 37.23 | 30.0 | 204 | 82 | 12 | 3.4 | Orecchio&al.(2014) |
| 228 | 20091108 | 06:51:16 | 14.55 | 37.83 | 15.0 | 310 | 21 | -54 | 4.5 | RCMT |
| 229 | 20091125 | 06:20:07 | 16.45 | 38.05 | 16.0 | 341 | 62 | -43 | 3.2 | Orecchio&al.(2014) |
| 230 | 20091215 | 11:49:07 | 15.57 | 38.96 | 24.0 | 195 | 61 | -54 | 3.7 | Orecchio&al.(2014) |
| 231 | 20091219 | 09:01:19 | 15.09 | 37.76 | 40.0 | 112 | 44 | 176 | 4.4 | Italian CMT |
| 232 | 20100101 | 22:01:13 | 16.29 | 39.20 | 36.0 | 267 | 58 | -60 | 3.8 | Orecchio&al.(2014) |
| 233 | 20100208 | 07:23:58 | 16.77 | 39.50 | 34.0 | 94 | 73 | -31 | 3.6 | Presti&al.(2013) |
| 234 | 20100317 | 11:01:11 | 14.73 | 38.57 | 8.0 | 78 | 56 | 81 | 3.3 | Presti&al.(2013) |
| 235 | 20100325 | 17:30:18 | 15.86 | 40.03 | 2.0 | 0 | 51 | -67 | 3.2 | Orecchio&al.(2014) |
| 236 | 20100402 | 20:04:47 | 15.11 | 37.76 | 2.0 | 274 | 55 | 10 | 4.2 | Italian CMT |
| 237 | 20100404 | 15:40:28 | 16.82 | 39.35 | 22.0 | 314 | 76 | -18 | 3.3 | Presti&al.(2013) |
| 238 | 20100413 | 12:12:14 | 17.15 | 39.35 | 18.0 | 128 | 29 | -26 | 3.5 | Presti&al.(2013) |
| 239 | 20100415 | 20:05:47 | 17.22 | 39.35 | 18.0 | 137 | 39 | -15 | 3.6 | Presti&al.(2013) |
| 240 | 20100511 | 10:28:47 | 16.22 | 39.75 | 6.0 | 152 | 56 | -90 | 2.8 | Orecchio&al.(2014) |
| 241 | 20100511 | 18:09:43 | 17.47 | 39.31 | 22.0 | 126 | 40 | -28 | 3.8 | Orecchio&al.(2014) |
| 242 | 20100606 | 16:49:53 | 15.11 | 38.27 | 10.0 | 237 | 82 | -34 | 3.5 | Presti&al.(2013) |
| 243 | 20100616 | 22:39:41 | 16.14 | 38.83 | 15.0 | 109 | 50 | -38 | 4.1 | Italian CMT |
| 244 | 20100801 | 21:31:53 | 14.46 | 38.61 | 4.0 | 37 | 60 | 78 | 3.1 | Presti&al.(2013) |
| 245 | 20100816 | 12:54:46 | 14.92 | 38.42 | 10.0 | 218 | 66 | 42 | 4.5 | Presti&al.(2013) |
| 246 | 20100822 | 10:23:05 | 19.95 | 37.27 | 17.0 | 325 | 16 | 99 | 5.5 | CMT |
| 247 | 20100910 | 19:19:48 | 16.21 | 38.54 | 26.0 | 198 | 53 | -60 | 3.3 | Orecchio&al.(2014) |
| 248 | 20100910 | 21:39:20 | 15.82 | 38.20 | 28.0 | 204 | 69 | -70 | 3.2 | Orecchio&al.(2014) |
| 249 | 20101008 | 17:26:58 | 16.33 | 36.91 | 38.0 | 190 | 79 | 17 | 3.6 | Orecchio&al.(2014) |
| 250 | 20101014 | 14:18:28 | 16.69 | 38.84 | 28.0 | 60 | 60 | 31 | 3.2 | Orecchio&al.(2014) |
| 251 | 20101015 | 05:21:20 | 16.66 | 38.87 | 15.0 | 287 | 62 | 173 | 4.4 | Italian CMT |
| 252 | 20101109 | 08:43:20 | 15.93 | 40.05 | 10.0 | 329 | 61 | -57 | 3.5 | Orecchio&al.(2014) |
| 253 | 20101127 | 08:45:49 | 15.64 | 38.08 | 38.0 | 332 | 22 | -12 | 3.7 | Orecchio&al.(2014) |
| 254 | 20110325 | 16:18:12 | 16.94 | 38.87 | 6.0 | 87 | 70 | -53 | 3.3 | Orecchio&al.(2014) |
| 255 | 20110325 | 18:31:31 | 16.96 | 38.87 | 6.0 | 281 | 53 | -19 | 3.6 | Orecchio&al.(2014) |
| 256 | 20110424 | 13:02:12 | 14.88 | 35.82 | 20.3 | 28 | 34 | -76 | 4.2 | RCMT |
| 257 | 20110426 | 21:02:30 | 15.16 | 38.15 | 2.0 | 33 | 40 | -90 | 3.2 | Totaro&al.(2016) |
| 258 | 20110503 | 22:24:52 | 16.68 | 37.78 | 36.0 | 323 | 49 | -41 | 3.6 | Orecchio&al.(2014) |



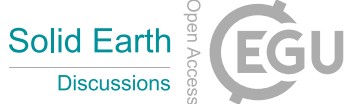

| 259 | 20110506 | 15:12:35 | 14.96 | 37.78 | 22.2 | 13 | 57 | 15 | 4.0 | Italian CMT |
|---|---|---|---|---|---|---|---|---|---|---|
| 260 | 20110609 | 16:16:36 | 19.75 | 40.69 | 10.0 | 171 | 78 | -170 | 4.4 | RCMT |
| 261 | 20110623 | 22:02:47 | 14.78 | 38.06 | 10.0 | 315 | 90 | -1 | 4.7 | Neri&al.(2014) |
| 262 | 20110624 | 09:00:08 | 16.61 | 39.61 | 22.0 | 330 | 21 | -20 | 3.1 | Totaro&al.(2013) |
| 263 | 20110627 | 05:23:41 | 14.74 | 38.02 | 8.0 | 313 | 63 | -2 | 3.2 | Totaro&al.(2016) |
| 264 | 20110627 | 22:13:45 | 14.74 | 38.04 | 10.0 | 299 | 44 | -31 | 3.4 | Totaro&al.(2016) |
| 265 | 20110629 | 09:04:17 | 14.73 | 38.05 | 8.0 | 305 | 90 | 2 | 3.2 | Totaro&al.(2016) |
| 266 | 20110629 | 19:15:15 | 14.74 | 38.06 | 10.0 | 139 | 81 | -10 | 3.5 | Totaro&al.(2016) |
| 267 | 20110706 | 09:08:39 | 14.78 | 38.05 | 10.0 | 124 | 42 | -41 | 3.7 | Neri&al.(2014) |
| 268 | 20110707 | 01:01:15 | 14.79 | 38.04 | 10.0 | 315 | 90 | 10 | 3.3 | Totaro&al.(2016) |
| 269 | 20110727 | 04:03:14 | 14.77 | 38.06 | 8.0 | 113 | 46 | -61 | 3.2 | Totaro&al.(2016) |
| 270 | 20110817 | 01:20:32 | 14.47 | 38.55 | 6.0 | 209 | 70 | 38 | 3.0 | Totaro&al.(2016) |
| 271 | 20110831 | 16:33:20 | 14.70 | 37.11 | 2.0 | 7 | 80 | 1 | 3.1 | Totaro&al.(2016) |
| 272 | 20111103 | 14:37:10 | 14.58 | 38.43 | 10.0 | 8 | 90 | 22 | 3.5 | Totaro&al.(2016) |
| 273 | 20111109 | 17:00:48 | 16.02 | 39.91 | 9.0 | 10 | 50 | -51 | 2.7 | Totaro&al.(2015) |
| 274 | 20111115 | 04:59:00 | 14.67 | 38.27 | 10.0 | 18 | 74 | 9 | 4.1 | Neri&al.(2014) |
| 275 | 20111119 | 10:19:16 | 14.35 | 37.81 | 14.0 | 121 | 70 | -25 | 3.4 | Totaro&al.(2016) |
| 276 | 20111123 | 14:12:34 | 16.01 | 39.92 | 9.0 | 7 | 40 | -48 | 3.5 | Totaro&al.(2015) |
| 277 | 20111201 | 14:01:20 | 16.01 | 39.93 | 9.8 | 7 | 48 | -52 | 3.3 | Totaro&al.(2015) |
| 278 | 20111202 | 21:25:38 | 16.01 | 39.92 | 9.8 | 156 | 49 | -90 | 3.2 | Totaro&al.(2015) |
| 279 | 20111214 | 17:59:49 | 16.20 | 39.38 | 6.0 | 178 | 39 | -43 | 3.1 | Orecchio&al.(2014) |
| 280 | 20111217 | 23:20:15 | 16.18 | 39.37 | 26.0 | 345 | 62 | -79 | 3.6 | Orecchio&al.(2014) |
| 281 | 20111218 | 15:01:06 | 12.68 | 36.10 | 12.0 | 183 | 85 | 2 | 4.7 | CMT |
| 282 | 20111224 | 20:17:50 | 16.02 | 39.92 | 9.1 | 348 | 44 | -90 | 3.2 | Totaro&al.(2015) |
| 283 | 20111227 | 01:07:45 | 16.92 | 39.58 | 20.0 | 121 | 28 | -26 | 3.6 | Totaro&al.(2016) |
| 284 | 20120103 | 23:47:41 | 16.84 | 39.64 | 16.0 | 156 | 29 | -29 | 3.1 | Orecchio&al.(2014) |
| 285 | 20120129 | 11:14:50 | 14.27 | 37.88 | 10.0 | 37 | 61 | -23 | 3.0 | Totaro&al.(2016) |
| 286 | 20120201 | 14:28:38 | 14.28 | 37.89 | 30.0 | 119 | 90 | 14 | 3.6 | Neri&al.(2014) |
| 287 | 20120208 | 16:15:56 | 14.30 | 37.89 | 12.0 | 200 | 50 | -51 | 3.1 | Totaro&al.(2016) |
| 288 | 20120225 | 20:34:35 | 13.55 | 38.54 | 12.0 | 20 | 27 | 19 | 4.3 | Neri&al.(2014) |
| 289 | 20120226 | 16:17:23 | 16.01 | 37.31 | 36.0 | 338 | 70 | -40 | 3.7 | Polonia&al.(2016) |
| 290 | 20120324 | 20:34:59 | 15.88 | 37.59 | 32.0 | 158 | 84 | -9 | 3.1 | Polonia&al.(2016) |
| 291 | 20120405 | 03:01:06 | 16.42 | 39.56 | 18.0 | 275 | 42 | -53 | 3.0 | Totaro&al.(2013) |
| 292 | 20120409 | 23:29:02 | 15.46 | 40.41 | 14.0 | 165 | 58 | -77 | 3.1 | Totaro&al.(2013) |
| 293 | 20120412 | 13:20:28 | 15.62 | 37.89 | 10.0 | 319 | 90 | 81 | 3.1 | Polonia&al.(2016) |
| 294 | 20120413 | 06:21:33 | 13.30 | 38.35 | 12.0 | 319 | 29 | -71 | 4.4 | Neri&al.(2014) |
| 295 | 20120528 | 01:06:27 | 16.12 | 39.89 | 7.0 | 146 | 49 | -90 | 4.3 | Totaro&al.(2015) |
| 296 | 20120528 | 01:32:10 | 16.10 | 39.90 | 9.3 | 236 | 51 | -27 | 3.1 | Totaro&al.(2015) |
| 297 | 20120531 | 03:16:22 | 15.57 | 39.89 | 8.0 | 211 | 45 | 2 | 3.0 | Orecchio&al.(2014) |
| 298 | 20120531 | 20:18:23 | 16.93 | 38.96 | 4.0 | 258 | 82 | -74 | 2.9 | Totaro&al.(2016) |
| 299 | 20120615 | 06:27:25 | 16.29 | 37.45 | 38.0 | 190 | 80 | 7 | 3.8 | Polonia&al.(2016) |
| 300 | 20120618 | 03:17:12 | 16.11 | 39.90 | 9.5 | 162 | 29 | -90 | 2.7 | Totaro&al.(2015) |
| 301 | 20120625 | 10:52:51 | 15.05 | 37.01 | 18.0 | 164 | 77 | -16 | 3.2 | Totaro&al.(2016) |
| 302 | 20120627 | 01:14:20 | 15.03 | 37.00 | 4.0 | 182 | 90 | 3 | 3.5 | Totaro&al.(2016) |
| 303 | 20120627 | 01:20:59 | 15.03 | 36.99 | 4.0 | 10 | 81 | -3 | 2.9 | Totaro&al.(2016) |
| 304 | 20120627 | 02:48:02 | 15.03 | 37.00 | 4.0 | 172 | 90 | -6 | 2.9 | Totaro&al.(2016) |





| 305 | 20120704 | 11:12:12 | 16.87 | 37.69 | 40.0 | 186 | 74 | 3 | 4.6 | RCMT |
|---|---|---|---|---|---|---|---|---|---|---|
| 306 | 20120715 | 11:51:55 | 16.91 | 39.65 | 18.0 | 118 | 27 | -17 | 3.0 | Orecchio&al.(2014) |
| 307 | 20120726 | 14:20:03 | 16.34 | 37.90 | 16.0 | 134 | 83 | -19 | 3.1 | Polonia&al.(2016) |
| 308 | 20120813 | 07:30:51 | 13.73 | 38.52 | 26.5 | 19 | 24 | 63 | 4.2 | RCMT |
| 309 | 20120819 | 17:45:08 | 16.02 | 39.89 | 8.8 | 154 | 46 | -90 | 3.5 | Totaro&al.(2015) |
| 310 | 20120819 | 21:28:29 | 16.03 | 39.89 | 8.3 | 341 | 37 | -62 | 2.7 | Totaro&al.(2015) |
| 311 | 20120828 | 23:12:15 | 15.71 | 38.25 | 45.4 | 130 | 10 | -18 | 4.6 | present work |
| 312 | 20120901 | 14:02:45 | 16.03 | 39.89 | 8.4 | 178 | 60 | -69 | 3.4 | Totaro&al.(2015) |
| 313 | 20120904 | 03:48:03 | 16.03 | 39.90 | 8.3 | 161 | 55 | -81 | 2.8 | Totaro&al.(2015) |
| 314 | 20120907 | 12:40:51 | 16.03 | 39.89 | 7.9 | 177 | 52 | -70 | 3.3 | Totaro&al.(2015) |
| 315 | 20120907 | 15:10:07 | 16.02 | 39.89 | 6.0 | 176 | 61 | -62 | 2.8 | Totaro&al.(2016) |
| 316 | 20120914 | 03:50:11 | 16.03 | 39.90 | 7.9 | 156 | 57 | -90 | 3.6 | Totaro&al.(2015) |
| 317 | 20120922 | 01:45:02 | 16.02 | 39.91 | 8.0 | 139 | 58 | -84 | 2.7 | Totaro&al.(2015) |
| 318 | 20120922 | 05:10:35 | 16.61 | 39.78 | 14.0 | 128 | 90 | 71 | 3.5 | Totaro&al.(2016) |
| 319 | 20120923 | 06:13:56 | 16.02 | 39.91 | 8.4 | 331 | 32 | -90 | 2.7 | Totaro&al.(2015) |
| 320 | 20120924 | 20:48:36 | 16.03 | 39.92 | 6.4 | 231 | 59 | -42 | 2.7 | Totaro&al.(2015) |
| 321 | 20120928 | 05:56:46 | 16.10 | 39.91 | 7.3 | 22 | 41 | -80 | 2.8 | Totaro&al.(2015) |
| 322 | 20121001 | 20:28:28 | 16.03 | 39.91 | 7.7 | 343 | 39 | -82 | 3.5 | Totaro&al.(2015) |
| 323 | 20121001 | 21:27:51 | 16.02 | 39.91 | 7.8 | 13 | 40 | -43 | 3.3 | Totaro&al.(2015) |
| 324 | 20121002 | 00:08:57 | 16.03 | 39.91 | 8.6 | 331 | 40 | -80 | 3.3 | Totaro&al.(2015) |
| 325 | 20121002 | 04:35:18 | 16.03 | 39.91 | 8.2 | 140 | 58 | -78 | 2.8 | Totaro&al.(2015) |
| 326 | 20121004 | 09:32:33 | 16.02 | 39.90 | 8.0 | 159 | 52 | -84 | 2.9 | Totaro&al.(2015) |
| 327 | 20121005 | 11:12:28 | 16.03 | 39.90 | 7.6 | 0 | 40 | -73 | 3.0 | Totaro&al.(2015) |
| 328 | 20121014 | 14:49:24 | 16.02 | 39.91 | 8.7 | 20 | 42 | -40 | 2.7 | Totaro&al.(2015) |
| 329 | 20121018 | 02:51:57 | 16.03 | 39.90 | 7.8 | 350 | 34 | -90 | 3.3 | Totaro&al.(2015) |
| 330 | 20121023 | 10:40:24 | 16.03 | 39.91 | 8.4 | 324 | 30 | -82 | 3.1 | Totaro&al.(2015) |
| 331 | 20121025 | 23:05:25 | 16.03 | 39.89 | 8.8 | 166 | 50 | -77 | 5.0 | Totaro&al.(2015) |
| 332 | 20121026 | 00:31:53 | 16.00 | 39.89 | 10.0 | 349 | 29 | -49 | 3.0 | Totaro&al.(2016) |
| 333 | 20121026 | 02:25:09 | 16.03 | 39.92 | 6.6 | 352 | 40 | -81 | 2.9 | Totaro&al.(2015) |
| 334 | 20121026 | 02:40:08 | 16.02 | 39.88 | 8.1 | 73 | 50 | -50 | 2.8 | Totaro&al.(2015) |
| 335 | 20121026 | 16:08:58 | 16.03 | 39.89 | 8.9 | 11 | 56 | -23 | 2.7 | Totaro&al.(2015) |
| 336 | 20121028 | 13:52:18 | 16.02 | 39.93 | 8.5 | 12 | 75 | 27 | 3.1 | Totaro&al.(2015) |
| 337 | 20121102 | 01:59:34 | 16.47 | 38.78 | 14.0 | 69 | 22 | -50 | 3.1 | Orecchio&al.(2014) |
| 338 | 20121102 | 17:50:44 | 16.03 | 39.91 | 7.9 | 244 | 58 | -66 | 3.0 | Totaro&al.(2015) |
| 339 | 20121102 | 17:58:47 | 16.03 | 39.92 | 7.8 | 39 | 52 | -56 | 2.7 | Totaro&al.(2015) |
| 340 | 20121105 | 12:06:32 | 16.01 | 39.94 | 8.8 | 21 | 71 | -11 | 3.4 | Totaro&al.(2015) |
| 341 | 20121108 | 11:11:57 | 16.10 | 39.91 | 8.4 | 158 | 16 | -79 | 3.1 | Totaro&al.(2015) |
| 342 | 20121112 | 03:03:53 | 16.01 | 39.92 | 8.7 | 229 | 42 | 20 | 3.0 | Totaro&al.(2015) |
| 343 | 20121121 | 06:43:25 | 16.02 | 39.92 | 8.2 | 63 | 60 | -39 | 2.9 | Totaro&al.(2015) |
| 344 | 20121122 | 01:59:52 | 16.02 | 39.92 | 9.0 | 0 | 41 | -78 | 3.2 | Totaro&al.(2015) |
| 345 | 20121122 | 09:10:41 | 14.96 | 37.80 | 10.0 | 258 | 65 | 154 | 4.1 | RCMT |
| 346 | 20121122 | 11:25:52 | 14.99 | 37.77 | 20.0 | 6 | 57 | 23 | 4.2 | RCMT |
| 347 | 20121124 | 22:24:26 | 16.02 | 39.92 | 7.9 | 0 | 47 | -79 | 2.8 | Totaro&al.(2015) |
| 348 | 20121125 | 08:28:39 | 16.02 | 39.92 | 9.4 | 360 | 42 | -72 | 3.5 | Totaro&al.(2015) |
| 349 | 20121125 | 08:42:25 | 16.03 | 39.93 | 5.5 | 0 | 41 | -74 | 2.9 | Totaro&al.(2015) |
| 350 | 20121125 | 08:53:33 | 16.01 | 39.89 | 7.6 | 168 | 45 | -90 | 3.0 | Totaro&al.(2015) |




| 351 | 20121125 | 17:48:02 | 16.01 | 39.92 | 9.5 | 7 | 43 | -69 | 3.1 | Totaro&al.(2015) |
| 352 | 20121128 | 02:43:46 | 16.01 | 39.92 | 9.1 | 45 | 86 | 2 | 2.9 | Totaro&al.(2015) |
| 353 | 20121211 | 14:28:43 | 16.01 | 39.88 | 9.4 | 339 | 29 | -70 | 3.3 | Totaro&al.(2015) |
| 354 | 20121213 | 04:44:03 | 16.03 | 39.88 | 8.8 | 20 | 70 | -31 | 3.2 | Totaro&al.(2015) |
| 355 | 20121218 | 11:03:18 | 16.17 | 39.84 | 2.0 | 130 | 49 | -90 | 3.3 | Totaro&al.(2016) |
| 356 | 20121218 | 11:05:43 | 16.17 | 39.84 | 4.0 | 158 | 62 | -30 | 3.0 | Totaro&al.(2016) |
| 357 | 20121226 | 08:22:48 | 16.31 | 39.50 | 8.0 | 41 | 29 | -41 | 3.2 | Totaro&al.(2016) |
| 358 | 20130104 | 07:50:06 | 14.72 | 37.88 | 12.0 | 318 | 43 | 41 | 4.4 | Neri&al.(2014) |
| 359 | 20130104 | 10:50:21 | 14.70 | 37.88 | 31.0 | 301 | 81 | -14 | 3.2 | Totaro&al.(2016) |
| 360 | 20130106 | 07:50:19 | 14.72 | 37.87 | 18.0 | 118 | 63 | -22 | 3.2 | Totaro&al.(2016) |
| 361 | 20130109 | 16:10:34 | 14.72 | 37.88 | 6.0 | 312 | 78 | -11 | 3.0 | Totaro&al.(2016) |
| 362 | 20130205 | 22:08:04 | 15.86 | 40.07 | 18.0 | 176 | 71 | -66 | 3.1 | Totaro&al.(2016) |
| 363 | 20130223 | 19:14:18 | 14.98 | 38.31 | 4.0 | 220 | 79 | -7 | 3.4 | Totaro&al.(2016) |
| 364 | 20130303 | 23:39:13 | 15.83 | 38.13 | 8.0 | 237 | 57 | -82 | 3.3 | Totaro&al.(2016) |
| 365 | 20130307 | 22:36:59 | 14.52 | 37.97 | 2.0 | 0 | 50 | -31 | 3.6 | Neri&al.(2014) |
| 366 | 20130317 | 14:22:15 | 15.89 | 39.62 | 32.0 | 31 | 79 | 48 | 3.3 | Totaro&al.(2016) |
| 367 | 20130319 | 07:50:06 | 14.51 | 37.98 | 4.0 | 164 | 78 | -43 | 3.4 | Totaro&al.(2016) |
| 368 | 20130319 | 08:37:04 | 14.51 | 37.98 | 4.0 | 159 | 69 | -39 | 3.3 | Totaro&al.(2016) |
| 369 | 20130319 | 08:38:45 | 13.56 | 37.84 | 2.0 | 183 | 69 | -8 | 2.8 | Totaro&al.(2016) |
| 370 | 20130324 | 15:47:22 | 16.50 | 37.76 | 30.0 | 257 | 87 | 178 | 4.6 | RCMT |
| 371 | 20130401 | 03:07:13 | 16.67 | 39.68 | 28.0 | 127 | 68 | 90 | 3.4 | Totaro&al.(2016) |
| 372 | 20130402 | 01:10:52 | 15.59 | 37.79 | 12.0 | 219 | 85 | -10 | 2.9 | Totaro&al.(2016) |
| 373 | 20130406 | 04:14:11 | 15.21 | 40.41 | 2.0 | 249 | 47 | 22 | 2.9 | Totaro&al.(2016) |
| 374 | 20130412 | 17:50:00 | 14.92 | 38.16 | 16.0 | 0 | 67 | 15 | 3.1 | Totaro&al.(2016) |
| 375 | 20130509 | 20:41:22 | 16.05 | 39.18 | 31.0 | 344 | 82 | -110 | 3.8 | TDMT |
| 376 | 20130622 | 08:41:00 | 19.58 | 40.21 | 10.0 | 324 | 21 | 55 | 4.4 | RCMT |
| 377 | 20130704 | 13:56:06 | 15.61 | 40.49 | 4.0 | 211 | 41 | -42 | 2.9 | Totaro&al.(2016) |
| 378 | 20130717 | 04:26:36 | 15.83 | 40.02 | 10.0 | 149 | 90 | -72 | 2.9 | Totaro&al.(2016) |
| 379 | 20130804 | 02:47:47 | 12.85 | 38.69 | 4.0 | 12 | 72 | -11 | 3.3 | Totaro&al.(2016) |
| 380 | 20130815 | 23:04:58 | 14.91 | 38.14 | 12.0 | 78 | 82 | 57 | 4.5 | Neri&al.(2014) |
| 381 | 20130815 | 23:06:51 | 14.92 | 38.15 | 12.0 | 77 | 82 | 55 | 4.6 | Neri&al.(2014) |
| 382 | 20130819 | 05:48:23 | 14.26 | 37.70 | 20.0 | 23 | 59 | -14 | 3.0 | Totaro&al.(2016) |
| 383 | 20130828 | 09:07:00 | 14.31 | 38.85 | 6.0 | 210 | 72 | 12 | 3.6 | Totaro&al.(2016) |
| 384 | 20130917 | 22:56:38 | 15.80 | 40.79 | 8.0 | 339 | 69 | 1 | 3.3 | Totaro&al.(2016) |
| 385 | 20130917 | 23:38:49 | 15.81 | 40.79 | 10.0 | 133 | 72 | -83 | 3.4 | Totaro&al.(2016) |
| 386 | 20130921 | 13:18:02 | 15.78 | 40.79 | 16.0 | 342 | 57 | -34 | 3.2 | Totaro&al.(2016) |
| 387 | 20130926 | 12:19:59 | 17.23 | 39.13 | 22.0 | 188 | 12 | 52 | 3.1 | Totaro&al.(2016) |
| 388 | 20131003 | 18:22:25 | 13.49 | 38.47 | 14.0 | 20 | 51 | 0 | 3.4 | Totaro&al.(2016) |
| 389 | 20131007 | 04:44:05 | 15.08 | 38.13 | 6.0 | 84 | 63 | -29 | 2.8 | Totaro&al.(2016) |
| 390 | 20131008 | 10:33:20 | 15.82 | 40.02 | 2.0 | 55 | 60 | -10 | 2.7 | Totaro&al.(2016) |
| 391 | 20131009 | 08:14:49 | 15.09 | 37.61 | 4.0 | 33 | 81 | 8 | 2.9 | Totaro&al.(2016) |
| 392 | 20131009 | 08:33:22 | 15.07 | 37.61 | 10.0 | 258 | 90 | 1 | 3.2 | Totaro&al.(2016) |
| 393 | 20131018 | 08:03:44 | 14.90 | 38.11 | 14.0 | 258 | 79 | -61 | 3.0 | Totaro&al.(2016) |
| 394 | 20131018 | 11:05:21 | 14.97 | 36.79 | 2.0 | 173 | 82 | 3 | 2.9 | Totaro&al.(2016) |
| 395 | 20131018 | 15:08:31 | 10.83 | 35.70 | 12.0 | 99 | 51 | -179 | 4.8 | CMT |
| 396 | 20131018 | 20:50:52 | 15.22 | 40.80 | 14.0 | 319 | 41 | -76 | 3.1 | Totaro&al.(2016) |

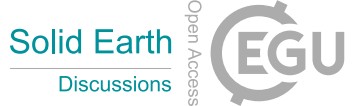

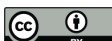

| 397 | 20131019 | 11:08:03 | 15.18 | 40.59 | 14.0 | 112 | 30 | 58 | 3.0 | Totaro&al.(2016) |
|-----|----------|----------|-------|-------|------|-----|----|------|-----|------------------|
| 398 | 20131021 | 19:37:00 | 10.93 | 35.60 | 10.0 | 89 | 71 | 165 | 4.5 | RCMT |
| 399 | 20131103 | 14:33:41 | 15.18 | 40.60 | 2.0 | 87 | 58 | -80 | 2.7 | Totaro&al.(2016) |
| 400 | 20131105 | 05:06:39 | 14.91 | 37.69 | 22.0 | 84 | 57 | -11 | 3.1 | Totaro&al.(2016) |
| 401 | 20131105 | 05:29:58 | 14.92 | 37.70 | 18.0 | 99 | 72 | -14 | 3.0 | Totaro&al.(2016) |
| 402 | 20131105 | 17:25:23 | 16.01 | 39.88 | 8.0 | 80 | 51 | -45 | 3.0 | Totaro&al.(2016) |
| 403 | 20131105 | 17:26:45 | 16.01 | 39.88 | 10.0 | 328 | 32 | -79 | 3.4 | Totaro&al.(2016) |
| 404 | 20131210 | 20:39:39 | 15.45 | 39.72 | 22.0 | 172 | 19 | 69 | 3.1 | Totaro&al.(2016) |
| 405 | 20131214 | 21:49:05 | 14.74 | 37.76 | 30.0 | 53 | 67 | -27 | 3.4 | Totaro&al.(2016) |
| 406 | 20131215 | 03:57:33 | 14.94 | 36.67 | 15.0 | 83 | 47 | 143 | 4.1 | RCMT |
| 407 | 20131223 | 04:20:39 | 15.57 | 38.22 | 2.0 | 31 | 61 | -60 | 3.5 | Neri&al.(2014) |
| 408 | 20131223 | 16:17:11 | 15.04 | 38.18 | 16.0 | 113 | 63 | -33 | 3.1 | Totaro&al.(2016) |
| 409 | 20140102 | 06:13:18 | 15.04 | 38.18 | 12.0 | 103 | 38 | 8 | 3.0 | Totaro&al.(2016) |
| 410 | 20140114 | 03:43:42 | 14.92 | 38.37 | 12.0 | 315 | 78 | 171 | 4.1 | RCMT |
| 411 | 20140114 | 04:35:00 | 14.94 | 38.36 | 11.0 | 308 | 47 | 159 | 4.2 | RCMT |
| 412 | 20140122 | 19:35:01 | 15.12 | 40.44 | 8.0 | 61 | 32 | -61 | 3.6 | Totaro&al.(2016) |
| 413 | 20140127 | 21:39:32 | 15.93 | 40.12 | 10.0 | 38 | 58 | -19 | 3.0 | Totaro&al.(2016) |
| 414 | 20140218 | 21:44:19 | 13.82 | 37.63 | 4.0 | 250 | 79 | 33 | 3.3 | Totaro&al.(2016) |
| 415 | 20140219 | 06:58:05 | 15.11 | 38.17 | 8.0 | 34 | 62 | 17 | 3.0 | Totaro&al.(2016) |
| 416 | 20140301 | 01:48:50 | 15.22 | 40.24 | 4.0 | 19 | 27 | 40 | 2.9 | Totaro&al.(2016) |
| 417 | 20140301 | 11:51:25 | 15.29 | 40.25 | 6.0 | 305 | 68 | 8 | 2.7 | Totaro&al.(2016) |
| 418 | 20140303 | 06:26:04 | 15.01 | 40.40 | 8.0 | 287 | 47 | -64 | 3.0 | Totaro&al.(2016) |
| 419 | 20140308 | 14:19:05 | 15.81 | 39.77 | 16.0 | 280 | 79 | 46 | 3.0 | Totaro&al.(2016) |
| 420 | 20140308 | 20:52:51 | 14.89 | 37.96 | 32.0 | 13 | 47 | -40 | 3.8 | Neri&al.(2014) |
| 421 | 20140314 | 03:32:24 | 14.81 | 38.22 | 2.0 | 19 | 90 | -1 | 3.0 | Totaro&al.(2016) |
| 422 | 20140314 | 03:37:16 | 14.80 | 38.20 | 6.0 | 331 | 62 | 4 | 2.8 | Totaro&al.(2016) |
| 423 | 20140323 | 18:31:52 | 16.48 | 37.47 | 38.0 | 177 | 61 | 21 | 3.6 | Totaro&al.(2016) |
| 424 | 20140325 | 07:56:52 | 16.54 | 39.29 | 28.0 | 276 | 59 | -50 | 3.6 | Totaro&al.(2016) |
| 425 | 20140405 | 10:24:46 | 17.18 | 38.82 | 64.0 | 92 | 59 | -168 | 4.8 | RCMT |
| 426 | 20140412 | 06:53:05 | 13.88 | 37.82 | 2.0 | 360 | 81 | -1 | 3.1 | Totaro&al.(2016) |
| 427 | 20140417 | 21:52:26 | 15.21 | 38.23 | 2.0 | 12 | 90 | 2 | 2.8 | Totaro&al.(2016) |
| 428 | 20140503 | 20:04:06 | 16.11 | 39.90 | 4.0 | 232 | 31 | -70 | 3.0 | Totaro&al.(2016) |
| 429 | 20140506 | 08:24:42 | 16.07 | 39.91 | 4.0 | 241 | 51 | -90 | 3.0 | Totaro&al.(2016) |
| 430 | 20140507 | 06:20:00 | 19.78 | 37.75 | 10.0 | 181 | 68 | 16 | 4.3 | RCMT |
| 431 | 20140507 | 17:08:35 | 16.02 | 39.89 | 8.0 | 160 | 49 | -58 | 2.9 | Totaro&al.(2016) |
| 432 | 20140517 | 22:38:44 | 15.61 | 37.41 | 30.0 | 191 | 78 | -38 | 3.1 | Totaro&al.(2016) |
| 433 | 20140519 | 00:59:23 | 19.94 | 40.91 | 20.0 | 59 | 49 | 11 | 5.1 | CMT |
| 434 | 20140520 | 04:43:26 | 19.86 | 40.92 | 2.0 | 63 | 74 | 8 | 4.4 | RCMT |
| 435 | 20140604 | 21:20:41 | 16.00 | 39.88 | 8.0 | 159 | 58 | -90 | 3.6 | Totaro&al.(2016) |
| 436 | 20140606 | 13:41:38 | 16.09 | 39.90 | 4.0 | 155 | 31 | -90 | 3.9 | Totaro&al.(2016) |
| 437 | 20140607 | 15:00:49 | 15.10 | 38.09 | 10.0 | 285 | 73 | 7 | 3.7 | Neri&al.(2014) |
| 438 | 20140607 | 15:13:20 | 15.10 | 38.08 | 10.0 | 238 | 39 | -49 | 2.6 | Totaro&al.(2016) |
| 439 | 20140617 | 12:25:02 | 16.98 | 38.96 | 2.0 | 239 | 40 | -90 | 2.7 | Totaro&al.(2016) |
| 440 | 20140627 | 02:56:49 | 14.64 | 37.83 | 14.0 | 360 | 37 | -26 | 3.3 | Totaro&al.(2016) |
| 441 | 20140629 | 04:24:28 | 15.60 | 39.92 | 14.0 | 248 | 22 | 1 | 3.8 | Totaro&al.(2016) |
| 442 | 20140629 | 05:32:22 | 15.60 | 39.91 | 6.0 | 47 | 65 | -47 | 2.6 | Totaro&al.(2016) |





| 443 | 20140629 | 05:42:56 | 15.59 | 39.91 | 14.0 | 240 | 21 | -21 | 3.3 | Totaro&al.(2016) |
|-----|----------|----------|-------|-------|------|-----|----|------|-----|------------------|
| 444 | 20140702 | 18:49:46 | 15.17 | 40.59 | 2.0 | 71 | 49 | -70 | 3.1 | Totaro&al.(2016) |
| 445 | 20140707 | 17:44:15 | 15.50 | 40.72 | 12.0 | 32 | 64 | 30 | 3.3 | Totaro&al.(2016) |
| 446 | 20140708 | 05:02:43 | 16.12 | 39.90 | 2.0 | 347 | 51 | -83 | 2.7 | Totaro&al.(2016) |
| 447 | 20140711 | 21:57:58 | 15.39 | 40.49 | 10.0 | 344 | 82 | -49 | 3.5 | Totaro&al.(2016) |
| 448 | 20140724 | 01:12:51 | 15.02 | 38.09 | 30.0 | 100 | 90 | -16 | 3.2 | Totaro&al.(2016) |
| 449 | 20140731 | 03:29:29 | 16.11 | 39.91 | 4.0 | 349 | 58 | -82 | 3.4 | Totaro&al.(2016) |
| 450 | 20140806 | 08:16:21 | 15.83 | 40.58 | 8.0 | 323 | 83 | -9 | 3.0 | Totaro&al.(2016) |
| 451 | 20140812 | 20:15:34 | 16.42 | 40.45 | 16.0 | 330 | 44 | -54 | 3.9 | RCMT |
| 452 | 20140813 | 10:08:09 | 16.43 | 40.44 | 4.0 | 160 | 63 | -25 | 3.0 | Totaro&al.(2016) |
| 453 | 20140816 | 05:00:12 | 13.42 | 38.53 | 10.0 | 161 | 86 | 59 | 3.4 | TDMT |
| 454 | 20140826 | 01:19:46 | 14.33 | 37.95 | 14.0 | 298 | 20 | -90 | 3.5 | Totaro&al.(2016) |
| 455 | 20140905 | 10:10:19 | 18.76 | 39.19 | 10.0 | 334 | 39 | -56 | 4.1 | RCMT |
| 456 | 20140907 | 09:56:25 | 19.88 | 37.61 | 10.0 | 329 | 33 | 104 | 4.6 | RCMT |
| 457 | 20140919 | 05:32:38 | 14.81 | 38.49 | 14.0 | 66 | 78 | 25 | 3.2 | Totaro&al.(2016) |
| 458 | 20140924 | 15:39:09 | 15.85 | 39.74 | 24.0 | 51 | 80 | -39 | 3.2 | Totaro&al.(2016) |
| 459 | 20140926 | 23:38:11 | 16.50 | 36.78 | 40.0 | 267 | 75 | 170 | 4.2 | RCMT |
| 460 | 20141009 | 22:58:28 | 14.85 | 38.51 | 10.0 | 76 | 40 | 80 | 4.1 | RCMT |
| 461 | 20141010 | 16:16:18 | 15.14 | 38.09 | 34.0 | 60 | 90 | -76 | 3.1 | Totaro&al.(2016) |
| 462 | 20141010 | 16:27:13 | 15.14 | 38.09 | 30.0 | 349 | 72 | -31 | 2.8 | Totaro&al.(2016) |
| 463 | 20141013 | 03:34:47 | 16.46 | 39.37 | 10.0 | 18 | 77 | -73 | 2.8 | Totaro&al.(2016) |
| 464 | 20141014 | 00:50:55 | 16.63 | 38.95 | 14.0 | 150 | 74 | -1 | 2.9 | Totaro&al.(2016) |
| 465 | 20141025 | 20:09:48 | 15.95 | 38.17 | 15.0 | 275 | 68 | -79 | 3.3 | TDMT |
| 466 | 20141116 | 12:38:42 | 15.08 | 38.24 | 6.0 | 276 | 84 | 1 | 2.8 | Totaro&al.(2016) |
| 467 | 20141127 | 09:09:44 | 16.49 | 38.86 | 20.0 | 28 | 39 | -81 | 3.2 | Totaro&al.(2016) |
| 468 | 20141228 | 21:43:38 | 16.36 | 39.29 | 10.0 | 179 | 49 | -69 | 4.2 | Totaro&al.(2016) |
| 469 | 20150110 | 03:40:38 | 13.79 | 38.07 | 4.0 | 327 | 81 | 1 | 2.9 | Totaro&al.(2016) |
| 470 | 20150120 | 07:17:22 | 15.57 | 38.11 | 18.0 | 199 | 38 | 32 | 3.3 | Totaro&al.(2016) |
| 471 | 20150208 | 19:39:22 | 15.22 | 37.35 | 22.0 | 87 | 52 | -50 | 3.1 | Totaro&al.(2016) |
| 472 | 20150211 | 01:42:09 | 14.75 | 38.05 | 4.0 | 127 | 37 | -57 | 3.0 | Totaro&al.(2016) |
| 473 | 20150211 | 03:57:01 | 14.74 | 38.05 | 4.0 | 272 | 70 | -37 | 3.1 | Totaro&al.(2016) |
| 474 | 20150312 | 15:29:04 | 16.23 | 38.43 | 2.0 | 230 | 59 | -78 | 3.3 | Totaro&al.(2016) |
| 475 | 20150328 | 22:07:51 | 16.21 | 38.09 | 20.0 | 233 | 79 | 79 | 3.1 | Totaro&al.(2016) |
| 476 | 20150329 | 10:48:46 | 16.21 | 38.09 | 12.0 | 52 | 76 | -83 | 3.5 | Totaro&al.(2016) |
| 477 | 20150420 | 01:07:43 | 15.12 | 37.80 | 3.0 | 178 | 83 | -164 | 3.5 | TDMT |
| 478 | 20150430 | 05:35:21 | 15.39 | 37.86 | 26.0 | 160 | 61 | 44 | 2.9 | Totaro&al.(2016) |
| 479 | 20150511 | 08:26:32 | 16.80 | 37.33 | 40.0 | 184 | 62 | 20 | 4.5 | RCMT |
| 480 | 20150511 | 08:26:30 | 16.79 | 37.18 | 44.0 | 187 | 69 | -9 | 4.2 | present work |
| 481 | 20150521 | 15:31:18 | 19.85 | 37.68 | 15.0 | 207 | 72 | 177 | 4.4 | RCMT |
| 482 | 20150521 | 22:13:22 | 14.72 | 38.45 | 2.0 | 281 | 53 | -10 | 2.9 | Totaro&al.(2016) |
| 483 | 20150522 | 06:31:16 | 19.84 | 37.67 | 10.0 | 206 | 80 | 178 | 4.2 | RCMT |
| 484 | 20150524 | 06:00:00 | 16.03 | 37.96 | 62.0 | 23 | 62 | -3 | 4.1 | present work |
| 485 | 20150611 | 04:31:45 | 14.01 | 37.88 | 8.0 | 148 | 60 | -54 | 2.7 | Totaro&al.(2016) |
| 486 | 20150617 | 09:44:07 | 14.15 | 37.66 | 24.0 | 332 | 82 | 8 | 3.4 | Totaro&al.(2016) |
| 487 | 20150703 | 01:07:24 | 16.52 | 39.92 | 16.0 | 188 | 60 | 62 | 3.2 | Totaro&al.(2016) |
| 488 | 20150715 | 04:19:11 | 14.43 | 37.22 | 18.0 | 164 | 79 | 11 | 3.1 | Totaro&al.(2016) |



| 489 | 20150715 | 16:29:49 | 15.05 | 37.62 | 10.0 | 214 | 90 | -11 | 3.0 | Totaro&al.(2016) |
| 490 | 20150726 | 13:39:38 | 14.76 | 38.50 | 6.0 | 204 | 84 | 48 | 2.9 | Totaro&al.(2016) |
| 491 | 20150801 | 02:46:51 | 15.86 | 37.64 | 42.0 | 195 | 90 | 9 | 3.4 | Totaro&al.(2016) |
| 492 | 20150803 | 07:27:49 | 16.49 | 39.15 | 16.0 | 242 | 19 | -33 | 3.9 | Totaro&al.(2016) |
| 493 | 20150803 | 13:52:37 | 16.14 | 37.39 | 38.0 | 205 | 59 | 1 | 3.6 | Totaro&al.(2016) |
| 494 | 20150804 | 23:36:31 | 14.15 | 37.64 | 10.0 | 168 | 84 | 14 | 3.2 | Totaro&al.(2016) |
| 495 | 20150806 | 01:59:43 | 15.19 | 38.24 | 8.0 | 214 | 76 | -61 | 3.1 | Totaro&al.(2016) |
| 496 | 20150808 | 22:46:24 | 14.27 | 38.55 | 8.0 | 173 | 90 | 14 | 3.8 | Totaro&al.(2016) |
| 497 | 20150818 | 05:59:15 | 15.45 | 40.64 | 12.0 | 326 | 70 | -45 | 3.0 | Totaro&al.(2016) |
| 498 | 20150826 | 04:28:36 | 16.91 | 38.79 | 24.0 | 10 | 90 | -33 | 3.3 | Totaro&al.(2016) |
| 499 | 20150829 | 20:25:13 | 12.12 | 38.54 | 14.0 | 332 | 81 | 8 | 4.0 | Totaro&al.(2016) |
| 500 | 20150920 | 22:27:58 | 15.61 | 37.16 | 30.0 | 226 | 58 | -2 | 3.8 | Totaro&al.(2016) |
| 501 | 20151220 | 09:46:00 | 13.58 | 38.35 | 10.0 | 24 | 20 | 59 | 4.4 | RCMT |
| 502 | 20151229 | 14:15:53 | 19.89 | 37.42 | 10.0 | 107 | 58 | -14 | 4.1 | RCMT |
| 503 | 20160102 | 12:36:24 | 12.03 | 36.46 | 10.0 | 268 | 79 | -178 | 4.2 | RCMT |
| 504 | 20160113 | 17:01:30 | 14.67 | 36.14 | 10.0 | 262 | 79 | 175 | 4.2 | RCMT |
| 505 | 20160208 | 15:35:43 | 14.90 | 36.99 | 10.0 | 280 | 42 | 178 | 4.5 | RCMT |
| 506 | 20160306 | 08:12:36 | 16.74 | 38.20 | 39.0 | 248 | 75 | 158 | 4.0 | TDMT |
| 507 | 20160329 | 01:05:33 | 19.96 | 37.25 | 17.0 | 338 | 19 | 121 | 5.4 | CMT |



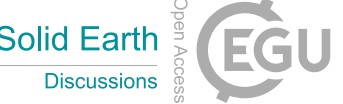

*Data availability*. Data used in the present study were collected from the databases of Istituto
Nazionale di Geofisica e Vulcanologia (www.ingv.it) and from catalogs and bibliographic sources
indicated in detail in the article.

*Author contributions*. All authors contributed to the scientific content, interpretations, message of
the paper, and the discussions. CT primarily worked on stress inversion of earthquake fault –plane
solutions, BO and DP on hypocenter locations in 3D velocity structures with the different
algorithms, GN linked the different contributions and produced the final paper with the support of
the other authors.

*Competing interests*. The authors declare that they have no conflict of interests.

*Acknowledgements*: This work has been performed in the framework of activity of the Research
Unit UNIME of CRUST - Interuniversity Center for 3D Seismotectonics with territorial
applications. Some figures were created using Generic Mapping Tools by Wessel and Smith (1991).

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



**Figures caption**

**Fig. 1.** Simplified sketch map of the Africa-Eurasia plate boundary. Black arrows indicate the
present-day sense of motion of Africa with respect to Eurasia according to Palano et al. (2015a).
Abbreviations: AP=Apennines, CA=Calabrian Arc, MA=Magrhebides. The dashed rectangle
including southern Italy and relative off-shore sectors indicates the main area of interest in the
present study (see Fig. 2). The top-left inset shows the western Mediterranean plate boundary
evolution in the last 30 Myrs (redrawn from Wortel and Spakman, 2000, with modifications
according to Neri et al., 2009). The solid curves with the sawtooth pattern indicate the location of
the boundary at different times as a consequence of ESE-ward rollback of the WNW-ward
subducting lithosphere. The sawteeth point in the direction of subduction or underthrusting. Black
sawteeth indicate in-depth continuous subducting slab in contrast to white sawteeth marking plate
boundary segments where slab detachment has occurred. The white arrow along the Apennines
shows the inferred direction of lateral migration of slab detachment.

**Fig. 2.** Tectonic map of southern Italy and relative off-shore areas (redrawn from Palano et al.,
2012, with integration of data in the Ionian Sea according to Polonia et al., 2011). Black arrows
indicate the present-day sense of motion of Africa with respect to Eurasia according to Palano et al.
(2015a). TFS stands for Tindari Fault System.

**Fig. 3.** Hypotheses of lithosphere fragmentation and microplate architecture proposed in the
literature for the central Mediterranean region. (a) Simplified tectonic setting of the central
Mediterranean region (redrawn from Cadet and Funiciello, 2004). (b) Main plate boundaries and
Adria microplate contours according to Anderson and Jackson (1987). RP indicates the pole of
rotation of Adria wrt Eurasia. (c) Macroplate and Adria representation in the view of Oldow et al.
(2002). (d) Adria separation in two independent blocks located between the main plates as reported





by Battaglia et al. (2004). (e) Sketch of the main tectonic and kinematic features in the central Mediterranean (redrawn from Serpelloni et al., 2007). (f) Adria and Apulia-Ionian-Hyblean blocks according to D'Agostino et al. (2008). (g-h) The two alternative scenarios proposed by Palano et al. (2012): (g) the Ionian domain is rigidly connected with the Sicilian-Hyblean-Malta block; (h) the Ionian domain diverges from the Sicilian-Hyblean-Malta block and moves together with the Calabrian block. (i) According to Pérouse et al. (2012) a rigid single block including the Hyblean Plateau, the Ionian basin, the Sirte plain, the Apulian peninsula and the southern Adriatic sea can be hypothesized, rotating clockwise wrt Africa (the star indicates the rotation pole). (j) New reconstruction of Adria domain between the main plates according to Sani et al. (2016). Shaded areas indicate uncertain limits among blocks.

**Fig. 4.** Section (a) displays the map of seismic stations used for hypocenter locations in the present study. The dashed rectangle indicates the study area for the earthquake locations reported in sections (b) to (d) of this figure. The continuous rectangles indicate the sectors Western Ionian (WI) and Sicily Channel (SC) where hypocenter locations were also performed by the Bayloc method (the results are shown in Figs 6 and 7). The shadowed curved belt shows the approximate location of the Apennine-Maghrebian chain in south Italy and Sicily. Section (b) displays the epicentral map of earthquakes of magnitude over 2.5 that occurred between 1981 and 2016 at depths less than 100 km in the area 10°-20°E 35°-41°N (circles are proportional to the earthquake magnitude, see legend). For these locations we used the linearized location method known as Simulps (Evans et al. 1994) and the 3D seismic velocity structure proposed for the study region by Orecchio et al. (2011). Sections (c) and (d) display the earthquakes of Section (b) after separation according to hypocenter depth (0-30 km and 30-100 km, respectively).

**Fig. 5.** Fault-plane solutions of earthquakes of magnitude over 2.5 occurring in the period 1977-2016 at depths less than 70 km in the area countoured by the dashed rectangle in Fig. 4. Only





solutions estimated by waveform inversion are reported. The main parameters and the bibliographic
source of each solution are given in Table A1, Appendix A. The different colors in the figure
identify different types of mechanisms following Zoback's (1992) classification based on values of
plunges of P and T axes: red = normal faulting (NF) or normal faulting with a minor strike-slip
component (NS); green = strike-slip faulting (SS); blue = thrust faulting (TF) or thrust faulting with
a minor strike-slip component (TS); black = unknown stress regime (U). "U" includes all focal
mechanisms that do not fall in the other five categories (Zoback, 1992). The beach ball size is
proportional to the earthquake magnitude (see legend). The curved line contours the study area for
stress inversion (results in Fig. 8)

**Fig. 6.** (a-c-d) Epicentral maps obtained by the Bayloc probabilistic method for the earthquakes
occurring during 1981–2016 in the sector WI, depth ranges 0-70 km (plot a), 0-30 km (plot c) and
30-70 km (plot d). For comparison, the structural information of Fig. 2 is reported. (b) Bayloc's
hypocentral vertical section along the AA′ profile indicated in plot a, +- 70 km around the profile.
AEF and IF stand for Alfeo-Etna Fault and Ionian Fault, respectively.

**Fig. 7.** (a-c-d) Epicentral maps obtained by the Bayloc probabilistic method for the earthquakes
occurring during 1981–2016 in the sector SC, depth ranges 0-70 km (plot a), 0-30 km (plot c) and
30-70 km (plot d). For comparison, the structural information of Fig. 2 is reported. (b) Bayloc's
hypocentral vertical section of earthquakes of plot (a) along a west-east profile.

**Fig. 8.** (a) Orientations of the principal stress axes (lower hemisphere stereographic projection)
obtained by inversion of the earthquake focal mechanisms shown in map. Red, green, and blue dots
indicate the orientations of the maximum (σ1), intermediate (σ2), and minimum (σ3) compressive
stresses, respectively. Crosses and squares indicate the 95% confidence areas for the σ1 and σ3
axes, respectively. F is the average of the individual earthquake misfits wrt the best model of stress





found by inversion (see text). (b) Stress inversion results obtained after subdivision of the study area
in two sub-areas W and E, west and east of the black line AB, respectively. (c) Stress inversion
results in the shadowed sector RZ where Polonia et al. (2017) have identified a rifting process with
opening in the SW-NE direction, approximately. See Table 1 for numerical values of stress
inversion results.

**Fig. 9.** This figure indicates the overall compressional domain caused by Africa-Eurasia
convergence in southern Italy disturbed by (i) extensional processes in the Calabrian Arc, (ii) rifting
in the westernmost Ionian offshore Sicily and (iii) subduction-related reduced compression in the
trench retreat zone offshore eastern Calabria. Black-to-grey transition of GPS crustal motion vectors
marks their clear orientation change from NW-ward to NE-ward and corresponds with the onshore
prolongation of the Ionian NW-trending rifting zone (diverging arrows are taken from Polonia et al.,
2017, GPS vectors are from Palano et al., 2012). Seismic data do not give significant information
on stress regimes in the white field of the southern Tyrrhenian sea.

**Table 1**. Stress tensor inversion of earthquake focal mechanisms performed for the earthquake sets
ALL, W, E and RZ described in the text and relative to the sectors indicated in Fig. 8. N is the
number of earthquakes (= focal mechanisms) belonging to the inversion set. F is the average of the
misfits of the individual earthquakes with respect to the best model of stress found by inversion. R
is the amplitude ratio $(\sigma2-\sigma1)/(\sigma3-\sigma1)$ where $\sigma1$, $\sigma2$, and $\sigma3$ represent the amplitudes of the
maximum, intermediate and minimum compressive stress, respectively. Pl and Az are the plunge
and azimuth, respectively, of the three main stress axes.






**Table 1.** Stress tensor inversion of earthquake focal mechanisms performed for the earthquake sets
ALL, W, E and RZ described in the text and relative to the sectors indicated in Fig. 8. N is the
number of earthquakes (= focal mechanisms) belonging to the inversion set. F is the average of the
misfits of the individual earthquakes with respect to the best model of stress found by inversion. R
is the amplitude ratio $(\sigma2-\sigma1)/(\sigma3-\sigma1)$ where $\sigma1$, $\sigma2$, and $\sigma3$ represent the amplitudes of the
maximum, intermediate and minimum compressive stress, respectively. Pl and Az are the plunge
and azimuth, respectively, of the three main stress axes.

| Set | N | F (°) | R | σ1 Pl (°) | σ1 Az (°) | σ2 Pl (°) | σ2 Az (°) | σ3 Pl (°) | σ3 Az (°) |
|-----|-----|-------|-----|-----------|-----------|-----------|-----------|-----------|-----------|
| All | 72 | 8.3 | 0.5 | 3 | 320 | 76 | 217 | 14 | 51 |
| W | 32 | 5.9 | 0.5 | 3 | 150 | 84 | 275 | 5 | 60 |
| E | 40 | 8.3 | 0.5 | 3 | 319 | 75 | 216 | 15 | 50 |
| RZ | 27 | 6.5 | 0.4 | 64 | 162 | 25 | 345 | 3 | 79 |





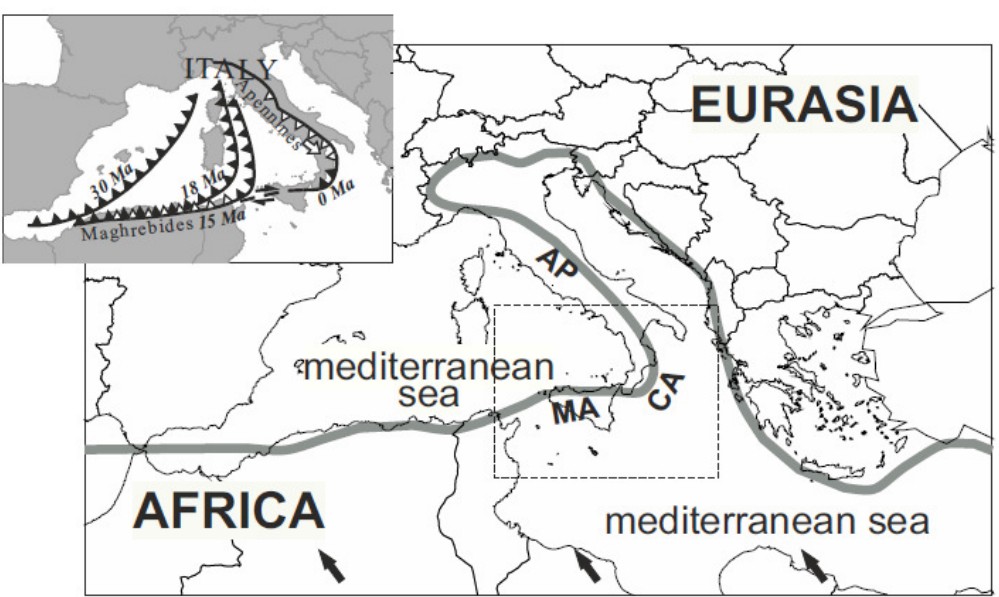

Figure 1





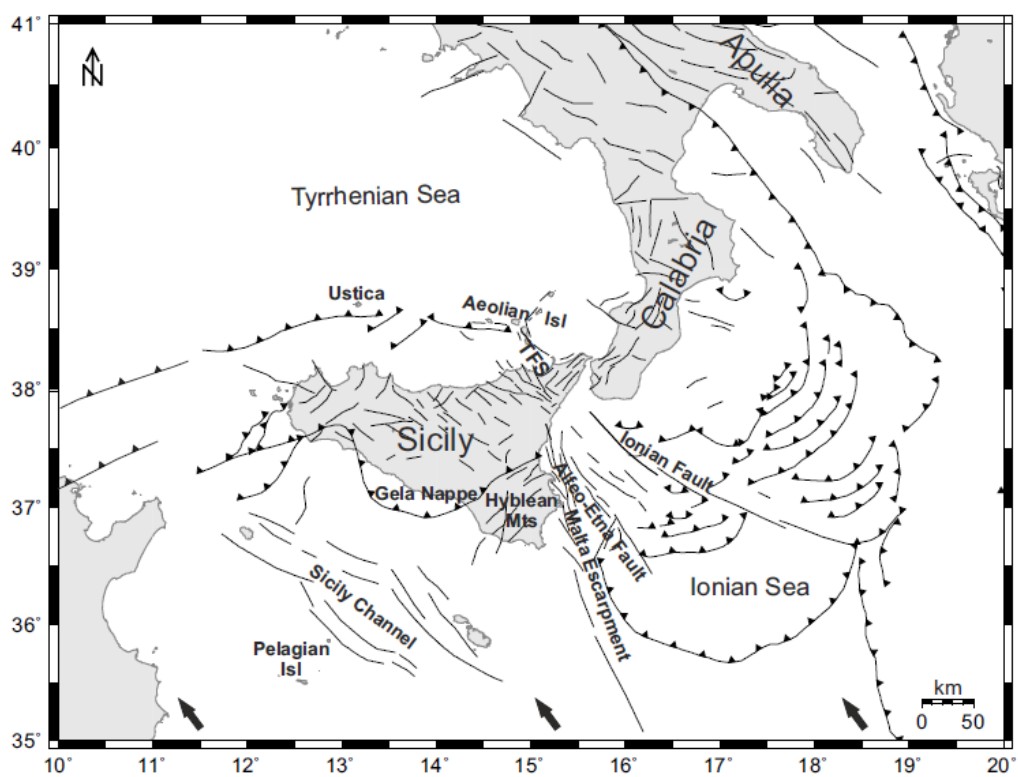

Figure 2





Figure 3






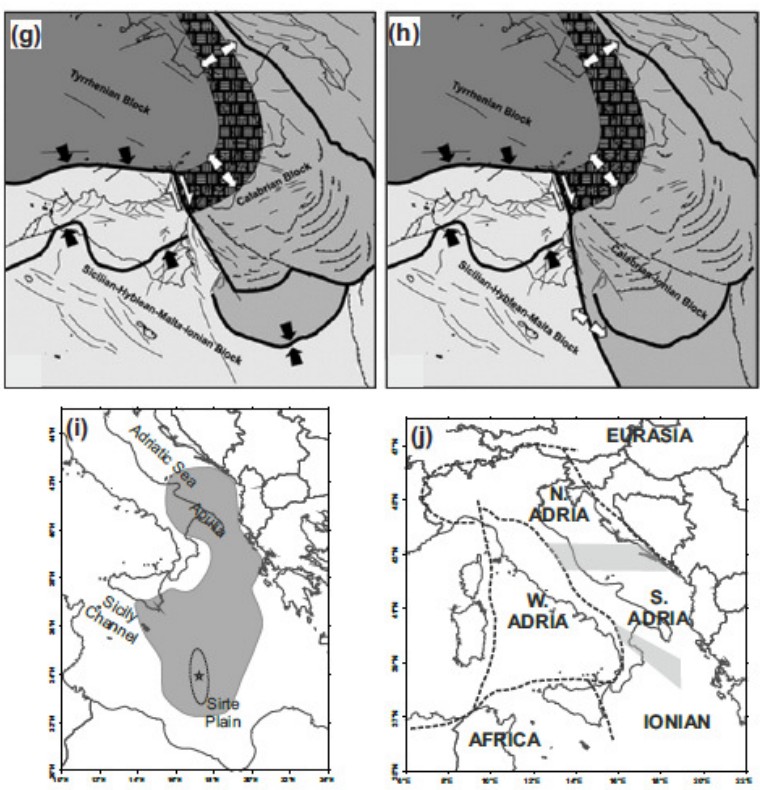

Figure 3 (continued)





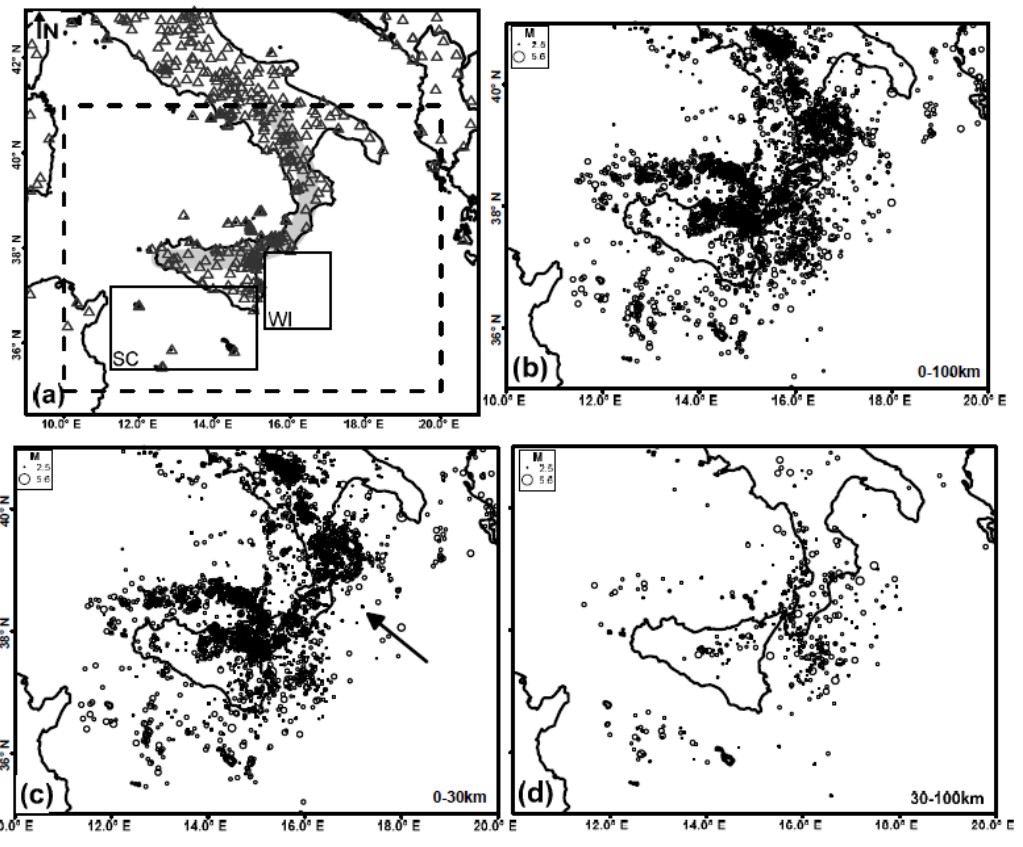

Figure 4





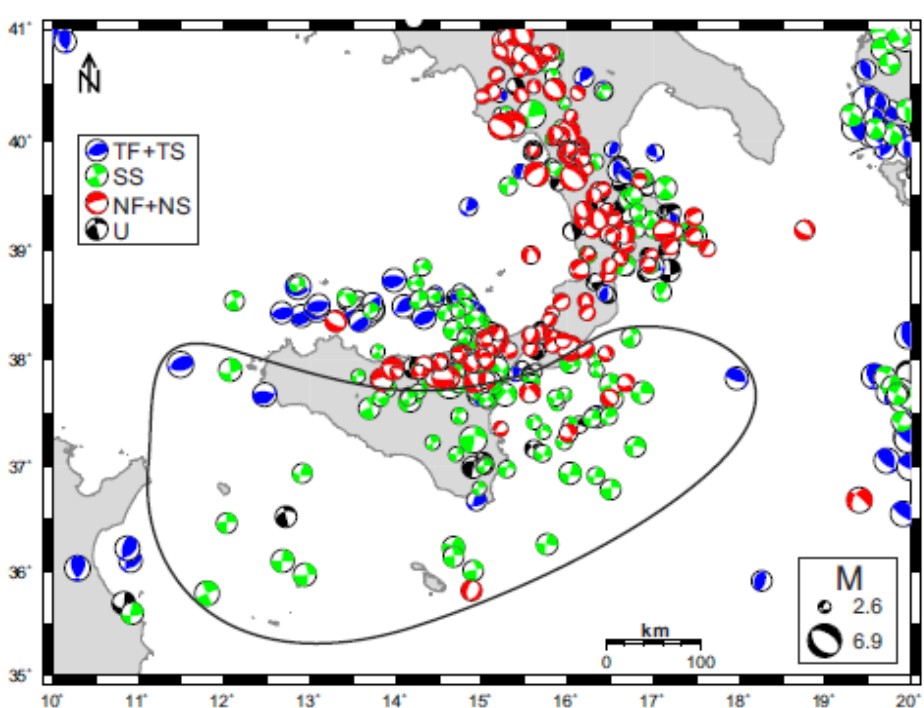

Figure 5





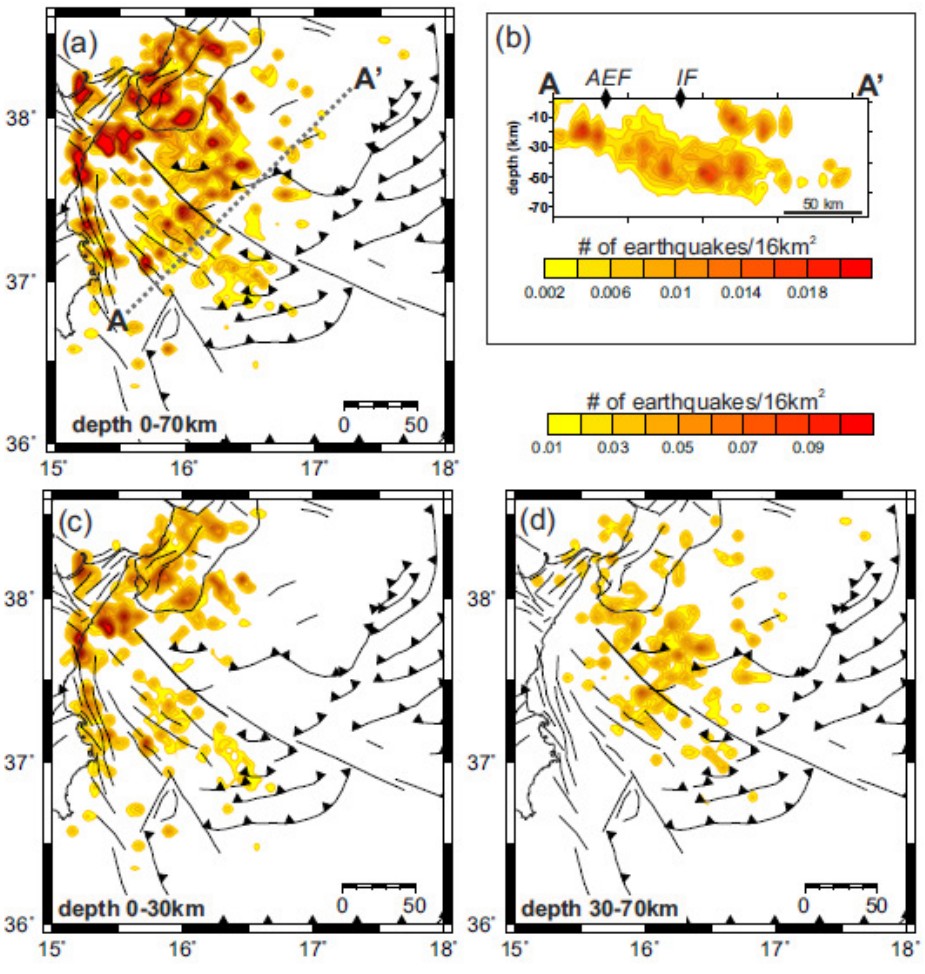

Figure 6





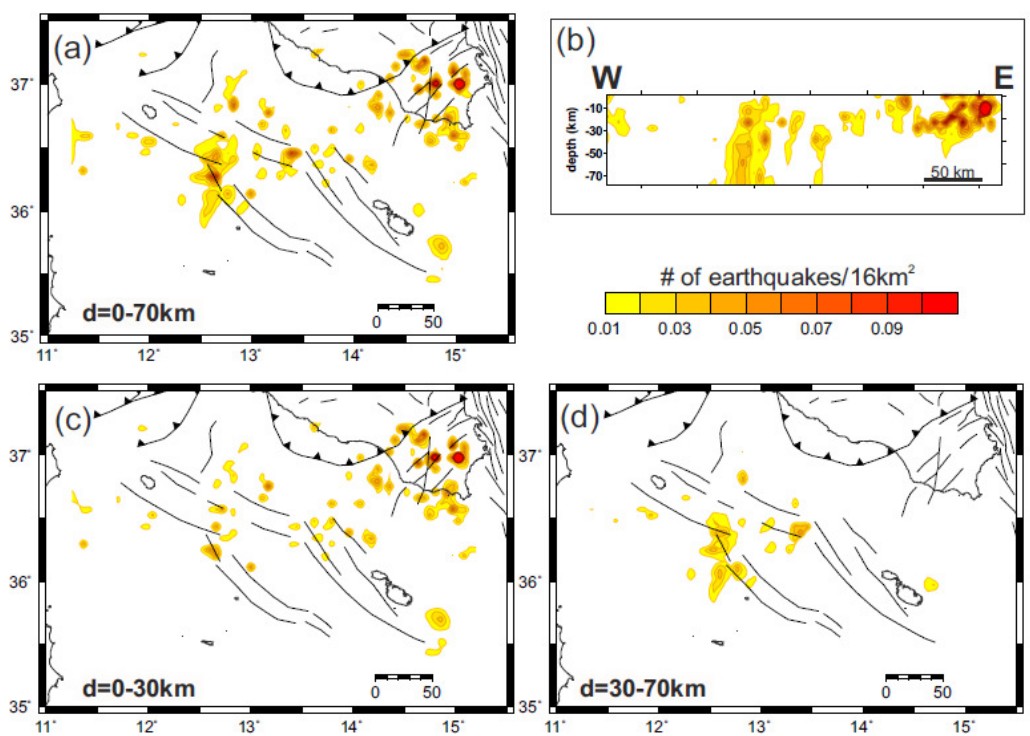

Figure 7









Figure 8



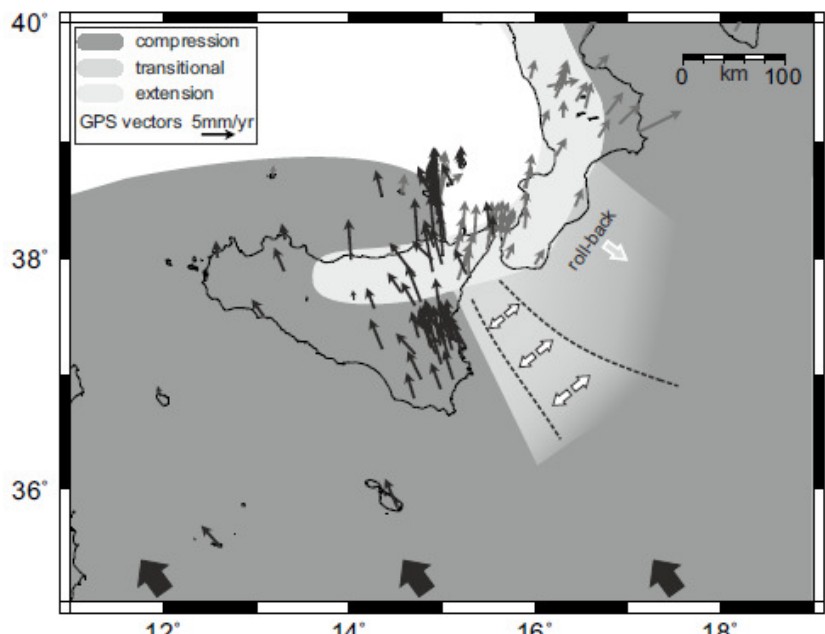

Figure 9
