# Peer review of "What seismicity offshore Sicily suggests about lithosphere dynamics and microplate fragmentation models in the Central Mediterranean"

_Solid Earth, 2018_

## Referee Comment (RC1) · Anonymous Referee #1 · 29 Nov 2018

In this manuscript, the authors reported the analysis of an updated seismic catalogue for the central Mediterranean area, namely, the surrounding region of Sicily. The authors analysed the seismic parameters (focal mechanisms and stress inversion) for events occurred both in the off-shore of the Messina straits, east of Sicily, and Sicily channel, south-west of Sicily, and on-shore, southern half of Sicily. The authors defined a separation, running roughly NE-SW, between the seismicity occurred in the Messina straits area and the on-shore seismicity plus seismicity of the Sicily channel. Such separation is commented in the framework of local geodynamic processes (slab retreat, STEP fault propagation and so on).

[Figure]

Overall, I found this study of limited interest due to the lack of a relevant new analysis of the seismic data and other main issues listed below. The authors seems to relocated off-shore events using Bayloc, which is a interesting operation due to the poor azimuthal coverage for such events, but, then, they did not move the analysis further. All the considerations about the stress distribution (i.e. the "main findings" of the paper) are based on previously published focal mechanisms. In my opinion, the manuscript should be rejected. I list below my main concerns.

Main points.

A. Limited new data analysis.

First of all, I specify that the methodology is poorly described and, also, uniformly distributed along all the manuscript. This took me a lot of time to clearly define the author's workflow for the analysis of the seismic data (I will go back to this point later). So, I could have mis-understood the process of data analysis. Say so, the new steps of analysis seem to be confined in: (a) relocation using Bayloc; and (b) computation of stress inversion. I consider the first step as a good idea, given the general poor location for off-shore events. However, it is not clear if the new locations impact, in any sense, on the main findings of this manuscript. That is: are the revised locations giving new fundamental information for the next stress inversion? A careful reading of the manuscript suggests a negative answer. Moreover, the seismicity distribution presented (for an area of hundreds of square kilometers) should be compared to the original one, to highlight the improvement with respect to standard locations. The revised catalogue should be published on a regional journal. Finally, I consider the "stress inversion" a tool for routine data analysis (to be clear, the references reported by the authors span between 1984 and 1996). I also use it to investigate my data-set, but I consider the results as a preliminary support to other, more complex, steps of data analysis. I discuss the separation of the stress inversion in two sectors below.

B. Segmentation of the stress inversion solutions.

The author claimed at line 308-310 (L308-310, hereinafter) that they obtained "more meaningful" results dividing the focal mechanisms in two areas and computing two different stress inversions. The authors overlooked to explain the meaning of "more meaningful" results. This could be fine, but then, few lines below (L337-339) they claimed that the E-W separation of the focal mechanisms, which was originally introduced by themselves without any detailed explanation, approximately corresponds to the Alfeo-Etna Fault. And this is a fundamental point in the findings of this manuscript. So, the author must illustrated how they produced the E-W separation of the stress inversions and on which criteria they based their claims.

C. Self-citation

The authors cited 20 of their own studies, out of 64 total references. This could be considered citation stack. Many of those self-citations come along with other citations or can be replaced, so the number of self-citations can be easily reduced without any impact on the manuscript.

D. Manuscript structure.

I found the manuscript not well organized and too long in many sections. For example, the Introduction and geodynamic background sections occupy 6.5 pages, while Data, methodology and results are described in 2.5 pages. Methodology is partially described in the Discussion section (e.g. see L300-311, where the authors introduced the partition of the stress inversions). Results should be clearly separated from Discussion section, while the authors describe details of the results throughout all the Discussion section.

Minor points

1. No new data.

The authors did not seem to add any additional seismic data in terms of raw recordings, new seismic stations, new pickings of seismic phases and so on. This is not

fundamental, but it must be clearly stated.

2. I'm not English native speaker, but I suggest to improve writing style. Also, writing style is somehow awkward (e.g. see L83-84, "below . . . cessation", where the subject is the "slab", but the slab is not close to cessation, is the slab retreat process that is close to cessation).

3. The authors used Bayloc to improve estimate of uncertainties in location parameters, which is a very good choice, from my point of view. But, then, they used, in the stress inversions, estimates of uncertainties on the focal mechanism parameters obtained from linearized inversions, which are, for such poorly located events (i.e. off-shore events) definitely un-realistic. Such estimates must be completely neglected. Focal parameters for off-shore events, with such poor azimuthal coverage, must have larger uncertainties than reported values.

---

## Author Comment (AC1) · 19 Dec 2018

The Referee 1 shows to have read with a certain attention the article. The overall conclusions he draws and reports in the first paragraph of his review are, however, influenced by some wrong axioms or preconceptions he clearly explicits in the list of "main concerns" or "Main points" A to D. The clear description of the Referee's arguments in the Main points A to D brings us to organize our replies starting from the Referee's criticisms reported in these points A to D.

Reply to Referee's Criticisms reported at Point A

[Figure]

There is a wrong axiom or a preconception in this point A when the Referee says: "I consider the stress inversion a tool for routine data analysis (to be clear, the references reported by the authors span between 1984 and 1996). I also use it to investigate my data-set, but I consider the results as a preliminary support to other, more complex, steps of data analysis". The Referee neglects that the application of a relative old (but of documented effectiveness) method may in many cases produce results which are able to improve the scientific knowledge. Although the efforts of the Referee to make "more complex steps of data analysis" can be fruitful and has to be appreciated, the advantages brought to Science by application of a relatively old method giving original information on the phenomena must not be refused. The Referee focuses only on the date of the stress inversion method used and does not consider the intrinsic complexity of the choice of the method of data analysis for a given study. We evidence that the literature shows that new stress inversion methods bring some advantages wrt older ones, but they may suffer of consequences of basic assumptions in many cases. For example, the relatively old method we used (GF84) is more conservative than new ones concerning the relative orientation of seismogenic stress and seismic dislocation surface, that is an important aspect in our study. We will furnish more information on this choice in the article.

There is another wrong axiom or preconception when the Referee says: "are the revised locations giving new fundamental information for the next stress inversion?" In this case the Referee assumes that the revised locations must contribute to stress inversion. He is probably used to follow this methodological path in his research and this is correct, but he neglegts that Science offers also other paths for improving knowledge. The information 'a' (earthquake locations) and 'b' (seismogenic stress inversion results) may contribute as "separate data" to the definition of a new model/theory. It is like in the case when 'a' is represented by gravimetric anomalies and 'b' by seismic data, i.e. two different types of information may concur to advance an hypothesis or to define a model (see, e.g., Neri et al., Int. J. Earth Sci, 2012). We are not absolutely obliged to state a close relationship between 'a' and 'b', or to use the first to obtain

the second, also because of the different scales of the processes of interest in the respective cases. The Referee seems more concentrated on the relationship between 'a' and 'b' than on their capability to improve the knowledge available on the geodynamic processes under investigation.

The Referee fears to have misunderstood the process of data analysis, but we can say that he has not, i.e. he has – in general - understood the process of data analysis. Because the Referee says that he has spent "a lot of time to clearly define the author's workflow" we are going to revise the text in order to solve this criticity.

The Referees writes: "the seismicity distribution presented (for an area of hundreds of square kilometers) should be compared to the original one, to highlight the improvement with respect to standard locations". It would be trivial – in our opinion - to compare our non-linear locations to the original ones from the INGV-CNT database. The latter were obtained by a linear method with a 1D velocity model of all Italy. The linear locations of the INGV-CNT database show unavoidable approximations witnessed, for example, by hypocenter depth fixed at 10 km for many events, a circumnstance well known to occur when the normal location process with four unknowns tends to fail. Also, we need to say that non-linear location methods have been proven to furnish more realistic estimates of location errors than linear ones at least since 2000 (Lomax et al., 2000; Lomax and MIchelini, 2001; and many others). It is well known and unanimously shared that the linear methods do underestimate location errors in the great majority of cases. For these reasons, the epicenter maps of low-location-error earthquakes obtained by linear methods are less accurate than maps obtained by non-linear methods. Concise information to be reported in the article to fulfill the eventual curiosity of the reader can be the rms of hypocenter locations by Bayloc vs SimulPS. We are going to include this comparison in the article.

Reply to Referee's Criticisms reported at Point B

We have limited for conciseness the information concerning the stress inversion runs,

but we understand that the Referee can ask this information. We are working to report the information requested as Supplementary Material.

Reply to Referee's Criticisms reported at Point C

Since several years the area/topic of interest of the present study do not receive particular attention by geophysicists and our team is one of the few teams which have worked on the topic in argument. This explains why the percentage of self-citations is 30%. This also explains why the rate of advancement of knowledge on this topic in this area is so low and allows us to bring significant progress of knowledge by the basic analyses reported in the present study. On the other hand, the problem of very limited approaches and rough knowledge of microplate geometry and kinematics in our study area was already posed by Nocquet in 2012 (paper cited in our article).

Reply to Referee's Criticisms reported at Point D

The Referee writes: "The introduction and geodynamic background sections occupy 6.5 pages, while Data, methodology and results are described in 2.5 pages".

The slow evolution of knowledge mentioned in our reply to Point C includes several contrasting views and still open questions on the plate/microplate geometry/kinematics in the study area. In our paper, the description of these views and a detailed list of the open questions are basic for helping the reader to understand what is the real progress of knowledge coming from our study. For this reason 6.5 pages may, in our opinion, be appropriate to frame our original contribution and to address future efforts.

Also, 2.5 pages (plus five Figures and two Tables!) for description of Data, methods and results should in our opinion be nearly appropriate, because: - we do not apply new methods, therefore we may give a relatively short description of the methods used and cite bibliography; - we present concisely the results in the 2.5-pages Section, whereas in the Discussion we comment/discuss the results also describing (when necessary) the details. Among other things, this helps reducing repetitions.

In any case, on the basis of the Referee's observations and misinterpretation reported at the Minor Points 1 and 3 (sse our Replies below), we are going to enlarge the presentation of data in the Section "Data, methods of analysis and results" with particular reference to (i) the description of new data analyzed in the present study wrt previous papers, (ii) the quality of focal mechanisms used for stress inversion, and (iii) the whole process of data analysis.

Reply to Referee's Criticisms reported at Minor Point 1

The statement of the Referee is wrong. Additional seismic data for hypocenter locations and focal mechanisms for stress inversion have been used in the present study with respect to the previous ones. For example, hypocenter locations have been performed on a dataset updated to December 31, 2016, while the previous study in the same area was updated to May 31, 2011 (Orecchio et al., 2014). 1420 earthquakes not belonging to the previous dataset have entered into the new one. This update has been mentioned several times in the paper (starting from the first line of the Abstract), however we are going to furnish more details on this point in the text.

Reply to Referee's Criticisms reported at Minor Point 2

We have understood that the Referee is not an English native speaker from many, repeated elementary grammar mistakes. "The authors seems to relocated . . ." "the authors must illustrated how . . ." and so on. Anyway, because a request to improve our English style and grammar is always pertinent, we will work for this purpose.

Reply to Referee's Criticisms reported at Minor Point 3

Concerning the uncertainty of focal mechanisms used for stress inversion, the Referee writes: "they used, in the stress inversions, estimates of uncertainties on the focal mechanism parameters obtained from linearized inversions, which are, for such poorly located events (i.e. off-shore events) definitely un-realistic. Such estimates must be completely neglected. Focal parameters for off-shore events, with such poor azimuthal coverage, must have larger uncertainties than reported values". The Referee's statement of un-realistic estimates of focal mechanism uncertainties for our off-shore earthquakes would have been correct if we had used focal mechanisms computed by inversion of P-onset polarities. As clearly explained in the article, we used only focal mechanisms estimated by waveform inversion, and the Referee's statement is wrong in this case. In this connection, we need to remark that the application of the CAP waveform inversion method to offshore earthquakes of the study area, together with the application of the non-linear procedure by Totaro et al. (2016) for fault parameter error estimates, show that uncertainties are of the order of 8-10 degrees, even less than the values of 10-15 degs we declared for our dataset. The declared values of 10-15 degs are conservative because we cannot control adequately the uncertainties of all the waveform inversion focal mechanisms taken from the official catalogs. Also, a main feature of the waveform inversion focal mechanisms (that the Reviewer seems to neglect) is that they are very stable when varying epicenter latitude and longitude and the focal depth in relatively large volumes around the hypocenter, this being widely demonstrated by a great number of tests published in previous papers (D'Amico et al., 2010, 2011; Totaro et al., 2016; Scolaro et al., 2017). We argue from the Referee's misinterpretation that we should enlarge this part of presentation of data and results in order to allow better understanding of the quality of our analysis. We are going to make this integration of text in the article.

Authors' concluding remarks

The Referee's axioms or preconceptions discussed in the above replies lead him to state that the manuscript should be rejected although he also requires at Main Points A and B additional clarifying information on the main results of earthquake locations and stress distributions. We remark that the real heart of the present work is that these results (i) contrast with the kinematic models of the study area presently available in the literature and (ii) answer some questions left open by the previous investigators.

In summary

- We do not know if the Referee is really convinced of his axioms.

- We know that he requires additional information on earthquake locations and stress distributions to verify the significance and usefulness of them.

- We are sure that he has not spent any word in his review on the main scientific problem, as posed by our work and also anticipated by Nocquet (2012) on the limitations of the approaches used for modeling geometry and kinematics of tectonic units in the Central Mediterranean.

We also tend to believe that a discussion with the Referee better focused on (i) improvements of the presentation of our results and (ii) comparison with the present geodynamic knowledge, would have been probably more fruitful than a discussion on the fundamentals of the Scientific Method.

With reference to what has been said above, we take the Referee's requests of

- (i) additional information on stress inversion runs and

- (ii) more detailed description of the contribution of Bayloc locations to our conclusions

as appropriate requests for improving the article and we are going to work on the paper for this purpose. Also, we will furnish more details in the text of the article on the structure/quality of the datasets used for hypocenter locations and stress inversion, in order to avoid misunderstandings and misinterpretations.

---

## Referee Comment (RC2) · Anonymous Referee #2 · 17 Feb 2019

The presented research work concerns an area which is still poorly understood, though close to the "heart" of Europe, therefore this is, in principle, a very interesting paper. The manuscript contains two (2) main technical results: a) A focused relocation for two areas south and to the east of Sicily and, b) a revised stress tensor inversion, for the same areas, including the southern section of Sicily located between the two previously mentioned regions. These findings are employed to discuss the geotectonic setting of this area of interest and propose a revised seismo-tectonic model for the region.

Unfortunately, I found the manuscript to contain several technical problems and obscure points in the data processing, that really make the whole approach questionable:

The reader cannot trust the results, therefore why should he proceed to evaluate the interpretations. Moreover, it is evident that there is no evaluation on what are the new results that this manuscript contributes, and which are used for the geotectonic model interpretation. In other words, the paper has both technical and originality issues, which lead me to the suggestion that it cannot be published. Personally, I think that the authors need to completely revise the manuscript and their approach. They need to really provide new data processing and information, and only then tackle the geotectonic model interpretation, which is their primary goal. In the following, I am making some specific comments to explain the main problematic issues and provide some specific suggestions to the authors.

Language: The manuscript is not bad, but in some sections or sentences the English is poor and the reader struggles to understand what the authors are saying. In some cases, the authors use complicated (but not accurate language), making the whole text a puzzle, e.g. in lines 63-66: "...The potentialities of the methods of seismological analysis we have planned to use, in part of recent conception and in all cases already proven to be effective in the study region (see e.g. Neri et al., 2005; Billi et al., 2010; D'Amico et al., 2010; Presti et al., 2013), make us confident that the above declared goal of the investigation may be reached...", where they are essentially saying that their methods are working for the study area, so they expect to realize the manuscript targets, but in very difficult to read (and not correct in some cases) language (by the way, I would suggest to avoid such statements: Let the reader evaluate if they are doing a good job, on the basis of the data and methods, you do not need to advertise this in advance!).

I would strongly suggest that the manuscript is read by an English language expert and focus on the simplification of expressions and complicated statements. Please use smaller paragraphs (in section 4, one paragraph starts in page 11 and ends in page 13!) to facilitate the reader to follow your arguments.

Organization, Figures and Tables: The paper is organized in three (3) parts. The first

presents the study area, existing models (for the broader area) and input data, and corresponds to Figures 1 to 5. The second section concerns the technical results (Figures 6, 7 and 8), while the last section and Figure 9 present the final model, proposed on the basis of the technical results. This is clearly not well-balanced. The authors spend 10 pages for the 1st part, 2.5 pages for the technical result presentation and 8.5 pages for Discussion-Conclusions. I agree with the 1st reviewer, the manuscript is lacking a proper data processing presentation. I am not talking about the data interpretation, but the actual commenting and evaluation of the result reliability (comparisons of relocations with initial catalogue, stress presentation on a map including comparison with previous findings by the same group such as Totaro et al. (2016) in a figure, and quantification of the new information that has been determined by the data processing, etc.).

The first part is simply too long: I found the discussion of Figure 3 interesting but tiring (for the reader) and in some places irrelevant. The authors present sequentially all models, though these models focus in different areas, e.g. Fig. 3g and h concern their region, most other models include the whole Adria and even northern Italy, where some of the discussed model differences have little to no implication for their area of interest. Because models are so different and are based on different datasets, the authors cannot explain in detail the models: For example, what exactly are we seeing in Fig.3g and 3h (not explained in the legend), what are the colored zones in 3e and why they have different colors (I suspect it is the faulting pattern, but why should the reader guess?)?

I understand that it is difficult to do this model comparison in a technical (and not a review paper). The authors should consider avoiding the discussion about the whole Italian, Adria, Tyrrhenian Sea, etc. territory and make smaller, focused model comparison only for the smaller area of interest. They do not need to discuss all models, just those ones that have fundamentally different implications for the geotectonic setting of the broader Sicily area. Since some models are evolutions of previous ones, stick to

the important ones and their main features. Presenting 3-4 models for the study area, where the main model features are discussed, instead of 10 models for a very large area, will make the manuscript smaller and better, more focused. I know that the authors want to present the "general" picture, but this belongs to a review article, not this one.

Data processing and Discussion sections: This is the main reason that I recommend rejection for the paper. My main reservations are explained in detail, and have to do with the reliability of the data processing, the input information, and the definition of the added value obtained from the relocations and stress inversion. More specifically:

A) The first technical target of the manuscript is to relocate seismicity in a relatively large area. The authors use the SIMULPS linearized code and the BayLoc non-linear locations, with a 3D velocity model (Orecchio_et_al., 2011). There are several problems here:

1) SIMULPS is not a location tool. Though you can optimize locations using a 3D model with SIMULPS, taking advantage of the linearization of the travel-time equation and solving only for locations (and this is what people have done sometimes), it does not have the capabilities built in standard linearized locations algorithms, even older ones (e.g. HYPOINVERSE), like azimuthal and distance weighting, outlier iterative rejection with progressing smaller misfit tolerance during location iterations, etc. It is correct that locations (with a large number of P and S arrivals) move spatially when using SIMULPS, responding to the improved velocity model, but at the same time they tend to disperse/defocus (especially events with a smaller number of phases), since SIMULPS is not made to handle these events. There are much better approaches for linearized inversions. A good choice for the authors would be to use TOMODD (Zhang and Thurber, 2003), which allows for the use of a 3D model, performs double-difference relocations (which really improve locations, like HypoDD) and has improved outlier rejection capabilities. SIMULPS is efficient, but simply not made to be a reference algorithm for linearized locations using 3D models, it is mainly a tomographic model

algorithm.

2) The authors claim to use a 3D P-velocity model (Orecchio_et_al., 2011), which improves the locations. However, this is clearly not the case. In the attached figure of this review (Fig. 1), I have superimposed the resolution area (red dashed line) of employed 3D model, as presented in Fig. 6 in Orecchio et_al. (2011) for the depth of 10km, which is the depth for which the resolved area shows the largest spatial extent (for all other depths e.g. 20km and 30km this area is smaller, therefore the actual 3D model resolution is worse than what I show here). The 3D model model stops at the depth 40km (see the plot in Fig 6, which is identical to the starting (a priori) model of Fig. 4 of Orecchio et_al. (2011)). In fact, the 3D model has section where the maximum resolved depth is at $\sim$20-30km (see section B-B' in Fig. 6 of the same work). Outside the red line there is no 3D model, just the starting (a priori) model that the authors have built from a limited number of seismic profiles, and which they present in Fig.4 of the Orecchio et_al. (2011).

From this figure several things are evident and several questions can be asked:

a) The Western Ionian (WI) area is outside the 3D model. Therefore, locations for this area of focus really do not have a 3D model determined from the inversions of Orecchio et_al. (2011), just an a priori model, which is clearly of much, much poorer accuracy.

b)The Sicily Channel (SC) area is even outside the a priori model of the Orecchio et_al. (2011), which the authors claim to use. I am wondering how did the authors actually perform any locations in this area where they focus their attention; did they expand the 3D model? How was this done?

c) What about locations with SIMULPS for the remaining area (black dashed line in the previous figure), which are clearly outside the 3D-model (red dashed line) or even the a priori, starting model (blue dashed line) of Orecchio et_al. (2011)? What did they do for these events? No information is provided in the manuscript.

d) The recovered Orecchio et_al. (2011) 3D model stops at 20-40km (depending the area). The a priori (starting) model stops at 40km. What model was used for locations down to 100km (as stated in Fig. 4d) or for the range 30-70km for Figures 6d and 7d?

e) What S model was used, since the Orecchio et_al. (2011) model is a P model? How was this constrained?

It is clear that there is no 3D tomographic model for either focus areas (WI or SC) and not even an educational guess model for the SC area (at least the authors do not describe one). This simply means there is no added accuracy in the recovered epicenters, and that the reported location errors (5km and 9km hypocenter errors for WI and SC) are simply super optimistic. It is well known that errors reported from any location algorithm as biased to low values when the models are wrong, especially when there is a large azimuthal gap, such as the case for WI and SC where the network is one-sided. Epicenters far from the coast may be of by >10 (or even tens) of km, especially if the crust is very different (e.g. fast crust in SC or WI areas, which are more similar to remnants of an old ocean, compared to the continental [slower velocities] of Sicily or southern Italy areas).

This is evident in the results: The locations in SC south of Lat~36.5 show this near vertical depth distribution from 10 to 70+ km, appearing in the cross-sections (e.g. Fig. 7) as a vertical band of events, typical of very poor depth control. The locations for the depth range 30-70km (mantle range) look meaningless, and rather random, not providing some important seismotectonic information.

4) Even if the reader tries try to forget the previous issues, we still are missing the most critical point: The demonstration of the improved accuracy of new relocations. The authors present no comparison (e.g. initial [catalogue] distribution of events, revised relocations with SIMULPS, focused relocatios with BayLoc in the focus area). What, exactly, did they (and we, as readers) gained from this whole effort? What seismo-tectonic features are now seen with the relocations, that are not observed e.g. in the

catalogue data? The authors do not say, and the reader is left to wonder about the technical data reliability and contribution to this work.

5) A minor issue: It is not possible to make quantitative comments on seismicity with this dataset, e.g. lines 284-285 "...Also, a clear drop of activity can be noted in the Ionian offshore of Southern Calabria (Figs 4c and 5)...". The authors have not studied or presented the catalogue completeness, or its spatial variability. Therefore, seismicity changes may be apparent e.g. poorer detection and location of small events, as we move away from the Sicily coast, due to network detectability. First, they need to demonstrate the catalogue space-time completeness, then show only data that are within this space-time completeness intervals, and only then can they make quantitative assessments of seismicity.

B) The second technical target of the manuscript is to define a revised stress field for the study area, using an updated Fault Plane Solution (FPS) catalogue. Again, the reader is confused:

1) The authors use the famous GF method, and some rather old criteria for stress homogeneity, from >20 years ago. While the work of Wyss et al. (1992) and Gillard et al. (1996) is reliable, the GF method has received a lot of criticism, and several modifications (e.g. Michael, 1987, Lund & Slunga, 1999, Vavrycuk, 2014, 2015, Karagianni et al., 2016) have been proposed that overcome the GF problems, either by adopting a different solution stability check, or by using different criteria such as the instability criterion adopted initially by Lund and Slunga (1999). Why did the authors not compare their results with those obtained by other, more recent methods, in order to verify the robustness of the recovered stress field?

2) The zonation procedure and the presentation-discussion of the recovered stress field is completely obscure and definitively was performed in a different way than the approach described by the authors. The authors state (lines 300-311) that they subdivided the FPS in tens subcatalogues, using their spatial characteristics, and made

sure to have at least 20 FPS in each group. They then computed the stress and used the F value to assign stress homogeneity, based on some previously proposed cut-off values. The F-values were used to decide on spatial clustering.

However, it is completely unclear how this was done. How do you separate FPS in a semi-automatic manner, and finally end up proposing areas separated by a straight line (Figures 8b and 8c)? How was this separation performed and what is the significance of these lines? Are we to simply trust the authors that these boundaries somehow magically appeared in their systematic search? The authors do not show any other results (e.g. in an electronic appendix) but simply state that they decided to "...report...results that we consider more meaningful...". Meaningful in what sense? According to which criteria, or even subjective evaluation? For example, how was the RZ area actually defined? A minor issue: How was he earthquake magnitude used in the FPS grouping, as the authors state in line 302? To the best of my knowledge, the earthquake magnitude can be taken into account as a weight, e.g. in the inversion process, but not in the sub-area separation. Please clarify.

3) The results are really not presented in a clear manner. First, it is clear that the only consistent stress axis is the $\sigma3$, horizontal extension (see last column of Table 1) and that the separation that they perform leads to a $\sigma1$ instability for the East part (E in Fig8b, and RZ in Fig. 8c). In my opinion, this simply means that they are mixing FPS belonging to very different regimes. For example, Fig.6a shows that the depth distribution in WI is complicated. Have the authors separated seismicity with depth, or simply included all FPS from shallow crustal depths (0-20km) to upper mantle (50-70km) in the same stress inversion? Is this realistic? In a subduction area you often have transpresional deep events along the slab, with normal events at shallow depths (back-arc area); grouping events from all depths, upper crust to sub-Moho mantle, will certainly give you a very heterogeneous (and probably wrong) stress field.

The stress recovered for RZ and E are meaningless. The 2-sigma area for $\sigma1$ and $\sigma2$ is practically a vertical plane (also identified by the authors), suggesting that compression

changes and that the dataset can not recover the changes, most probably due to the adopted spatial grouping.

4) Even without considering the previous problems, no discussion and visual comparison is made with previous results. Authors from the same group (Totaro et al., 2016) present results for the WI and part of the SC areas (see their Fig. 3). The stress recovered for area 6 of Totaro et al. (2016) is nearly identical to the one shown by the authors in Fig.8b for the East area (E). Visual comparison of the FPS datasets also confirms that the majority of FPS is identical, which explains the similar results. Therefore, what additional information did the additional FPS bring into the final results? What is the new information on the stress field of the WI area?

For the West (W in Fig 8) area, the authors have many more new FPS, but the obtained result for this area are very similar to the results obtained for the (smaller) area 7 by Totaro et al. (2016). Therefore, what is the new knowledge that we gain by the authors (very unclear) processing, that is used by the authors for their model and that was not present and discussed by themselves in Totaro et al. (2016), a few years ago?

5) Minor issue: The authors state that the stress in the west (W) is more consistent with the average from all data (ALL, Fig 8a). However, it is the East cluster (E) stress that is identical to the overall average (see Table 1). In fact, stress for W presents a ∼10 clockwise rotation with respect to ALL and E, which should be considered as significant, if the FPS errors are of the order of 10 degrees, as the authors state. In other words, the numbers tell a slightly different story, that the one presented by the authors.

I could provide additional comments on the Discussion section. Perhaps an important one is that, after an extensive introduction, spending nearly one page (p. 13) to present and discuss features already known (as the authors themselves state), seems like a waste of time. However, specific comments are simply meaningless, since the input information (seismicity relocations and stress inversion) are not reliable: The process-

ing has a lot of obscure points, even some that can be considered as partly flawed, weak presentation and no identification of the added value gained by new, additional information. This makes the last part of the manuscript practically irrelevant and hard to evaluate.

My last remark is a rather personal one: Reading the response of the authors to the comments of the first reviewer, I would strongly recommend to the authors to adopt a less aggressive attitude to reviewer comments. A reviewer that is not a native English speaker, can still understand that the English of the manuscript is rather poor, therefore there is no point to sarcastically address his criticism by mocking his own English fluency and accuracy. The same applies for the other scientific comments: I agree with the overall assessment of reviewer 1, that the manuscript appears overloaded in the Introduction and very weak in the technical section, as I also explained. I would strongly recommend that the authors take the advice proposed here, which is what two readers independently suggest, and use it to completely re-process data and re-write the paper, and not simply neglect these comments as negative criticism.

―――――――――――――――――――

**Fig. 1.** Superposition of Fig.4a of the submitted ms with the 3D model area (red dashed line) and the starting model area (blue dashed line) from Orecchio et_al. (2011)

---

## Author Comment (AC2) · 14 Mar 2019

**Replies of the Authors to Reviewer 2**
**and information concerning our reply of Dec 19, 2018 to Reviewer 1**
* * *
**Replies of the Authors to the Comments by Reviewer 2**

After a general introductory comment, the Reviewer 2 explicited his criticisms in detailed comments. We list below each Reviewer's detailed comment and our corresponding reply indicating also the changes made in the MS.

These changes are reported in the Chapter "Changes" of the present document.

In this document, we have reported only concise bibliographic references (author, journal, year) because we consider that the papers cited (for example, those proving the important role that Gephart and Forsyth's (1984) method still plays today in the stress inversion practice) can be easily reached in the international literature without additional data.

_Comment_

_Language: The manuscript is not bad, but in some sections or sentences the English is poor and the reader struggles to understand what the authors are saying. In some cases, the authors use complicated (but not accurate language), making the whole text a puzzle, e.g. in lines 63-66: ": : :The potentialities of the methods of seismological analysis we have planned to use, in part of recent conception and in all cases already proven to be effective in the study region (see e.g. Neri et al., 2005; Billi et al., 2010; D'Amico et al., 2010; Presti et al., 2013), make us confident that the above declared goal of the investigation may be reached: : :", where they are essentially saying that their methods are working for the study area, so they expect to realize the manuscript targets, but in very difficult to read (and not correct in some cases) language (by the way, I would suggest to avoid such statements: Let the reader evaluate if they are doing a good job, on the basis of the data and methods, you do not need to advertise this in advance!)._

_I would strongly suggest that the manuscript is read by an English language expert and focus on the simplification of expressions and complicated statements. Please use smaller paragraphs (in section 4, one paragraph starts in page 11 and ends in page 13!) to facilitate the reader to follow your arguments._

Reply

We will improve the English language and style, also by simplification and shortening of several phrases. We will do this work with an English language expert.
* * *
*Comment*

*Organization, Figures and Tables: The paper is organized in three (3) parts. The first presents the study area, existing models (for the broader area) and input data, and corresponds to Figures 1 to 5. The second section concerns the technical results (Figures 6, 7 and 8), while the last section and Figure 9 present the final model, proposed on the basis of the technical results. This is clearly not well-balanced. The authors spend 10 pages for the 1st part, 2.5 pages for the technical result presentation and 8.5 pages for Discussion-Conclusions. I agree with the 1st reviewer, the manuscript is lacking a proper data processing presentation. I am not talking about the data interpretation, but the actual commenting and evaluation of the result reliability (comparisons of relocations with initial catalogue, stress presentation on a map including comparison with previous findings by the same group such as Totaro et al. (2016) in a figure, and quantification of the new information that has been determined by the data processing, etc.).*

*The first part is simply too long: I found the discussion of Figure 3 interesting but tiring (for the reader) and in some places irrelevant. The authors present sequentially all models, though these models focus in different areas, e.g. Fig. 3g and h concern their region, most other models include the whole Adria and even northern Italy, where some of the discussed model differences have little to no implication for their area of interest. Because models are so different and are based on different datasets, the authors cannot explain in detail the models: For example, what exactly are we seeing in Fig.3g and 3h (not explained in the legend), what are the colored zones in 3e and why they have different colors (I suspect it is the faulting pattern, but why should the reader guess?)? I understand that it is difficult to do this model comparison in a technical (and not a review paper). The authors should consider avoiding the discussion about the whole Italian, Adria, Tyrrhenian Sea, etc. territory and make smaller, focused model comparison only for the smaller area of interest. They do not need to discuss all models, just those ones that have fundamentally different implications for the geotectonic setting of the broader Sicily area. Since some models are evolutions of previous ones, stick to the important ones and their main features. Presenting 3-4 models for the study area, where the main model features are discussed, instead of 10 models for a very large area, will make the manuscript smaller and better, more focused. I know that the authors want to present the "general" picture, but this belongs to a review article, not this one.*

Reply

We would have preferred to conserve the description of the models proposed for the broader region because the modelling of boundaries of blocks and microplates far from the main study sectors (Sicily Channel and Western Ionian) has shown to have strong implications in the modelling of block boundaries in SC and WI. This is also related to the assumption of block rigidity adopted by previous investigators. A wider view of the modelling process in the last decades could help the reader to better frame our analysis and the value of our results in contrast to the previous models based on assumption of block rigidity.

On the other hand, we also understand the reasons of the Reviewer requiring to shorten the description of models in this technical (and not review) article. We have made the change requested by the Reviewer by presenting 4 models instead of 10 in Figure 3 and shortening also significantly the description of the state of art of knowledge in the text of the article, so that the old Section 3 has been incorporated in Section 2 (see the Chapter "Changes", lines 8-100 and new Fig. 3).

Concerning the comments on data and processing we are going to reply to the specific comments below.
* * *
*Comment*

*Data processing and Discussion sections: This is the main reason that I recommend rejection for the paper. My main reservations are explained in detail, and have to do with the reliability of the data processing, the input information, and the definition of the added value obtained from the relocations and stress inversion. More specifically:*
*A) The first technical target of the manuscript is to relocate seismicity in a relatively large area. The authors use the SIMULPS linearized code and the BayLoc non-linear locations, with a 3D velocity model (Orecchio_et_al., 2011). There are several problems here*

Reply

We reply to the detailed comments below.
* * *
*Comment A – 1)*

*1) SIMULPS is not a location tool. Though you can optimize locations using a 3D model with SIMULPS, taking advantage of the linearization of the travel-time equation and solving only for locations (and this is what people have done*

*sometimes), it does not have the capabilities built in standard linearized locations algorithms, even older ones (e.g. HYPOINVERSE), like azimuthal and distance weighting, outlier iterative rejection with progressing smaller misfit tolerance during location iterations, etc. It is correct that locations (with a large number of P and S arrivals) move spatially when using SIMULPS, responding to the improved velocity model, but at the same time they tend to disperse/defocus (especially events with a smaller number of phases), since SIMULPS is not made to handle these events. There are much better approaches for linearized inversions. A good choice for the authors would be to use TOMODD (Zhang and Thurber, 2003), which allows for the use of a 3D model, performs double-difference relocations (which really improve locations, like HypoDD) and has improved outlier rejection capabilities. SIMULPS is efficient, but simply not made to be a reference algorithm for linearized locations using 3D models, it is mainly a tomographic model algorithm.*

Reply to comment A-1)

Our team has used HypoDD with 3D models in previous works (Totaro et al., EGU 2014; Totaro et al., ESC 2014; Totaro et al., Bull. Seism. Soc. Am., 2015; Brozzetti et al., Jour. Struct. Geology, 2017; among others). It is well known that HypoDD is especially useful in regions with a dense distribution of seismicity, such as in the case of investigations of seismic swarms or sequences where earthquakes are strongly clustered in space and time (see, among others, Waldhauser, 2001; Waldhauser and Ellsworth, 2002; Richards et al., 2006). Waldhauser (2001) wrote: "This approach is especially useful in regions with a dense distribution of seismicity, i.e. where distances between neighboring events are only a few hundred meters". Richards et al. (2006) wrote: "Thus, double-difference results are best if events are densely distributed (a few km event separations when using local stations separated by several tens of km)". Our experience fully confirms these statements evidencing that, in the cases of less dense seismicity like in the present study, HypoDD is less powerful and tend to lose relatively large percentages of events during computation.

We also need to remark that we have not used SIMULPS as *reference* linearized algorithm for comparison to our non-linear locations. Non-linear locations have been widely proven to be more accurate than linearized ones when the recording network geometry is not optimal (Lomax *et al.* 1998, 2000; Lomax and Michelini, 2001; Husen and Smith, 2004; Lippitsch *et al.*, 2005; among others). We used SIMULPS only for earthquake locations in the broader region of southern Italy because, as explained in the original MS, Bayloc's computations become very heavy when large areas/volumes are investigated.
Anyway, considering that
  (i) we are now definitely restricting our location analyses to the most crucial SC and WI sectors;
  (ii) the Reviewer in the forthcoming Comment A-4 asks "*What seismotectonic features are now seen with the relocations, that are not observed e.g. in the catalogue data?*", we have worked as follows in the revised MS:
- (a) we have performed locations by Bayloc in sectors SC and WI (as in the original MS);
- (b) we have commented our Bayloc's locations in SC and WI compared to the catalogue data;
- (c) we have reported the catalogue locations in the new Figure 4 (broader region).
* * *
*Comment A – 2)*

*2) The authors claim to use a 3D P-velocity model (Orecchio_et_al., 2011), which improves the locations. However, this is clearly not the case. In the attached figure of this review (Fig. 1), I have superimposed the resolution area (red dashed line) of employed 3D model, as presented in Fig. 6 in Orecchio et_al. (2011) for the depth of 10km, which is the depth for which the resolved area shows the largest spatial extent (for all other depths e.g. 20km and 30km this area is smaller, therefore the actual 3D model resolution is worse than what I show here). The 3D model model stops at the depth 40km (see the plot in Fig 6, which is identical to the starting (a priori) model of Fig. 4 of Orecchio et_al. (2011)). In fact, the 3D model has section where the maximum resolved depth is at _20-30km (see section B-B' in Fig. 6 of the same work). Outside the red line there is no 3D model, just the starting (a priori) model that the authors have built from a limited number of seismic profiles, and which they present in Fig.4 of the Orecchio et_al. (2011). From this figure several things are evident and several questions can be asked:*
*a) The Western Ionian (WI) area is outside the 3D model. Therefore, locations for this area of focus really do not have a 3D model determined from the inversions of Orecchio et_al. (2011), just an a priori model, which is clearly of much, much poorer accuracy.*
*b) The Sicily Channel (SC) area is even outside the a priori model of the Orecchio et_al. (2011), which the authors claim to use. I am wondering how did the authors actually perform any locations in this area where they focus their attention; did they expand the 3D model? How was this done?*
*c) What about locations with SIMULPS for the remaining area (black dashed line in the previous figure), which are clearly outside the 3D-model (red dashed line) or even the a priori, starting model (blue dashed line) of Orecchio et_al. (2011)? What did they do for these events? No information is provided in the manuscript.*
*d) The recovered Orecchio et_al. (2011) 3D model stops at 20-40km (depending the area). The a priori (starting) model stops at 40km. What model was used for locations down to 100km (as stated in Fig. 4d) or for the range 30-70km for Figures 6d and 7d?*
*e) What S model was used, since the Orecchio et_al. (2011) model is a P model? How was this constrained?*
*It is clear that there is no 3D tomographic model for either focus areas (WI or SC) and not even an educational guess model for the SC area (at least the authors do not*

*describe one). This simply means there is no added accuracy in the recovered epicenters, and that the reported location errors (5km and 9km hypocenter errors forWI and SC) are simply super optimistic. It is well known that errors reported from any location algorithm as biased to low values when the models are wrong, especially when there is a large azimuthal gap, such as the case for WI and SC where the network is one-sided. Epicenters far from the coast may be of by >10 (or even tens) of km, especially if the crust is very different (e.g. fast crust in SC or WI areas, which are more similar to remnants of an old ocean, compared to the continental [slower velocities] of Sicily or southern Italy areas).*

*This is evident in the results: The locations in SC south of Lat_36.5 show this near vertical depth distribution from 10 to 70+ km, appearing in the cross-sections (e.g. Fig. 7) as a vertical band of events, typical of very poor depth control. The locations for the depth range 30-70km (mantle range) look meaningless, and rather random, not providing some important seismotectonic information.*

Reply to Comment A-2)

We have cited a wrong reference (Orecchio et al., 2011), we are sorry for having misled the Reviewer, the right reference is Neri et al. (2012).

As we now explain in the new text (Chapter "Changes", lines 144-153), Neri et al. (2012) basically integrated the velocity structures proposed by Orecchio et al. (2011) and Neri et al. (2009) for the depth ranges 0-40 km and 65-260 km, respectively. Both the shallow and intermediate-depth velocity structures were the result of P- and S-wave tomographic inversions, with velocity values in the nodes external to the well resolved inversion zones assigned as follows. Velocity values outside the inversion zone of Orecchio et al. (2011) were taken from the Orecchio et al.'s (2011) "a-priori" model estimated for the study area with a method similar to that used by Waldhauser et al. (1998, 2002). Velocities outside the inversion zone of Neri et al. (2009) were fixed to the standard AK135 model (Kennet et al., 1995). The Vp/Vs values outside the well resolved inversion zones were fixed to 1.73.

Concerning the reliability of the shallow part of the velocity structure used in the present study, corresponding to Orecchio et al's (2011) model, we reply as follows:

    (i) the Reviewer's comment on the "a-priori" model by Orecchio et al. (2011) is not correct. As widely explained by Orecchio et al. (2011) the "a-priori" model has been built using several types of velocity data available, not only profiles and tomography but also data from surface wave inversion and Moho depth maps referring also to sectors typically not well resolved by local earthquake tomography. The method used by Orecchio et al. (2011) for the "a-priori" model estimation is very similar to the method followed by Waldhauser et al. (1998, 2002) and may help definition of the velocity structure in sectors where seismic data are not enough for well constraining tomographic inversion. It is obvious that tomographic inversion may improve the velocity structure starting from the "a-priori" model when the data available allow reliable inversion, but this does not reduce the dignity and usefulness of Orecchio et al's (2011) "apriori" model, as outlined in several parts of Orecchio et al's paper, for example at the beginning of the Conclusions "The main result of this study is the P-wave velocity model of the Calabrian Arc lithosphere obtained by integrating all the "a-priori" information available for the study region". Therefore, when the Reviewer looks only at the area contoured by Spread Function in the Figure 6 of Orecchio et al. (2011), he is wrong because the quality of the velocity structure outside may also be suitable for locations.

(ii) The zone of major crustal thinning in the Ionian sea (see, e.g., Grad et al., 2009) is outside (east of) our ray-tracing zone of the WI sector (see the figure FIG-REPLY-n.1 at the end of these Replies). The minor crustal thinning detectable in the WI sector (wrt to Calabria-Sicily) is reproduced by the Orecchio et al's (2011) "a-priori" model.

The zone of crustal thinning existing in the Sicily Channel (see Grad et al., 2009) lies inside the ray-tracing zone of the SC sector (see the figure FIG-REPLY-n.2), but it corresponds to a very mince west-trending band located between Sicily and Tunisia structural domains, where crustal thinning is also relatively small (wrt Sicily and Tunisia) and very far from an oceanic-like crustal structure; there is, of course, some approximation of the "a-priori" model in correspondence with this mince band but this approximation did not appear – according to our experience - capable to preclude a meaningful analysis. In any case, starting from the Reviewer's criticisms we have located again the earthquakes of sector SC using the tomographic model recently published by Scarfì et al (Scientific Reports, 2018) which covers a greater part of our ray-tracing zone wrt our "a-priori" model. Also, we have relocated the earthquakes of sector WI with the Scarfì et al. (2018) model. As explicited in the new text (see Chapter "Changes" lines 158-166), the results we obtained using the Scarfì et al. (2018) model when locating the earthquakes of sectors WI and SC are nearly identical to those obtained with our velocity model (see also the figures FIG-REPLY-n.3 and n.4 at the end of these Replies). In the same locations, very similar estimates of the hypocenter location errors have been obtained using our velocity model and Scarfì et al's one, respectively, in both sectors WI and SC, see the table here:

|  | Neri et al.'s (2012) velocity model | | Scarfì et al.'s (2018) velocity model | |
| --- | --- | --- | --- | --- |
|  | ERH (km) | ERZ (km) | ERH (km) | ERZ (km) |
| Western Ionian | 3.0 | 4.8 | 3.2 | 5.0 |
| Sicily Channel | 4.3 | 9.0 | 4.3 | 8.5 |

The comparative analysis between the results coming from use of (i) our model (Neri et al., 2012) and (ii) Scarfì et al.'s (2017) tomographic model (see the figures FIG-REPLY-n.3 and n.4) furnishes a reasonable indication of the reliability of our hypocentral locations and relative errors, at least in relation to the purposes of the present study. We agree, of course, that velocity model improvements are required in this region.

All the information and tests available indicate that the location error estimates obtained using our velocity model and declared in the original MS are realistic. Also, it is realistic the nearly vertical distribution of seismicity from surface to 70 km depth we have found in SC (this vertical band of hypocenters was used by the Reviewer as a proof of poor location). Among other things, as we clearly declared at lines 396-397 in the original MS, this nearly vertical distribution of seismicity from surface to 70 km depth in that sector was already found by Calò and Parisi (2014) with a tomographic analysis performed in the same area.
* * *
The authors have not found any Comment A – 3 in the review
* * *
*Comment A – 4)*

*4) Even if the reader tries try to forget the previous issues, we still are missing the most critical point: The demonstration of the improved accuracy of new relocations. The authors present no comparison (e.g. initial [catalogue] distribution of events, revised relocations with SIMULPS, focused relocatios with BayLoc in the focus area). What, exactly, did they (and we, as readers) gained from this whole effort? What seismotectonic features are now seen with the relocations, that are not observed e.g. in the catalogue data? The authors do not say, and the reader is left to wonder about the technical data reliability and contribution to this work.*

Reply to Comment A-4)

The catalogue data are inevitably affected by marked approximations as also witnessed by fixed-depth locations of many events. As explained in the new text (see Chapter "Changes" lines 117-121 and new Figure 4), they may furnish here an introductory overall view of the study region and starting data for more accurate locations in the sectors of our greatest interest (SC and WI). Visual comparison of catalogue data (see the figures FIG-REPLY-n.5 and n.6 at the end of these Replies) with the maps and sections obtained by Bayloc (new Figures 5 and 6 in the article) evidences that the features of seismicity space distribution detected in the Bayloc's plots (see Chapter "Changes" lines 293-326, 334-348, 387-401) are less clear in the catalogue data plots. On the other hand, this is a quite obvious situation considering the preliminary character and the goals of a national bulletin constructed with a 1D velocity model and reporting unrealistic location errors estimated with a linearized algorithm. Therefore, we do not furnish graphical details in the revised MS. However, in the new text (see Chapter "Changes" lines 293-326, 334-348, 387-401) we discuss more explicitly what is the contribution that our locations give to the solution of some open geodynamic questions presented in the old Section 3 (original MS), now Section 2 (revised MS).
* * *
*Detailed comment A – 5)*

*5) A minor issue: It is not possible to make quantitative comments on seismicity with this dataset, e.g. lines 284-285 ": : :Also, a clear drop of activity can be noted in the Ionian offshore of Southern Calabria (Figs 4c and 5): : :". The authors have not studied or presented the catalogue completeness, or its spatial variability. Therefore, seismicity changes may be apparent e.g. poorer detection and location of small events, as we move away from the Sicily coast, due to network detectability. First, they need to demonstrate the catalogue space-time completeness, then show only data that are within this space-time completeness intervals, and only then can they make quantitative assessments of seismicity.*

Reply to Comment A-5)

The significance of the drastic drop of seismicity detected from south to north in the immediate Ionian offshore of Calabria is supported by the knowledge existing concerning the completeness magnitude of the seismic catalog (Schorlemmer et al., 2003, 2010). The completeness magnitude does not exceed 2.5 for depths less than 100 km in the area in argument, while it reaches values as high as 2.8 at deeper depths in the slab zone (see, e.g., Schorlemmer et al., 2003). A value of 2.0-2.5 has also been estimated in a more recent work at a depth of 30 km in the same area, probability level 0.99 (Schorlemmer et al., 2010). See Chapter "Changes" lines 371-372.
* * *
Comment B - 1)

*1) The authors use the famous GF method, and some rather old criteria for stress homogeneity, from >20 years ago. While the work of Wyss et al. (1992) and Gillard et al. (1996) is reliable, the GF method has received a lot of criticism, and several modifications (e.g. Michael, 1987, Lund & Slunga, 1999, Vavrycuk, 2014, 2015, Karagianni et al., 2016) have been proposed that overcome the GF problems, either by adopting a different solution stability check, or by using different criteria such as the instability criterion adopted initially by Lund and Slunga (1999). Why did the authors not compare their results with those obtained by other, more recent methods, in order to verify the robustness of the recovered stress field?*

Reply to Comment B – 1)

The authors of the present study have used several stress inversion methods in the last decades and have widely compared them. The choice of using the Gephart and Forsyth (1984) method in the present study comes from this long experience of studies and tests. Our experience leads us to say that the assumption of "more recent method" = "more efficient method" may be uncorrect in the study case.
However, we start using the experience of other workers in recent papers to reply this Reviewer's Comment.

Maury et al. (Boll. Soc. Geol. France, 2013), in their paper reporting a review of stress inversion methods, wrote, on page 324, "We note that only Gephart and Forsyth consider the Coulomb rupture criterion in its completeness: they make no a priori hypothesis on the friction coefficient on the pore pressure value, or on the cohesion of the slipping surface".
After several analyses, on page 331, Maury et al. also wrote "Based on the differences in the physical hypothesis implicit with each method, we prefer the results obtained with Gephart and Forsyth's method."
Again, on page 333, the same authors wrote "From a purely physical point of view, the Gephart and Forsyth method is that which implies the smallest number of prerequisite hypotheses, namely parallelism of slip vector with resolved shear stress, homogeneous stress field in the domain under consideration and independence of focal mechanisms. However none of the three methods consider the effects of heterogeneity and fault interaction. We have introduced two criteria that insure that these prerequisite are satisfied and then applied the inversion methods to this new dataset. Only Gephart and Forsyth's solution satisfies the complete statistical requirement."

In their paper entitled "Reviewing the active stress field in Central Asia by using a modified stress tensor approach", Karagianni et al. (J. Seismol, 2015) on page 6/25 wrote (please consider that FMSI is Gephart and Forsyth's algorithm):
"Using the Fisher distribution, FD-BSM selects the best over a set of possible stress models produced by FMSI, considering a Fisher distribution for each one of the three principal stress axes, using the 95 % confidence areas of the FMSI solutions. The idea is that among the large number of possible stress models estimated by FMSI, the bestmodel is the one where all three stress axes are closer to the corresponding Fisher mean directions. In this case, the minimum misfit is not considered as the single measure for the selection of the final stress model. More specifically, for every data group, the entire set of possible stress models calculated by FMSI (for a specific confidence level) exhibits a certain distribution of the three main stress axes. These models are also the input data for FD-BSM."
As the reader may understand, Karagianni et al. (2015) start just from Gephart and Forsyth's (1984) inversion results and only apply the Fisher distribution for each one of the three principal stress axes (that is not properly a revolutionary approach).

Now, we also report here one of our tests performed on stress inversion methods. We have compared GF84 with Vavricuk's (2014) method.

We have prepared a set of synthetic focal mechanisms. This set includes 40 focal mechanisms (i.e. nearly the number of real focal mechanisms available in sector E in the Figure 8 of our paper). The 40 mechanisms are 20 SS, 5 TF, 15 NF (see Figure FIG-REPLY n. 7). SS and TF focal mechanisms may be produced by compressional stress (plate convergence for example) acting on differently oriented planes, NF ones may be produced by an extensional stress also present somewhere in the study volume (a local rifting process for example). After perturbation by random errors these focal mechanisms were used for stress inversion by Gephart and Forsyth's (1984) and Vavricuk's (2014) methods, respectively. The results by GF84 and Vavricuk (2014) are displayed in the Figure FIG-REPLY n.7.

GF84 produces a solution showing clearly the combination of the compressional and extensional stresses present in the study volume, with the sigma1 confidence limit covering orientations from horizontal to vertical.

Diversely from GF84, Vavricuk's method produces a solution where the compressional stress compatible with 25 focal mechanisms (out of 40) appears to be fully dominant and the significant degree of stress heterogeneity in the dataset due to 15 NF events does not emerge as it would be correct.

In conclusion, the recent literature clearly shows that (i) the Gephart and Forsyth (1984) method can give better results than other more recent stress inversion methods in several situations and (ii) some very recent works start just from the GF84 inversion results and then introduce minor strategies based on Fisher's distribution to select a part of the stress tensors lying inside the GF84 confidence limits. The recent literature clearly evidences that Gephart and Forsyth's (1984) method is still central to the stress inversion practice. Additional considerations will be reported in the reply to Comment B – 4.

In the new text, starting from comments made by both Reviewers 1 and 2 on the use of GF84, and according to our Dec2018 reply to Reviewer 1, we have added the following part in the section of Methods (see lines 206-213 in the Chapter "Changes"): "The advantage of using Gephart and Forsyth's (1984) method instead of other more recent stress inversion methods (see, e.g., Arnold and Townend, 2007; Vavricuk, 2014; Karagianni, 2015) is that GF1984 is more conservative than newer ones concerning the relative orientation of seismogenic stress and seismic dislocation surface. Caution is appropriate in the present study because we do not make any assumption here concerning the date of formation of the faults activated during the period of investigation."

Comment B - 2)

*2) The zonation procedure and the presentation-discussion of the recovered stress field is completely obscure and definitively was performed in a different way than the*

*approach described by the authors. The authors state (lines 300-311) that they subdivided the FPS in tens subcatalogues, using their spatial characteristics, and made sure to have at least 20 FPS in each group. They then computed the stress and used the F value to assign stress homogeneity, based on some previously proposed cut-off values. The F-values were used to decide on spatial clustering. However, it is completely unclear how this was done. How do you separate FPS in a semi-automatic manner, and finally end up proposing areas separated by a straight line (Figures 8b and 8c)? How was this separation performed and what is the significance of these lines? Are we to simply trust the authors that these boundaries somehow magically appeared in their systematic search? The authors do not show any other results (e.g. in an electronic appendix) but simply state that they decided to ": : :report: : :results that we consider more meaningful: : :". Meaningful in what sense? According to which criteria, or even subjective evaluation? For example, how was the RZ area actually defined? A minor issue: How was the earthquake magnitude used in the FPS grouping, as the authors state in line 302? To the best of my knowledge, the earthquake magnitude can be taken into account as a weight, e.g. in the inversion process, but not in the sub-area separation. Please clarify.*

Reply to Comment B – 2)

OK, as also anticipated in our replies to Reviewer 1, we introduce an electronic appendix in the Revised MS (Supplementary Material, see Chapter "Changes" lines 453-497 and related Figures S1-S15) in order to show other stress inversion results obtained during our analysis and to better clarify the procedure followed to draw the conclusions reported in the main body of the article. In this connection, we have also made a change in the main text of the article, see lines 218-224 in the Chapter "Changes": "Stress inversion in this area has been performed, in particular, by partitioning the whole dataset of focal mechanisms according to earthquake locations and magnitude, with the purpose of detecting space variations of stress and inferring the scale of geodynamic processes causing the observed stresses. The stress inversion results summarizing the main findings of this analysis are reported in Table 1 and Fig. 8, other results useful for a wider view of the computational process and the overall coherence of results are shown in the Supplementary Material." In the above lines we also cite the decision of partitioning the FM dataset according to earthquake magnitude that is a very usual procedure in the literature originally introduced by Rebai et al. (Geophys. J. Int., 1992) and more recently adopted by other workers.

We have worked with different procedures of seismogenic stress inversion and zonation (see, e.g., Caccamo et al., Geophys. J. Int., 1996; Neri et al., Phys. Earth Planet. Int., 2005; Totaro et al., Geophys. Res. Letters, 2016) and also deeply evaluated other semiautomatic methods available in the literature (e.g., Hardebeck and Michael, Jour. Geophys. Res., 2006; Yang and Hauksson, Geophys. J. Int., 2013; Martínez-Garzón et al., Jour. Geophys. Res., 2016). Most of procedures of seismogenic stress zonation present a subjective component of evaluation and show different advantages and disadvantages in different situations (tectonic features of the study area, number of focal mechanisms available, seismicity space-time-magnitude distribution, stress heterogeneity level, etc.). Our expertise on these methods allowed us to prefer the zonation method adopted in the present study for analysis of Sicily and offshores, but we are ready to test other methods if this can be of interest for someone.

Comment B - 3)

*3) The results are really not presented in a clear manner. First, it is clear that the only consistent stress axis is the _3, horizontal extension (see last column of Table 1) and that the separation that they perform leads to a _1 instability for the East part (E in Fig8b, and RZ in Fig. 8c). In my opinion, this simply means that they are mixing FPS belonging to very different regimes. For example, Fig.6a shows that the depth distribution in WI is complicated. Have the authors separated seismicity with depth, or simply included all FPS from shallow crustal depths (0-20km) to upper mantle (50-70km) in the same stress inversion? Is this realistic? In a subduction area you often have transpresional deep events along the slab, with normal events at shallow depths (back-arc area); grouping events from all depths, upper crust to sub-Moho mantle, will certainly give you a very heterogeneous (and probably wrong) stress field. The stress recovered for RZ and E are meaningless. The 2-sigma area for _1 and _2 is practically a vertical plane (also identified by the authors), suggesting that compression changes and that the dataset can not recover the changes, most probably due to the adopted spatial grouping.*

Reply to Comment B – 3)

The test commented in our Reply to Comment B–1 may help the Reviewer to understand our interpretation of the stress results obtained in sector E and RZ of Figure 8 of the original MS.

As said in the Reply to Comment B–1 we have prepared a set of synthetic focal mechanisms. It is convenient for clarity to repeat a part of the text of the previous Reply. The synthetic set includes 40 focal mechanisms (20 SS, 5 TF, 15 NF), see FIG-REPLY n.7 at the end of these Replies. SS and TF focal mechanisms may be produced by compressional stress (plate convergence for example) acting on differently oriented planes. NF focal mechanisms may be produced by an extensional stress also present somewhere in the study volume (a local rifting process, for example). After perturbation by random errors these focal mechanisms were used for stress inversion by the Gephart and Forsyth (1984) method. The stress results coming from inversion of synthetic focal mechanisms (i) look very similar to those obtained from real data in sectors E and RZ and (ii) indicate that our interpretation of the stress results obtained in sectors E and RZ (Figure 8 in the paper) is highly reasonable. We need to remark that this interpretation is just a simple inference, given that GF84 produces a solution showing clearly the combination of the compressional and extensional stresses, with the belt-like sigma1 confidence limit covering orientations from horizontal to vertical.

The Supplementary material should allow better understanding of the adopted spatial grouping also in relation to depth.

Comment B - 4)

*4) Even without considering the previous problems, no discussion and visual comparison is made with previous results. Authors from the same group (Totaro et al., 2016) present results for the WI and part of the SC areas (see their Fig. 3). The stress recovered for area 6 of Totaro et al. (2016) is nearly identical to the one shown by the authors in Fig.8b for the East area (E). Visual comparison of the FPS datasets also confirms that the majority of FPS is identical, which explains the similar results. Therefore, what additional information did the additional FPS bring into the final results? What is the new information on the stress field of the WI area? For the West (W in Fig 8) area, the authors have many more new FPS, but the obtained result for this area are very similar to the results obtained for the (smaller) area 7 by Totaro et al. (2016). Therefore, what is the new knowledge that we gain by the authors (very unclear) processing, that is used by the authors for their model and that was not present and discussed by themselves in Totaro et al. (2016), a few years ago?*

Reply to Comment B – 4)

In the previous paper (Totaro et al., 2016) we have used the stress inversion method by Arnold and Townend (2007). Some tests allowed us to evidence that the AT2007 method (as others proposed in the recent literature, see Reply to Comment B-1) are less conservative than GF84 due to stronger basic assumptions. Then, we have used GF84 in the present study with the motivation explained above and also reported at lines 206-213 in the Chapter "Changes":

"The advantage of using Gephart and Forsyth's (1984) method instead of other more recent stress inversion methods (see, e.g., Arnold and Townend, 2007; Vavricuk, 2014; Karagianni, 2015) is that GF1984 is more conservative than newer ones concerning the relative orientation of seismogenic stress and seismic dislocation surface. Caution is appropriate in the present study because we do not make any assumption here concerning the date of formation of the faults activated during the period of investigation."

An added value of this paper wrt the previous one is that we better control here the heterogeneity of stress in the focal mechanism subsets, in particular we better detect in sector E the extensional component due to rifting into the compressional regime due to plate convergence. This improvement is fundamental for the conclusions of the present study. At the same time, the GF84 approach better guarantees the stress homogeneity detected in the W sector, with other important implications for geodynamic modeling of the study region. We have better clarified these aspects at lines 376-385 in the Chapter "Changes".

Comment B - 5)

*5) Minor issue: The authors state that the stress in the west (W) is more consistent with the average from all data (ALL, Fig 8a). However, it is the East cluster (E) stress that is identical to the overall average (see Table 1). In fact, stress for W presents a _10 clockwise rotation with respect to ALL and E, which should be considered as significant, if the FPS errors are of the order of 10 degrees, as the authors state. In other words, the numbers tell a slightly different story, that the one presented by the authors.*

Reply to Comment B – 5)

The phrase "The best model of stress in this western sector is similar to that obtained by inversion of the mechanisms of the whole study area" reports the word "similar", not "equal". We do not understand the criticism.
In any case, the phrase is clearly not decisive in the frame of the work, the discussion works also without it (see lines 264-278 in the Chapter "Changes").

*Comment post B-5)*

*I could provide additional comments on the Discussion section. Perhaps an important one is that, after an extensive introduction, spending nearly one page (p. 13) to present and discuss features already known (as the authors themselves state), seems like a waste of time. However, specific comments are simply meaningless, since the input information (seismicity relocations and stress inversion) are not reliable.*

Reply to comment post B-5)

We have shortened the extensive introduction mentioned by the Reviewer reducing it to a few lines, see Chapter "Changes" lines 234-237.

**Graphical support for Replies (FIG-REPLY n.1)**

[Figure]

[Figure]

**Graphical support for Replies (FIG-REPLY n.2)**

[Figure]

[Figure]

**Graphical support for Replies (FIG-REPLY n.3)**

[Figure]

**Graphical support for Replies (FIG-REPLY n.4)**

[Figure]

*Velocity model by Neri et al. (2012)*

*Velocity model by Scarfì et al. (2018)*

**Graphical support for Replies (FIG-REPLY n. 5)**

[Figure]

**Graphical support for Replies (FIG-REPLY n.6)**

[Figure]

**Graphical support for Replies (FIG-REPLY n. 7)**

[Figure]

**GF84          ## Vavricuk 2014**

[Figure]

**Chapter "Changes in the MS"**

(Line numbers in this Chapter help finding connection between the above reported Replies and the text below)

**In the Section 1 (Introduction) we have deleted the last period "The**
**potentialities of the methods ….. may be reached", under Reviewer's request.**
-------------
**The Section 2 of the revised MS (Geodynamic frame of the study region)**
**includes both Sections 2 and 3 (shortened) of the original MS. The main change**
**(shortening of the description of past geodynamic models requested by the**
**Reviewer) is highlighted in yellow. The new Figure 3 includes only the sections f,**
**g, h and i of the old Figure 3.**
In the Mediterranean region (Fig. 1), after a long period between Late Paleogene and Neogene of
Africa NW-ward subduction beneath Eurasia, subduction has almost ceased (Billi et al., 2011).
With the progression of Africa-Eurasia convergence, a tectonic reorganization of the plate boundary
has started to accomodate contraction, in particular contractional deformation in several segments
of the boundary (such as Sicily) has shifted from the former subduction zone to the margins of the
back-arc oceanic basins (Fig. 2). The tectonic reorganization of the boundary is still strongly
controlled by the inherited tectonic fabric and rheological attributes, which are strongly
heterogeneous along the boundary.
In Italy (Fig. 1) the Neogene convergence and associated subduction between Africa and Eurasia
resulted in the NW-trending Apennine and the W-trending Maghrebian fold-thrust belts in
peninsular Italy and Sicily, respectively (Malinverno and Ryan, 1986). The two belts are connected
through the Calabrian Arc (Minelli and Faccenna, 2010), below which a narrow remnant of the
former subducting slab seems to be still active, but close to cessation (Neri et al., 2009; 2012;
Orecchio et al., 2015; Chiarabba and Palano, 2017).
Contraction in Sicily is, at present, mainly accomodated at the rear of the fold-thrust belt, in the
southern Tyrrhenian sea (Fig. 2), where a series of contractional earthquakes located in an E-
trending belt between Ustica and Eolian islands have been recorded in the last decades (Pondrelli et
al., 2004; Billi et al., 2007; Presti et al., 2013; Orecchio et al., 2017). Toward the east, the
compressional seismic belt is delimited by the seismically active Tindari Fault System (TFS in Fig.
2), to the east of which, both earthquakes and GPS data provide evidence for an ongoing
extensional tectonics possibly connected with the residual subduction beneath the Calabrian Arc
and related back-arc extension (Palano et al., 2015b).
The plate margins strongly simplified in Fig. 1 have undergone significant changes over time as a
consequence of the above processes and of the whole geodynamic activity occurring in the
Mediterranean region. The investigations performed thanks to the increasing set of geophysical data
available have brought the researchers to propose different scenarios of microplate fragmentation
and reorganization along the boundary, in particular along the Italian portion of the boundary. The former studies suggested the existence of an independent microplate roughly corresponding to the Adriatic sea (Adria; Anderson and Jackson, 1987) that more recent investigators proposed to consist of two separate blocks (North and South Adria) confining in the central Adriatic sea, approximately (Battaglia et al., 2004). In the last ten years, the hypothesis of a South Adria block or microplate having its northern boundary in the central Adriatic sea has been shared by other authors (see, e.g., D'Agostino et al., 2008; Perouse et al., 2012) who, however, expanded this microplate to south and west with different geometries. D'Agostino et al. (2008) interpreted their GPS and earthquake slip data by distinguishing a northern Adriatic microplate (Adria) and a microplate to south including the Apulia promontory, the Ionian sea and the Hyblean region in southern Sicily (Fig. 3a). According to these authors, the hypothesis of Hyblean region belonging to such a microplate would be supported by apparently low GPS-derived deformation in the western Ionian. In any case the authors admit that lack of data in the Ionian offshore of Sicily does not allow decisive checking of their hypothesis. Based on the analysis of GPS velocities and earthquake focal mechanisms in the Central and Eastern Mediterranean, Pérouse et al. (2012) proposed that the area including the south Adriatic sea, the Apulian peninsula, the Ionian basin, the Hyblean Plateau and the Sirte plain may be considered as a single rigid block rotating clockwise wrt Africa, inducing an opening of a couple of mm/yr in the Sicily Channel (Fig. 3b).

By analysis of a relatively long period of 18 years of GPS observations, Palano et al. (2012) advanced the hypothesis of an Hyblean block independent with respect to Africa and Apulia (Fig. 3c-d). They located the regional contraction existing between the Tyrrenian and Hyblean blocks in two distinct belts identified in the northern Sicily offshore (Ustica-Eolie) and across the Sicily front (Sicilian basal thrust according to Lavecchia et al., 2007). The authors discussed also the role played by the Ionian domain and suggested two possible scenarios, one assuming that the Ionian is rigidly connected with the Hyblean block (Fig. 3c), the other assuming that the Ionian domain diverges from the Hyblean block and moves to northeast wrt to Eurasia (Fig. 3d). They concluded that the lack of islands (i.e. of data) in the Ionian offshore does not allow to make a choice among these scenarios. This work with its uncertainties confirms and further supports the doubts advanced in the early work of Serpelloni et al. (2007) who imputed to the lack of GPS data and poor seismic network geometry in offshore sectors like the Ionian basin the difficulty of detecting microplate contours in this region and left open the question "where strain locates between Hyblean and Apulia domains?".

In his review of papers and investigations regarding the crustal kinematics in the Mediterranean region, Nocquet (2012) drew the conclusion that it is quite difficult to state from the available data and analyses where stable Africa finishes and other eventual blocks like Apulia begin in the Central Mediterranean area at the longitude of Italy. In a very recent analysis performed by integration of multibeam, seismic reflection, magnetic and gravity data, Polonia et al. (2017) have concluded that: (i) tearing at the southwestern edge of the SEward-retreating Calabria subduction slab may be the deep source of shallow deformation detected in correspondence of the transtensional Ionian Fault in the Ionian basin (Fig. 2); (ii) the NW-SE trending belt comprised between the Ionian Fault and the Alfeo-Etna Fault (Fig. 2) hosts a rifting zone opening SW-NE where serpentinite diapirs have been identified by the authors. On their hand, Gutscher et al. (2017), by analysis of multi-beam bathymetric data and seismic profiles, proposed the Alfeo-Etna right-lateral fault system as shallow tectonic expression of the retreating slab tear or STEP fault. In the view of these authors some sinistral lateral component appearing in the southeasternmost part of the Ionian Fault should exclude the latter as potential expression of the STEP fault. Another major fault system marking the transition between the Ionian basin and the Hyblean plateau, e.g. the Malta escarpment (Fig. 2), shows not to be currently active along most of its lenght and shows signs of recent faulting only in its northernmost segment (see, e.g., Argnani, 2009; Gutscher et al., 2016). West of the Malta escarpment (Fig. 2), a detailed analysis of reflection profiles and stratigraphic and structural data allowed Cavallaro et al. (2016) to evidence clear time evolution from tensional to compressive regimes in the Sicily Channel WNW-trending main structural system, until its present-day behaviour as transcurrent fault system under compression due to NW-SE Africa-Eurasia convergence. Cavallaro et al. (2016) proposed, in particular, that local volcanism stopped in late Miocene and Miocene-time normal faults were reactivated during Zanclean-Piacenzian age as right-lateral strike-slip faults.
* * *
**The new Section 3 (Data, methods of analysis and results) has been significantly changed wrt the original MS (old Section 4) in order to furnish more details concerning the velocity model, the reliability of hypocenter locations, the reason for choice of Gephart and Forsyth's (1984) method, the reference to a section of Supplementary Material.**

The Figure 4 shows the earthquakes of local magnitude > 2.5 occurring between 1981 and 2016 at depths less than 100 km in the area bounded by the dashed rectangle of section 'a', according to data reported in the Italian national seismic catalog (http://istituto.ingv.it/l-ingv/archivi-e-banche-dati/). The sections 'b' to 'd' of this figure display the earthquake epicenter maps corresponding to different hypocentral depth ranges. Hypocenter locations of the catalog are estimated with a 1-D velocity structure of all Italy and result to be generally quite approximate as also witnessed by fixed-depth locations and layer-boundary sticking of many events: they may give here an introductory overall view of the study region and starting data for more accurate locations to be performed in the sectors of our greatest interest in the present study.

We have, then, focused our attention on the sectors WI and SC indicated in Fig. 4a because, as explained in the previous Sections, the geophysical knowledge in these offshore sectors is still relatively poor and their exploration can be decisive for answering several open geodynamic questions in the region. We have relocated the hypocenters of the earthquakes of sectors WI and SC of Fig. 4, after integration of the P- and S-wave readings of the Italian national catalog with those available from the databases of the local seismic networks operating in Sicily and Calabria between 1981 and 2001 (Orecchio et al., 2011). For hypocenter relocations, we have selected the events of local magnitude Ml > 2.5 for which a minimum of 15 P+S arrival times were available, and used the Bayesian non-linear location algorithm named Bayloc (Presti et al., 2004 and 2008). As well known from the literature (Lomax et al. 1998, 2000; Lomax and Michelini, 2001; Husen and Smith, 2004; Presti et al. 2004, 2008; Lippitsch et al., 2005; among others), the non-linear probabilistic location methods furnish more accurate estimates of hypocenter locations and relative errors compared to linearized methods when the recording network geometry is not optimal: this is clearly the situation of our offshore sectors WI and SC (Figure 4). Starting from seismic phase arrival times at the recording stations, Bayloc computes for an individual earthquake a probability cloud marking the hypocenter location uncertainty. Then, Bayloc estimates the spatial distribution of probability relative to a set of earthquakes by summing the probability densities of the individual events. This method has been shown to help detection of seismogenic structures through better hypocenter location and more accurate estimation of location errors compared to linearized methods (see Presti et al., 2008, for details).

Bayloc's locations have been performed in the 3D velocity structure proposed for the study region by Neri et al. (2012) who, basically, integrated the velocity structures proposed by Orecchio et al. (2011) and Neri et al. (2009) for the depth ranges 0-40 km and 65-260 km, respectively. Both the shallow and intermediate-depth velocity structures were the result of P- and S-wave tomographic
inversions, with velocity values in the nodes external to the well resolved inversion zones assigned
as follows. Velocity values outside the inversion zone of Orecchio et al. (2011) were taken from the
Orecchio et al.'s (2011) "a-priori" model estimated for the study area with a method similar to that
used by Waldhauser et al. (1998, 2002). Velocities outside the inversion zone of Neri et al. (2009)
were fixed to the standard AK135 model (Kennett et al., 1995). The Vp/Vs values outside the well
resolved inversion zones were fixed to 1.73.
The epicenter maps and hypocenter vertical sections obtained by Bayloc using the above described
velocity model are shown in the Figs 5 and 6. Epicenter and hypocentre errors of the order of 3 km
and 5 km have been estimated for the earthquakes of Sector WI, rising to values of 4 km and 9 km
in Sector SC. Because ray tracing to recording stations of our earthquake sets of sectors WI and SC
tends to cover, in part, zones of Western Ionian and Sicily Channel external to the tomographic
inversion domains (graphs omitted for conciseness), we have performed for comparison Bayloc's
locations of the same events with the most recent seismotomographic model proposed for the same
region (Scarfi et al., 2018). Scarfi et al's model covers nearly all our ray tracing area. The results we
obtained using the Scarfi et al. (2018) model are nearly identical to those obtained with our velocity
model and displayed in the Figures 5 and 6. Also, very similar estimates of the hypocenter location
errors have been obtained using our velocity model and Scarfi et al's one, respectively, in both
sectors WI and SC.
With the purpose of estimating the seismogenic stress tensor orientations in the area of interest of
the present study and confining sectors we have collected from literature and the international
catalogs the focal mechanisms estimated by waveform inversion for the earthquakes occurring in
the area bounded by the dashed rectangle of Figure 4. We have limited our selection to the seismic
events of magnitude over 2.5 occurring in the period 1977-2016 at depths less than 70 km. The map
of these focal mechanisms is shown in Fig. 7, the list of their focal parameters is furnished in Table
A1, Appendix A.
In the dataset of Figure 7 the focal mechanisms computed by the CAP method (Li et al., 2007;
D'Amico et al., 2010, 2011; Presti et al., 2013; Orecchio et al., 2014; Totaro et al., 2016) are
affected by errors of the order of 8-10 degrees (see, among others, Totaro et al., 2016). The
literature and the bibliographic sources of the other focal mechanisms of Fig. 7 indicate that these
are typically characterized by fault parameter errors of the order of 10-15 degrees (see, e.g.,
Hellfrich, 1997; Frohlich and Davies, 1999; Pondrelli et al., 2006; Hjörleifsdóttir and Ekstrom,
2010), then generally smaller than errors of focal mechanisms computed by inversion of P-onset
polarities in areas of critical network geometry like ours (D'Amico et al., 2011; Presti et al., 2013;
Musumeci et al., 2014). The overall level of uncertainty of focal mechanisms of the dataset of Fig. 7
makes it suitable for application of the method by Gephart and Forsyth (1984) and Gephart (1990)
for calculating the seismogenic stress tensor directions in the study region. This method searches for
the stress tensor showing the best agreement with the available focal mechanisms (FMs). Four
stress parameters are calculated: three of them define the orientations of the main stress axes; the
other is a measure of relative stress magnitudes, $R = (\sigma_2-\sigma_1)/(\sigma_3-\sigma_1)$, where $\sigma_1$, $\sigma_2$ and $\sigma_3$ are the
values of the maximum, intermediate and minimum compressive stresses, respectively. In order to
define discrepancies between the stress tensor and observations (FMs), a misfit variable is
introduced: for a given stress model, the misfit of a single focal mechanism is defined as the
minimum rotation about any arbitrary axis that brings one of the nodal planes, and its slip direction
and sense of slip, into an orientation that is consistent with the stress model. Searching through all
orientations in space by a grid technique operating in the whole space of stress parameters, the
minimum sum of the misfits of all FMs available is found. The confidence limits of the solution are
computed by a statistical procedure described in the papers by Parker and Mc Nutt (1980) and
Gephart and Forsyth (1984). The size of the average misfit corresponding to the best stress model provides a guide as to how well the assumption of stress homogeneity is fulfilled (Michael 1987). In the light of results from a series of tests carried out by Wyss et al. (1992) and Gillard et al. (1996) to identify the relationship between FM uncertainties and average misfit in the case of uniform stress, we will assume that the condition of homogeneous stress distribution is fulfilled if the misfit, F, is smaller than 6°, and that it is not fulfilled if F>9°. In the range 6°<F<9°, the solution is considered as acceptable, but may reflect some heterogeneity.

The advantage of using Gephart and Forsyth's (1984) method instead of other more recent stress inversion methods (such as, for example, Arnold and Townend, 2007; Vavricuk, 2014; Karagianni, 2015) is that GF1984 is more conservative than newer ones concerning the relative orientation of seismogenic stress and seismic dislocation surface. Caution is appropriate in the present study because we do not make any assumption here concerning the date of formation of the faults activated during the period of investigation. The effectiveness of this relatively ancient method in several conditions, also compared to more recent methods, is well documented (see Hardebeck and Hauksson, 2001; Maury et al., 2013; Karagianni, 2015; among others).

For the application of Gephart and Forsyth's (1984) method in the present study, we have focused on the area contoured by the thin line in Fig. 7 including southern Sicily, the Sicily Channel and the Western Ionian, e.g. the sectors where the knowledge of seismogenic stress distributions is poorer than elsewhere in the region (see, among others, Totaro et al., 2016). Stress inversion in this area has been performed, in particular, by partitioning the whole dataset of focal mechanisms according to earthquake locations and magnitude, with the purpose of (i) detecting space variations of stress and (ii) inferring the scale of geodynamic processes causing the observed stresses. The stress inversion results summarizing the main findings of this analysis are reported in Table 1 and Fig. 8, other results useful for a wider view of the computational process and the overall coherence and stability of results are shown and commented in the Supplementary Material.

\---------------

**Several changes have been made in the Section "Discussion" (old Section 5, new Section 4) according to Reviewer's requests.**

The epicenter and focal mechanism maps of Figs 4 and 7 evidence the seismicity associated to extensional processes of the Apennine-Maghrebian chain, occurring inside the overall compressional domain of the study region imputable to Africa-Eurasia convergence (Nocquet, 2012; Presti et al., 2013; Totaro et al., 2016; among others). We have discussed in Section 2 the efforts made by many investigators to identify opening zones between diverging microplates in the Sicily Channel and the Ionian sea with the purpose of explaining the space variations of crustal motions measured by GPS networks in the Central Mediterranean region. Lack of data in wide offshore sectors is the main reason why the debate on microplate geometry and kinematics in the region is still open. The results of the present study may, in our opinion, furnish a useful contribution to this debate.

Stress tensor inversion of earthquake fault-plane solutions in the area including Southern Sicily, the Sicily Channel and the Western Ionian sea (Fig. 8a, set ALL) reveals moderate stress heterogeneity (F-value of 8.3°) around a best model of stress characterized by a sub-horizontal, NW-trending $\sigma_1$ clearly reconductible to Africa-Eurasia convergence (Fig. 8a and Table 1). Starting from this result, we have sub-divided the dataset of focal mechanisms of Fig. 8a in tens of subsets according to the epicenter distribution, focal depth and magnitude of the earthquakes, in order to search for subsets satisfying the condition of stress homogeneity. As explained in the previous Section, this condition can be considered reasonably satisfied in the present study when the F-value of inversion is lower than 6°. In order to guarantee the significance of stress computations in the different subsets we have decided to fix a minimum number of 20 earthquakes (= focal mechanisms) for the creation of the individual subset. The stress inversion results obtained after the first step of partitioning of the dataset into different subsets indicated new strategies of partitioning or data grouping. For sake of conciseness, we do not report in the main body of this article the stress inversion results obtained for all the subsets investigated, we only report in Fig. 8 and Table 1 the results able to better outline the stress patterns and tectonic features in the study area. The reader interested in the results obtained for other subsets investigated in the present study, useful for a wider view of the computational process and overall coherence of results, can find them in the Section of Supplementary Material.

Fig. 8b and Table 1 (lines W and E) report the stress inversion results obtained by subdividing the study area in two sectors W and E located, respectively, west and east of a NW-trending separation line indicated as AB in the same Fig. 8b. A F-value of 5.9° shows that stress is homogeneous or close to homogeneity in the W sector (Table 1). The 95% confidence limits of stress orientations in W (Fig. 8b) show an acceptable level of constraint of $\sigma_1$ orientation compatible with NW-SE plate convergence. On the other hand, the confidence area of $\sigma_3$ orientation in the same sector is relatively large and extends from vertical to SW-NE horizontal direction, i.e. $\sigma_3$ and $\sigma_2$ are unconstrained on the SW-NE vertical plane. This is plausibly related to co-existence of reverse and strike-slip seismic faulting in the study volume under SE-NW compression due to plate convergence. These results are in good agreement with the geostructural and geodynamic reconstruction of this area proposed by Cavallaro et al. (2016) who evidenced, in particular, long term evolution from tensional to compressive regimes in the Sicily Channel until the present-day state of compression led by NW-SE Africa-Eurasia convergence. Our stress result does not support the hypothesis of active extension in the rift zone of the Sicily Channel advanced by previous investigators (D'Agostino et al., 2011; Perouse et al., 2012).

The inversion of the fault-plane solutions available in sector E of Fig. 8b leads to a F-value of 8.3° indicating a certain degree of stress heterogeneity as already inferred from the analysis of the whole dataset of Fig. 8a. The best model of stress in sector E (Fig. 8b and Table 1) is quite similar to those of sets ALL and W (Table 1), and this indicates that NW-SE compression from plate convergence is again detectable in this eastern part of the study area. However, the 95% confidence limits of stress orientations in E reveal that $\sigma_1$ orientation is practically unconstrained from horizontal NW-SE to vertical, and this leads us to suppose that some extensional process opening SW-NE acts together with NW-trending plate convergence in this sector. This result does not support the hypothesis of a unique rigid microplate including western Ionian and Hyblean platform advanced by previous investigators (D'Agostino et al., 2011; Perouse et al., 2012; and Palano et al., 2012 see our Fig. 3c).

By comparing the location of the separation line between the E and W domains of Fig. 8b with the structural map of Fig. 2 we can note that the separation line approximately corresponds with the location of the Alfeo-Etna Fault. Again, by comparing the Bayloc's epicenter distribution of the earthquakes shallower than 30 km in Sector WI (Fig. 6c) with the structural map of Fig. 2 we may note (i) a certain degree of activity in correspondence with the northern segment of the Malta escarpment, the only considered active in this structural system by several authors (see, e.g., Argnani, 2009; Gutscher et al., 2016), (ii) a clear belt of activity trending NW-SE between the Alfeo-Etna and the Ionian faults and (iii) a drop of seismicity going to NE across the Ionian Fault. In the 30-70 km depth range (Fig. 6d) there is no seismicity in correspondence with the Malta escarpment and most of seismic activity is dispersed around the Ionian Fault. The SW-NE vertical section of Fig. 6b highlights that sesmicity is as shallow as 20 km west of the Alfeo-Etna Fault but deepens to depths of the order of 40 km between the Alfeo-Etna and the Ionian faults, and remains stable at this depth level for several tens of km around the Ionian Fault. In the same vertical section the presence of sesmic activity can also be noted in the upper 20 km halfway between the Alfeo-Etna and the Ionian faults (AEF and IF).

Previous investigators (see, e.g., Polonia et al., 2011; 2016; Palano et al., 2015b) have proposed that the Ionian fault zone corresponds with the southwestern edge of the Ionian subducting slab dipping to NW and presumably still affected by slow SE-ward residual rollback. The Ionian fault zone would represent the shallow expression of a STEP fault (Polonia et al., 2016). The above commented epicenter and hypocenter distributions of Fig. 6 support this interpretation, evidencing however that the dislocation process between the subducting slab (northeast) and the adjacent lithosphere (southwest) is distributed over a relatively wide zone probably because the subduction and slab rollback kinematics are very slow and do not mimic fast STEP dynamics (Gallais et al., 2013; Orecchio et al., 2014). The earthquake activity detected in the upper 20 km halfway between the Alfeo-Etna and Ionian faults (Fig. 6b) occurs in a NW-trending highly heterogeneous and fractured zone identified by Polonia et al. (2017) as the site of a rifting process with SW-NE opening direction where serpentinite diapirs rise from deeper depths. This extensional process can be a cause of stress heterogeneity evidenced by stress inversion in the E domain of Fig. 8b, where SW-NE extension appears to be superimposed to NW-SE compression related to Africa-Eurasia convergence. Therefore, our earthquake space distribution and stress orientations in the western Ionian are compatible with the geodynamic hypothesis of Polonia et al. (2017) assuming a rifting process near the southwestern edge of the subducting slab. This interpretation appears also to match well with the space variation of GPS vectors in Sicily and Calabria (Fig. 9; vector data from Palano et al., 2012) showing clear rotation of crustal motions when crossing the onshore prolongation of the NW-trending zone between the Alfeo-Etna and Ionian faults. Having more than 20 focal mechanisms in the NW-trending zone where Polonia et al. (2017) detected the rifting process, we have performed a stress inversion run in this zone (RZ in Fig. 8c). The results (shown in the same Fig. 8c and in Table 1) are similar to those obtained in the E domain of Fig. 8b and confirm our hypothesis of combination of NW-SE compression due to plate convergence and SW-NE rifting. Other information on stress inversion in different partitions of the available dataset, highlighting the coherence and stability of results, is reported in the Section of Supplementary material.

The Bayloc's maps of epicenters in Sector SC (Fig. 6) show that seismicity is mainly located in correspondence with a ca. NNE-SSW trending fault system crossing the WNW-ESE Pelagian rift area. The depths of these events cover all the range of investigation of our analysis, i.e. 0-70 km. The seismic activity of this NNE-SSW trending fault system covering this relatively large depth range has already been detected by previous investigators (Calò and Parisi, 2014). Our results show also that some activity occurs in the Hyblean corner of Sicily and south of it, but in this case the seismicity is shallow and never exceeds the depth of 30 km. Only minor activity is located in correspondence with the fault system of the Pelagian rift. These results from Bayloc's hypocenter locations, together with the nearly homogeneous stress tensor marking SE-NW plate convergence in sector W of Fig. 8b, do not support active rifting in the Pelagian rift zone of the Sicily Channel. In other words, our results of minor seismicity in the Pelagian rift zone and homogeneous compressional stress in the whole SC area do not support the hypothesis of microplate separation and divergence in the Pelagian rift zone of the Sicily Channel, hypothesis advanced in previous works based on analysis of poorly distributed geodetic data and microplate rigidity assumptions (see, e.g., D'Agostino et al., 2008; Perouse et al., 2012). On the other hand, our results match well with those obtained in the same area by Cavallaro et al. (2016) through detailed analysis of reflection profiles and stratigraphic and structural data. These authors evidenced clear time evolution from tensional to compressive regimes in the Sicily Channel WNW-trending main structural system, until its present-day behaviour as transcurrent fault system under compression due to NW-SE Africa-Eurasia convergence. According to the same authors, Miocene-time extensional structures of the Sicily Channel WNW-trending structural system were reactivated during Zanclean-Piacenzian age as right-lateral strike-slip faults under convergence-related compression.

If we enlarge the frame of analysis to include the southern Tyrrhenian region (Figs. 7 and 9) we find additional signature of the SE-NW Africa-Eurasia convergence in the southern Tyrrhenian east-trending compressional belt located offshore northern Sicily (see also Presti et al., 2013 and Totaro et al., 2016 for stress orientations in the southern Tyrrhenian seismic belt). On this scale of analysis our results match well with the very recent Nijholt et al's (2018) conclusion according to which in this south-central part of the Mediterranean region "the Calabrian arc is now further transitioning towards a setting dominated by Africa-Eurasia plate convergence, whereas during the past 30 Myrs slab retreat continually was the dominant factor". Also, our investigation leads us to evidence that: (i) perpendicular-to-convergence opening in the Ionian sea (between the Alfeo-Etna and Ionian faults) introduces some seismogenic stress heterogeneity in the eastern compartment of our stress inversion area (Figs 8b, 8c and 9); (ii) residual, very slow subduction and SE-ward rollback of the Ionian lithosphere northeast of the Ionian Fault reduces convergence-related compression in the shallow Ionian offshore of southern Calabria and causes there a drastic drop of seismicity indicated by the arrow in Fig. 4c (the completeness magnitude in this sector was estimated between 2.0 and 2.5 by Schorlemmer et al., 2003 and 2010). This process of reduced compression can also influence stress parameters in this sector with particular reference to the Ionian offshore of Calabria, but with the data available we cannot analyze this aspect in a greater detail. We look at a future availability of additional data in the eastern sector of Fig. 8b to explore in a greater detail the local space variations of stress and the contributions by the different tectonic factors. On the other hand, we evidence the new contribution to knowledge of regional geodynamic processes given by the present study in comparison to the most recent stress analyses carried out in the same area (Totaro et al., 2016). In fact, although our results confirm the primary seismogenic role attributed to plate convergence by Totaro et al. (2016), the use of a slightly larger dataset and of the more conservative stress inversion method by Gephart and Forsyth (1984) allow us to better control in the present study the heterogeneity of stress in the focal mechanism subsets. In particular, we detect in sector E the rifting-related extensional component into the plate convergence domain and better guarantee stress homogeneity in the W sector. Both improvements are relevant for the conclusions of the present study.

Our results may contribute to answer some questions put by previous investigators such as, for example, Serpelloni et al. (2007) (where strain locates between Hyblean and Apulia domains?) or Palano et al. (2012) (is Ionian rigidly connected with the Hyblean block or diverging from it and moving to northeast wrt to Europe?). Our earthquake locations and stress distributions allow us to state that (i) the Western Ionian is a main site of strain release between Hyblean and Apulia domains and (ii) the Ionian is not "rigidly connected with the Hyblean block". Following the reasoning of Palano et al. (2012) who concluded with the proposal of two alternative scenarios reported in our Fig. 3c-d, our results bring us to favour the hypothesis of a Ionian block diverging from the Sicilian-Hyblean-Malta block (3d), with the boundary between the respective blocks located, however, as in Fig. 3c, approximately. Our results lead us to propose that the rifting process suggested by Polonia et al. (2017) between Alfeo-Etna and Ionian faults in the Ionian basin should be taken in consideration in the future modeling of microplate geometry and kinematics. Concerning the doubts of D'Agostino et al. (2008) whether their assumption of a rigid Apulian-Ionian-Hyblean microplate is correct or not, our results in the Western Ionian showing evidence of a rifting process as proposed by Polonia et al. (2017) suggest that it is not.

We strongly feel that the usual assumption of rigid blocks or microplates in the physical modeling of geodynamic processes should be overcome and different crustal rheologies should be tested when modeling tectonic deformation and microplate relative motions in the Central Mediterranean region. For this purpose, we have recently started a research collaboration with the geophysical team of University of Milan for Finite Element Modeling of tectonic stress and strain distributions
through high-resolution 3D thermo-rheological representation of lithosphere in the region of our
interest. This line of research is grounded on methodological developments described in the papers
by Splendore and Marotta (2013) and Marotta et al. (2015), among others.

**List of Figures and Tables of the revised MS**

**Fig. 1 (confirmed)**
**Fig. 2 (confirmed)**
**Fig. 3 (modified) – reported in the following**
**Fig. 4 Catalogue data – reported in the following**
**New Fig. 5 = old Fig. 6**
**New Fig. 6 = old Fig. 7**
**New Fig. 7 = old Fig. 5**
**Fig. 8 (confirmed)**
**Fig. 9 (confirmed)**
**Table 1 (confirmed)**
**Appendix (confirmed)**
**Supplementary material – reported in the following**

[Figure]

Figure 3

[Figure]

Figure 4

[Figure]

New Figure 5 - Old Figure 6

[Figure]

New Figure 6 - Old Figure 7

[Figure]

New Figure 7 - Old Figure 5

**SUPPLEMENTARY MATERIAL**

We report here a selection of stress inversion runs performed in the present study. The analysis
made on different partitions of the focal mechanism dataset brought us to draw the conclusions on
stress distribution in the study area reported in the main body of the article. In particular, we have
concluded that seismogenic stress tensor orientations in and around Sicily mark the dominant action
of Africa-Eurasia NW-oriented plate convergence in this part of the Mediterranean region.
Evidences of other tectonic factors superimposed to plate convergence have been found in the
Western Ionian sea: here, a rifting process with opening direction perpendicular to convergence at
the southwestern edge of the subducting slab adds extensional stress to convergence-related
compression. In addition, we concluded in the main body of the article that presumably slow trench
retreat offshore southern Calabria may reduce convergence-related compression and introduce
additional stress heterogeneity. Finally, no seismic evidence of active rifting or microplate
separation and divergence is detected in the Pelagian area of the Sicily Channel, where clear
signatures of plate convergence are found.

Partitioning of whole FM dataset according to depth (S1 and S2) shows a certain degree of stress
heterogeneity in both depth ranges investigated (0-20 km and 20-70 km, respectively). These results
suggest extensional dynamics possibly due to rifting superimponed to primary compressional
dynamics related to plate convergence. Different depth-partitioning (S3, 0-30 km; S4, 30-70 km)
shows that homogeneous compressional stress dominates at depths deeper than 30 km, while
extensional dynamics (always superimposed to primary compressional ones) are confined to the
upper 30 km. Stress heterogeneity with rifting-related extensional dynamics superimposed to
compressional dynamics is confirmed when stress inversion is performed on the whole 0-70 km
depth range in the Ionian Rifting Zone (S5). When the inversion is performed in the upper 30 km of
the whole western Ionian area (S6) the relative contribution of the extensional dynamics in the
dataset increases and, then, the level of stress heterogeneity becomes really large (F = 9.9°).
Partitioning according to magnitude (S7, magnitude >=4.0; S8, magnitude <4) shows that the
stronger events are slightly less heterogeneous than weaker ones. Perhaps, this difference may be
due to greater presence of rifting-related extensional events in the subset of weaker events. The
partitions made by a north-trending separation line (S9, S10, S11) show clear stress heterogeneity of
the eastern domains (already commented above), while the western domains in the same figures
tend to homogeneity when eastern Sicily is excluded from the subset (even though in the latter case
the low number of data available for inversion does not allow good constraint of the stress solution).
This result is again easy to explain in terms of greater (to east) or lesser (to west) contribution by
rifting-related extensional dynamics in the framework of primary compressional dynamics due to
plate convergence. The partitions made by a northwest-trending separation line (S12, S13) show
again stress heterogeneity in the eastern domains and decreasing heterogeneity of the western ones
when the Ionian area northeast of the Alfeo-Etna line (S12) is excluded from stress computation.
Other results are shown in the Figures S14 and S15. Figure S14 reports the stress tensor in the
whole study area with the exclusion of the depth range 0-30 km of the rifting zone (RZ), Fig. S15
shows the stress solution in the whole study area with the exclusion of the depth range 0-30 km of
sector E. The two figures remark the role of the shallow parts of rifting zone RZ and of the sector
east of it concerning the introduction of stress heterogeneity in the whole dataset.

[Figure]

Figure S1

[Figure]

Figure S2

[Figure]

Figure S3

[Figure]

Figure S4

[Figure]

Figure S5

[Figure]

Figure S6

[Figure]

Figure S7

[Figure]

Figure S8

[Figure]

Figure S9

[Figure]

Figure S10

[Figure]

Figure S11

[Figure]

Figure S12

[Figure]

Figure S13

[Figure]

Figure S14

[Figure]

Figure S15

**Information concerning our reply of 19 December 2018 to Reviewer 1 (changes made in the MS)**

With reference to **Comments by Reviewer 1** we have already replied with a document uploaded on the SE website on December 19, 2018. In that document we specified that we were going to make some changes to the MS. Here we indicate the changes made according to the Reviewer 1 requests by using the numbered lines of the Chapter "Changes" positioned above in the present document.

- Concerning the use of Gephart and Forsyth (1984) method and the presentation/interpretation of results, see our Replies to the Comments B1 to B5 by Reviewer 2. In these replies we also indicate the changes made in the MS.
- Concerning the request of better clarifying the author's workflow, please see the Chapter "Changes" lines 113-224.
- Concerning the comparison of our earthquake locations with the original ones of the catalogue, see our Replies to the Comments A1 and A4 by Reviewer 2. In these replies we also indicate the changes made in the MS.
- Concerning the request of additional information on stress inversion runs, see our Replies to the Comments B2 to B4 by Reviewer 2. In these replies we also indicate the changes made in the MS, with particular reference to the inclusion of a section of Supplementary material.
- Concerning the requests of (i) reducing the description of the past geodynamic models and (ii) expanding the description of Data, methods and results, see (i) our Reply to the Comment on "*Organization, Figures and Tables*" by Reviewer 2 and (ii) our reorganization of the paragraph "Data, methods ...." in the Chapter "Changes" lines 113-224.
- Concerning the request of improving the English style we will work for this.
- Concerning the uncertainties of focal mechanisms used for stress inversion, we have specified (lines 176-184 in the Chapter "Changes") the uncertainties of the different fault plane solutions coming from different bibliographic sources.
- Concerning the request of a more detailed description of the contribution of Bayloc locations to our conclusions we have made some integrations to the text, see Chapter "Changes" lines 293-326, 334-348, 387-401.